# Local Anti-Concentration Class: Logarithmic Regret for Greedy Linear Contextual Bandit

**Seok-Jin Kim**
Columbia University
New York, NY, USA
seok-jin.kim@columbia.edu

**Min-hwan Oh**
Seoul National Univeristy
Seoul, South Korea
minoh@snu.ac.kr

## Abstract

We study the performance guarantees of exploration-free greedy algorithms for the linear contextual bandit problem. We introduce a novel condition, named the *Local Anti-Concentration* (LAC) condition, which enables a greedy bandit algorithm to achieve provable efficiency. We show that the LAC condition is satisfied by a broad class of distributions, including Gaussian, exponential, uniform, Cauchy, and Student's $t$ distributions, along with other exponential family distributions and their truncated variants. This significantly expands the class of distributions under which greedy algorithms can perform efficiently. Under our proposed LAC condition, we prove that the cumulative expected regret of the greedy algorithm for the linear contextual bandit is bounded by $\mathcal{O}(\mathrm{poly}\log T)$. Our results establish the widest range of distributions known to date that allow a sublinear regret bound for greedy algorithms, further achieving a sharp poly-logarithmic regret.

## 1 Introduction

In the contextual multi-armed bandit problem [2, 6, 24, 25], an agent uses revealed contextual information in each round to decide which arm to pull, receiving a reward corresponding to the pulled arm. The stochastic version of this problem observes rewards as random samples, with their expectation tied to the arm's contextual information. The agent's goal is to design a sequential arm-pulling strategy to maximize cumulative rewards, necessitating a balance between exploration and exploitation. Linear contextual bandits, where expected reward is modeled as a linear function of contextual information, serve as the fundamental framework for contextual bandits [1, 10, 26]. Various exploration strategies, including upper confidence bound (UCB) [1, 11], Thompson sampling (TS) [3, 4], and $\epsilon$-greedy [19] are widely used and studied in the theoretical analysis for linear contextual bandits. However, exploration can often be challenging in practice, possibly leading to over-exploration and performance deterioration. Some domains may find exploration infeasible or even unethical, and it may appear unfair in applications such as healthcare and clinical domains. Furthermore, exploration strategies tend to add complexity for algorithm designers and decision-making systems.

A greedy policy, i.e., pure exploitation without exploration, selects arms greedily based on current problem parameter estimates. While a greedy policy's effectiveness cannot be guaranteed in general since it may fail to find optimality in the worst case, the possibility of its favorable performances in certain scenarios has been of interest both practically and theoretically. Therefore, understanding when a greedy policy can perform effectively, i.e., when exploration is not needed, is a fundamental research question. Recently, a simple greedy policy has been proved to achieve near-optimal regret bounds for linear contextual bandit problems under some stochastic conditions of contexts [8, 20, 30, 33, 34]. Such efficient learning is possible if the greedy policy can benefit from suitable *diversity* in the contexts (or the features of arms) — so that even with exploration-free action selection, parameter estimation is

38th Conference on Neural Information Processing Systems (NeurIPS 2024).

effectively possible. However, distributions known in the existing literature to allow efficient greedy algorithms are mostly limited only to Gaussian and uniform distributions [8, 20, 30, 33]. Hence, the following research questions arise.

*Is it possible for a wider range of distributions to allow efficient learning for greedy algorithms? If so, how can we characterize such distributions?*

We answer the above questions affirmatively by proposing a general distributional condition that allows for a broad range of distributions to achieve provable efficiency of greedy linear bandit algorithms. In this work, we present a new *Local Anti-Concentration* (LAC) condition for distributions that encompasses a wider range of context distributions compared to the previous findings. We demonstrate that the class of distributions that satisfy LAC, which we denote as *LAC class*, include Gaussian, exponential, Cauchy, Student's $t$, and uniform distributions, as well as other exponential family distributions and their truncated variants. This study provides the first evidence that greedy algorithms can perform efficiently beyond Gaussian and uniform distributions. Our findings significantly expand the class of admissible distributions that are suitable for greedy algorithms for linear contextual bandits.

Our proposed LAC condition not only broadens the class of permissible distributions for greedy bandit algorithms but also facilitates a sharper regret guarantee, achieving a poly-logarithmic $\mathcal{O}(\text{poly} \log T)$ regret for greedy algorithms. Our regret analysis constitutes a distinct improvement over the previously known results for greedy linear contextual bandit algorithms. The existing results are primarily categorized into two folds: (i) Gaussian-distributed contexts could only yield $\mathcal{O}(\sqrt{T})$ regret for greedy algorithms for single-parameter linear contextual bandits [20, 30, 33]; (ii) Context diversity (e.g., Assumption 3 in [8]) alone was previously regarded as not sufficient to derive a poly-logarithmic regret but additionally assuming a margin condition (e.g., Assumption 2 in [8]) can achieve $\mathcal{O}(\text{poly} \log T)$ regret.[1] In either case, there are limited prior results about context diversity beyond Gaussian and uniform distributions. As for the margin condition, to the best of our knowledge, no prior work in greedy contextual bandits has rigorously derived the scaling of the margin constant, rather than simply treating it as a universal constant. To this end, we establish that Gaussian and uniform distributions as well as all of the common distributions that satisfy the LAC condition (see Table 1) induce $\mathcal{O}(\text{poly} \log T)$ regret *without* having to additionally assume a margin condition.

The key difference between the analysis of greedy algorithms and that of exploration-based algorithms, such as UCB and TS, for linear contextual bandits lies in the estimation bounds. While UCB [1] and TS [4] analyses involve bounding the weighted estimation error of the parameter using self-normalized martingales, the analysis of greedy algorithms relies on the $\ell_2$ estimation bound in all directions. Ensuring this estimation consistency is more challenging, especially when actions are chosen adaptively, resulting in non-i.i.d. data.

In this work, we prove that for a broad class of context distributions, $\sqrt{t}$-consistency of the estimator can be guaranteed, enabling poly-logarithmic regret for greedy algorithms. Our newly proposed class of context distributions represents the largest known class from which $\sqrt{t}$-consistency of the estimator can be derived, even with adaptively chosen (non-i.i.d.) contexts. To establish this consistency, we derive two key technical results. First, we show that the minimum eigenvalue of the Gram matrix increases sufficiently under the LAC condition. Additionally, we demonstrate that under this condition, the suboptimality gap can be bounded probabilistically—a result derived from our analysis rather than assumed explicitly.

## 1.1 Contributions

The main contributions of our paper are summarized as follows:

- We propose a novel condition, called *Local Anti-Concentration* (LAC) condition, for a greedy linear contextual bandit algorithm to achieve provable efficiency. The newly proposed

---

[1]It is important to note that the problem setting of Bastani et al. [8] (multiple parameters with shared context) differs from our setting, which involves a single parameter with separate contexts, as is predominantly studied in the linear contextual bandit literature [1, 4, 20, 30, 33]. While Bastani et al. [8] demonstrated the diversity condition and the existence of a margin for Gaussian, uniform, and Gibbs distributions in the two-armed case within multi-parameter settings, we show in Appendix K that the Gibbs distribution does not satisfy the diversity condition when extended to cases with more than two arms.

Table 1: Comparisons of Greedy Linear Contextual Bandit Studies

| Paper | Results | | Problem Setting |
| | Context Distribution | Regret Bound | |
| --- | --- | --- | --- |
| Kannan et al. [20] | Gaussian | $\widetilde{\mathcal{O}}(\sqrt{T})$ | Single parameter |
| Sivakumar et al. [33] | Gaussian | $\widetilde{\mathcal{O}}(\sqrt{T})$ | Single parameter |
| Raghavan et al. [30] | Gaussian | $\widetilde{\mathcal{O}}(T^{\frac{1}{3}})^{\dagger}$ | Single parameter |
| Bastani et al. [8]$^{\ddagger}$ | Gaussian Uniform | $\mathcal{O}(\text{poly}\log T)$ | Multiple parameters |
| **This work** | Gaussian Uniform Laplace Truncated exponential Truncated Student's $t$ Truncated Cauchy PDF $f \propto \exp(-\pi)$ with polynomially growing $\pi$ | $\mathcal{O}(\text{poly}\log T)$ | Single parameter |

†   The regret bound of Raghavan et al. [30] is shown in the Bayesian regret, which is a weaker notion of regret than the frequentist regret that we consider in our work.

‡   The problem setting of Bastani et al. [8] is a multi-parameter linear contextual bandit, where a context vector is shared across the arms, but each arm has a separate parameter. Bastani et al. [8] show that Gaussian and uniform distributions satisfy their covariate diversity condition (Assumption 3 in [8]) and the margin condition. For the two-armed case ($K = 2$), they also show that Gibbs distribution satisfies those conditions. However, we show in Appendix K that Gibbs distribution fails to satisfy the conditions for the multi-armed cases with $K \geq 3$.

      LAC condition is satisfied by a wide rage of common distributions, including Gaussian, exponential, Cauchy, Student's $t$, and uniform distributions, and many common distributions, as well as their truncated variants. This significantly expands the class of admissible distributions that are suitable for greedy algorithms and is, to our best knowledge, by far the largest class of distributions that induces efficient learning for greedy algorithms.

- Under our proposed LAC condition, we prove that the cumulative expected regret for the greedy algorithm is bounded by $\mathcal{O}(\text{poly}\log T)$ (Theorem 1), the sharpest known bound for greedy algorithms in linear contextual bandits with a single parameter.

- By leveraging the proposed condition, we can guarantee both (i) the growth of the minimum eigenvalue of the Gram matrix and (ii) a probabilistically bounded suboptimality gap. These two steps are key technical components for analyzing greedy bandit algorithms and were explicitly assumed in existing literature [8] to achieve poly-logarithmic regret. Notably, we do not assume these steps; instead, we prove that distributions satisfying the LAC condition inherently induce these two technical results (Theorems 2 and 3), which may be of independent interest.

- Various context distributions have been empirically shown to allow favorable performances of greey algorithms (see Appendix M). However, the distributions previously known in the literature that enable efficient greedy algorithms were primarily limited to Gaussian and uniform distributions. Our theoretical results offer a significant step toward bridging this gap between theory and practice, providing insights into why greedy algorithms can be effective under a wide range of distributions.

## 1.2   Related Work

Linear contextual bandits and generalized linear bandits have been widely studied [1, 2, 4, 6, 10, 11, 18, 23, 27, 32]. Upper confidence bound (UCB) algorithms for the linear contextual bandit have been proposed and analyzed for their regret performance [1, 6, 10, 11, 32]. Thompson sampling [35] algorithms for linear contextual bandits have also been widely studied, with results demonstrating

their effectiveness both theoretically and empirically [3, 4, 9]. While UCB [1] and Thompson sampling [3, 4] analyses rely on bounding the weighted estimation error of the parameter using self-normalized martingales, the analysis of greedy bandit algorithms depends on the $\ell_2$ estimation bound in all directions. Ensuring this estimation consistency is more challenging, especially in adaptive action settings where data are not i.i.d.

Recent studies [8, 20, 30, 33] have shown that a greedy algorithm can achieve near-optimal regret performance for linear contextual bandit problems under stochastic contexts by providing sufficient conditions under which the greedy algorithm can be efficient. These conditions typically focus on the diversity of the context distribution, ensuring that the greedy policy benefits from sufficient *context diversity* for effective parameter estimation even without exploratory actions.

However, the existing literature has mainly limited itself to Gaussian [8, 20, 30, 33] and uniform [8] distributions, leaving open questions about broader applicability. Specifically, it is unclear if other distributions could also support efficient greedy algorithms and what fundamental characteristics these distributions should have to enable consistent parameter estimation without exploration. Our work addresses this gap by identifying broader conditions under which diverse distributions can effectively support greedy algorithms in linear contextual bandits.

## 2 Preliminaries

### 2.1 Notations

We use $\|x\|_p$ to denote the $\ell_p$-norm of vector $x \in \mathbb{R}^d$. For a positive definite matrix $A \in \mathbb{R}^{d \times d}$, we define $\|x\|_A = \sqrt{x^\top A x}$. We use $\lambda_{\min}(A)$ to denote the minimum eigenvalue of the positive definite matrix $A$. We denote $\mathbb{D}_R^d := [-R, R]^d$ and $\mathbb{B}_R^d := \{x \in \mathbb{R}^d : \|x\|_2 \leq R\}$. If $d$ is clear, we just write $\mathbb{B}_R^d := \mathbb{B}_R$ and $\mathbb{D}_R^d := \mathbb{D}_R$. We define $[n]$ for a set $[n] := \{1, 2, \ldots, n\}$. We write $\mathbb{S}^{d-1}$ for a $d$-dimensional unit sphere. We set $\|X\|_{\psi_1} = \sup_{p \geq 1}\{p^{-1}\mathbb{E}^{1/p}|X|^p\}, \|X\|_{\psi_2} = \sup_{p \geq 1}\{p^{-\frac{1}{2}}\mathbb{E}^{1/p}|X|^p\}$ for a random variable $X$. If $X$ is a $d$-dimensional random vector, then we write $\|X\|_{\psi_2} = \sup_{\|u\|_2=1} \|\langle u, X \rangle\|_{\psi_2}, \|X\|_{\psi_1} = \sup_{\|u\|_2=1} \|\langle u, X \rangle\|_{\psi_1}$. We use the notation $\mathcal{O}()$ or $\lesssim$ to hide constants, and $\widetilde{\mathcal{O}}()$ to hide constants and logarithmic terms. We use the notation $a \asymp b$ when $a \lesssim b$ and $b \lesssim a$. We use $c, c_1, c_2 \ldots$ for absolute constant, which *may differ from line by line*.

### 2.2 Linear Contextual Bandits with Stochastic Contexts

We consider the linear contextual bandit problem with $K$ arms ($K \geq 2$), where in each round $t = 1, 2, \ldots, T$, the set of context vectors $\mathcal{X}(t) = \{X_i(t) \in \mathbb{R}^d, i \in [K]\}$ is drawn from some unknown distribution $P_{\mathcal{X}}(t)$. Each arm's feature $X_i(t) \in \mathcal{X}(t)$ for $i \in [K]$ need not be independent of each other and can possibly be correlated. The agent then pulls an arm $a(t) \in [K]$. Each context vector $X_i(t)$ for $i \in [K]$ is associated with stochastic reward $Y_i(t) \in \mathbb{R}$ with mean $X_i(t)^\top \theta^\star$ where $\theta^\star \in \mathbb{R}^d$ is a fixed, *unknown* parameter. For simplicty, we assume $\|\theta^\star\|_2 \leq 1$. After pulling arm $a(t)$, the agent receives a stochastic reward $Y_{a(t)}(t)$ as a bandit feedback: $Y_{a(t)}(t) = X_{a(t)}(t)^\top \theta^\star + \eta_{a(t)}(t)$, where $\eta_{a(t)}(t) \in \mathbb{R}$ is a zero mean noise. We assume that there is an increasing sequence of sigma fields $\{\mathcal{H}_t\}$ such that each $\eta_{a_t}(t)$ is $\mathcal{H}_t$-measurable with $\mathbb{E}[\eta_{a_t}(t)|\mathcal{H}_{t-1}] = 0$. In our problem, $\mathcal{H}_t$ is the sigma field generated by random variables of the arms chosen $\{a(1), ..., a(t)\}$, their context vectors $\{X_{a(1)}(1), ..., X_{a(t)}(t)\}$, and the corresponding rewards $\{Y_{a(1)}(1), ..., Y_{a(t)}(t)\}$. Also, $\eta_{a(t)}(t)$ is assumed to be conditionally $\sigma$-sub-Gaussian, i.e., for all $\lambda \in \mathbb{R}, \mathbb{E}[e^{\lambda \eta_{a(t)}(t)} \mid \mathcal{H}_{t-1}] \leq \exp(\lambda^2 \sigma^2/2)$ for $\sigma \geq 0$. Observing context vector $\mathcal{X}(t)$, let $a^*(t)$ denote the optimal arm in round $t$, that is, $a^*(t) = \arg\max_{i \in [K]} X_i(t)^\top \theta^\star$. Then the *instantaneous* expected regret ($\text{reg}(t)$) and *cumulative* expected regret ($\mathbf{Reg}(T)$) are defined respectively as

$$\mathbf{Reg}(T) := \sum_{t=1}^T \text{reg}(t) := \sum_{t=1}^T \mathbb{E}\left[X_{a^\star(t)}(t)^\top \theta^\star - X_{a(t)}(t)^\top \theta^\star\right]$$

which are respectively the instantaneous and cumulative differences between the optimal expected reward and the expected reward of the pulled arms. The expectation is taken with respect to the stochasticity of history, containing randomness of contexts. The goal of the agent is to minimize the cumulative expected regret.

## 2.3 `LinGreedy`: Exploration-Free Algorithm for Linear Contextual Bandits

In this work, we focus on identifying sufficient conditions that enable exploration-free greedy algorithms to efficiently learn the optimal policy. Specifically, we analyze a greedy algorithm for linear contextual bandits, which we refer to as `LinGreedy` (Algorithm 1). The `LinGreedy` algorithm selects arms greedily based on the OLS estimator, without any exploratory actions.

---

**Algorithm 1** `LinGreedy`: Greedy Linear Contextual Bandit

---

Initialize $\Sigma(0) = 0 \cdot I_d$, $b(0) = \mathbf{0}$, $\theta_0 \in \mathbb{R}^d$.
**for** $t \in [T]$ **do**
    **while** $\lambda_{\min}\big(\Sigma(t-1)\big) = 0$ **do**
        Choose $a(t) = \arg\max_{i \in [K]} X_i(t)^\top \theta_0$ and observe reward $Y_{a(t)}$.
        Update $b(t) = b(t-1) + X_{a(t)}(t)Y_{a(t)}$ and $\Sigma(t) = \Sigma(t-1) + X_{a(t)}(t)X_{a(t)}(t)^\top$.
    **end while**
    Choose $a(t) = \arg\max_{i \in [K]} X_i(t)^\top \hat{\theta}_{t-1}$ and observe reward $Y_{a(t)}$.
    Update $b(t) = b(t-1) + X_{a(t)}(t)Y_{a(t)}$ and $\Sigma(t) = \Sigma(t-1) + X_{a(t)}(t)X_{a(t)}(t)^\top$.
    Update $\hat{\theta}_t = \Sigma(t)^{-1}b(t)$.
**end for**

---

**Description of Algorithm 1.** The algorithm performs a greedy action in each round based on estimated rewards. In the initial rounds, when the Gram matrix $\Sigma(t)$ is not yet invertible, parameter estimation is deferred, and the algorithm selects actions based on an initial parameter $\theta_0$. Once the Gram matrix becomes invertible—which can be shown with high probability after sufficient time, the algorithm computes an OLS estimator and performs a greedy action based on the estimated parameter in each subsequent round. This algorithm is exploration-free. In the following sections, we present a novel and more general condition that enables efficient learning for greedy algorithms.

# 3 Local Anti-Concentration Class

In this section, we introduce a new sufficient condition for efficient greedy contextual bandits. This condition is general and encompasses a wide range of common distributions, including Gaussian, exponential, uniform, Cauchy, and Student's $t$ distributions, as well as their truncated variants. To the best of our knowledge, this is the most extensive class of distributions considered in the greedy contextual bandit literature [8, 20, 28, 30, 31, 33], which has primarily focused on Gaussian, uniform distributions, and their truncated variants.

Our proposed condition centers on the rate of the log density of stochastic contexts, a concept we term *Local Anti-Concentration* (LAC). We now formally introduce the novel LAC class.

**Definition 1 (Local Anti-Concentration (LAC))** *A density function $f_X$ of a random variable $X \in \mathbb{R}^n$ is said to satisfy the Local Anti-Concentration (LAC) condition with a non-decreasing polynomial $\mathcal{L}$ if*
$$\|\nabla \log f_X(x)\|_\infty \leq \mathcal{L}(\|x\|_\infty)$$
*for all $x \in \mathbb{R}^n$. We refer to $\mathcal{L}$ as the LAC function of $X$. We denote the class of distributions that satisfy this LAC condition as the Local Anti-Concentration class.*

## 3.1 Intuition of LAC Condition

The LAC condition implies that a density is not overly concentrated at any given point, leading to a gradual decay in density across all directions—hence the term *local* anti-concentration. A geometric interpretation of the LAC condition and a rigorous definition of this decay rate are provided in Appendix D. Section 3.2 demonstrates that the LAC condition applies to a broad range of common distributions. To the best of our knowledge, very few distributions have been previously shown to support efficient performance guarantees for greedy algorithms. However, we prove that the LAC condition holds for a wide range of distributions, including a variety of exponential families. Note that $\mathcal{L}$ can be a constant when contexts have bounded support (see Appendix C). In the following sections, we further explore the characteristics of the LAC condition.

## 3.2 Generality of LAC Condition

We show that the LAC condition is applicable to various distributions, significantly expanding the class of admissible distributions for greedy linear contextual bandits. The LAC condition is satisfied when the exponential component in the exponential family has a polynomial scale. Included in the LAC class are distributions such as Gaussian, exponential, uniform, Student's $t$, and Cauchy distributions, along with their truncated variants.

The following proposition demonstrates that the LAC function does not directly depend on the dimension of $X$. We further discuss in Appendix C that the LAC condition is more closely related to the correlation structure of $X$ rather than its dimensionality. Therefore, we suggest that the LAC condition provides a suitable framework for comparing regret when both the number of arms and the dimension are large.

**Proposition 1** *Suppose the random variable $X = (X_1, X_2)$, where $X_1 \in \mathbb{R}^{n_1}$ and $X_2 \in \mathbb{R}^{n_2}$, consists of two independent components. If $X_1$ and $X_2$ satisfy the LAC condition with functions $\mathcal{L}_1(\cdot)$ and $\mathcal{L}_2(\cdot)$, respectively, then $X$ satisfies the LAC condition with $\mathcal{L}(x) = \max(\mathcal{L}_1(x), \mathcal{L}_2(x))$.*

This holds because, when we take the logarithm of the density, the independent coordinates decompose as the sum of each density. Upon taking the gradient and evaluating the $\ell_\infty$ norm, the expression decomposes perfectly. Using this proposition, the LAC condition remains robust across dimensions if the coordinates are independent, making it dimension-free in such cases. Furthermore, this condition is very accessible because it can be readily computed for a given density function. For many well-known exponential families, the exponential component of the density often scales polynomially.

**Examples of Distributions with LAC Condition**
We present a few examples of known distributions satisfying the LAC condition. We provide rigorous proofs for the examples in Appendix C.

- **Gaussian distribution:** For a Gaussian random variable $X = (x_1, \ldots, x_n) \sim N(\mu, \Sigma)$, if $\Sigma$ is diagonal, it satisfies the LAC condition with $\mathcal{L}(x) = \frac{4}{\lambda_{\min}(\Sigma)}(\|x\|_\infty + \|\mu\|_\infty)$. For the general (non-diagonal) case of $\Sigma$, see Appendix C.

- **Exponential distribution:** The exponential distribution's density $f_X(x) = \frac{1}{\lambda}\exp(-\lambda x)$ satisfies the LAC condition with a constant function $\mathcal{L}(x) = \lambda$.

- **Uniform distribution:** The uniform distribution has constant density and satisfies the LAC condition with a constant function $\mathcal{L}(x) = 1$.

- **Student's $t$-distribution:** The 1-dimensional Student's $t$-distribution has density $f_X(x) = \frac{\Gamma(\frac{\nu+1}{2})}{\sqrt{\nu\pi}}\Gamma(\frac{\nu}{2}) \cdot (1 + \frac{x^2}{\nu})^{-(\nu+1)/2}$ and satisfies the LAC condition with $\mathcal{L}(x) = c_\nu$ for some $\nu$ dependent constant $c_\nu > 0$.

- **Laplace distribution:** The Laplace distribution has density $f(x) = \frac{1}{2b}\exp\left(-\frac{|x-\mu|}{b}\right)$ and satisfies the LAC condition with $\mathcal{L}(x) = c$ for some constant $c > 0$.

If each coordinate's density independently adheres to one of the aforementioned distributions, according to Proposition 1, they all share the same LAC function irrespective of the dimension.

Consider the density $f(x)$ with $f(x) \propto \exp(-V(x))$ for some differentiable function $V(x)$. If $\nabla V(x)$ has polynomial growth, i.e., is bounded by a polynomial, then the density $f(x)$ meets the LAC condition. This holds because $f(x) = C\exp(-V(x))$, $\nabla \log f(x) = \nabla \log \exp(-V(x)) = -\nabla V(x)$. If $\nabla V(x)$ exhibits polynomial-scale growth, then the supremum norm confirms that the LAC condition is satisfied.

This observation makes the LAC condition easily verifiable for exponential family distributions with density forms $f_X(x|\theta) = h(x)\exp[\eta(\theta) \cdot T(x) - A(\theta)]$, where the exponential part $T(x)$ has polynomial growth. In many exponential family cases, $T(\cdot)$ indeed exhibits polynomial growth. Proofs and further details can be found in Appendix C.

## 4 Statistical Challenges of Greedy Linear Contextual Bandits

In this section, we outline the key statistical challenges in analyzing greedy algorithms for linear contextual bandits: (i) ensuring the diversity of the adapted Gram matrix (Section 4.1) and (ii)

bounding the suboptimality gap to achieve logarithmic regret (Section 4.2). For ease of exposition, we use the vectorized context expression $\mathbf{X}(t) = (X_1^\top(t), \ldots X_K^\top(t)) \in \mathbb{R}^{dK}$ of $\mathcal{X}(t)$, which combines context vectors $X_i(t)$ for $i \in [K]$. We define $X_{ij}(t)$ as the $j$-th coordinate of the context $X_i(t)$.

## 4.1 Diversity of Adapted Gram Matrix

The first key challenge lies in ensuring sufficient $\ell_2$-concentration of the estimator. This requires sufficient eigenvalue growth of the Gram matrix, constructed from the *policy-selected* contexts. For the OLS estimator used in Algorithm 1, if the minimum eigenvalue of the adapted Gram matrix $\lambda_{\min}(\Sigma(t))$ increases linearly with the number of rounds $t$, we can obtain the high probability $\ell_2$ error bound $\|\hat{\theta}_t - \theta^\star\|_2$ with a convergence rate of $\mathcal{O}(1/\sqrt{t})$ using martingale concentration [13, 29]. In fact, the growth of $\lambda_{\min}(\Sigma(t))$ is a necessary condition to obtain $\sqrt{t}$-consistency of the estimator.

However, estimating the covariance of the selected contexts $X_{a(t)}(t)$ is relatively challenging, as its distribution differs significantly from the overall distribution (before selection) of $\mathbf{X}(t)$. Some studies have investigated the statistical properties of selected contexts [20, 28]; however, the known results are limited to specific distributions, such as arm-independent Gaussian and uniform distributions. Therefore, it remains an open question whether the growth of $\lambda_{\min}(\Sigma(t))$ can be ensured for a broader class of distributions and what characteristics such distributions would need to satisfy.

## 4.2 Bounding Suboptimality Gap: Road to Logarithmic Regret

The next challenge that we face particularly in order to achieve logarithmic regret is to bound the suboptimality gap. We first denote the *suboptimality gap* as the difference between the optimal expected reward and the second highest expected reward:

$$\Delta(\mathbf{X}(t)) := X_{a^\star(t)}(t)^\top \theta^\star - \max_{i \neq a^\star(t)} X_i(t)^\top \theta^\star,$$

which is determined by the true parameter $\theta^\star$. We aim to bound this suboptimality gap probabilistically, as described precisely in Challenge 2 of Section 4.3.

When this challenge is resolved, along with the growth of the minimum eigenvalue of the adapted Gram matrix discussed in Section 4.1, we can achieve logarithmic expected regret using analysis techniques for linear contextual bandit with stochastic contexts [7, 19]. A high-level description of the role of the margin constant in the regret bound is provided in Appendix D.1, with a rigorous analysis in Appendix J.

## 4.3 Formal Statements of Two Key Challenges

As mentioned above, we encounter two primary challenges: ensuring the diversity of the chosen contexts (i.e., the growth of the minimum eigenvalue of the Gram matrix of the selected contexts) and bounding the suboptimality gap. Importantly, we do not assume these conditions to hold a priori; rather, we will demonstrate that they are satisfied in the stochastic context under the LAC condition. In this section, we formally define these challenges to be addressed.

Before delving into the formal statements for each of the two challenges, we first define the concept of the diversity constant, which depends on the minimum eigenvalue of the adapted Gram matrix.

**Definition 2 (Diversity Constant)** *For a linear contextual bandit with contexts $\mathbf{X}(t)$ and history $\mathcal{H}_{t-1}$, the diversity constant $\lambda_\star(t)$ is defined as the value satisfying*

$$\mathbb{E}[X_{a(t)}(t)X_{a(t)}(t)^\top \mid \mathcal{H}_{t-1}] \succeq \lambda_\star(t)I_d, \tag{1}$$

*for all $t > 0$, where $a(t)$ denotes the arm selected by the algorithm in round $t$.*

Then, the first challenge is to ensure a positive diversity constant $\lambda_\star(t) > 0$, which involves sufficient eigenvalue growth of the adapted Gram matrix. We explore this further in Appendix F.1.

**Challenge 1 (Positive Diversity Constant)** *Our goal is to ensure $\lambda_\star(t) > 0$.*

Achieving a positive diversity constant is challenging, as it requires analyzing the behavior of a context selected by the greedy policy in a specific direction rather than relying on the overall context distribution. In Section 5.3.1, we demonstrate that the minimum eigenvalue of the Gram matrix grows sufficiently, thereby ensuring a positive diversity constant.

We now formally state our second challenge of bounding the suboptimality gap.

**Challenge 2 (Probabilistic Suboptimality Gap)** *We aim to bound the constant $C_\Delta(t)$, which holds under the given history $\mathcal{H}_{t-1}$ and for any $\varepsilon > 0$,*

$$\mathbb{P}\left[\Delta\big(\mathbf{X}(t)\big) \leq \varepsilon\right] \leq \varepsilon C_\Delta(t) + \frac{1}{\sqrt{T}} \,. \tag{2}$$

*We also refer to this constant $C_\Delta(t)$ as the margin constant.*

Note that Eq. (2) is a relaxed version of the margin condition presented in [5, 7, 8, 19]. The aforementioned literature explicitly assumes this condition to hold. However, we instead show that the suboptimality gap can be bounded without directly assuming it (Section 5.3.2). Rigorously, $C_\Delta(t)$ depends on $T$, as it is a function of $T$. However, we emphasize that our algorithm does not require prior knowledge of $T$; this dependency is needed only for the analysis.

## 5 Regret Analysis

We present the main results of our paper. We prove that the regret of the greedy algorithm (Algorithm 1) for linear contextual bandits can be bounded at a logarithmic scale in the time horizon $T$, provided that the context distribution satisfies the LAC condition with a polynomial function $\mathcal{L}$.

**Assumption 1 (Independently distributed contexts)** *The context sets $\mathcal{X}(1), \ldots, \mathcal{X}(T)$ are independently distributed across time.*

**Discussion of Assumption 1.** To the best of our knowledge, all analyses of greedy linear contextual bandits assume the independence and identical distribution (i.i.d.) of context sets [8, 20, 28, 33]. In Assumption 1, we only require context sets to be independent; they may be non-identically distributed. Additionally, much of the literature on linear contextual bandits that investigates $\sqrt{t}$-consistency of estimators also assumes independence of context sets [21, 22]. Note that under Assumption 1, context vectors within the same round are permitted to be dependent.

### 5.1 Considerations for Context Boundedness

We first provide detailed considerations on context boundedness. In the linear contextual bandit setting, $\ell_2$ boundedness is commonly assumed. However, for light-tailed distributions (such as Gaussian or exponential), the $\ell_2$ norm is unbounded. In such cases, a general approach in the statistical literature is to assume bounded $\psi_1$ or $\psi_2$ norms [14, 17, 37, 38]. Therefore, we divide our analysis into cases of bounded and unbounded contexts.

**Bounded Contexts vs. Unbounded Contexts.** In linear contextual bandit studies, boundedness of the $\ell_2$ norm of contexts is commonly assumed. In this paper, for bounded contexts, we consider both truncated contexts (e.g., truncated Gaussian, truncated Cauchy distributions) and naturally bounded contexts (e.g., uniform distribution). For unbounded contexts, we assume a bounded $\psi_1$ norm, which is a standard assumption for handling light-tailed distributions (e.g., as in [14, 17], which assume bounded $\psi_2$ norms).

**Assumption 2 (Boundedness)** *For unbounded contexts, we assume $\|X_i(t)\|_{\psi_1} \leq x_{\max}$. For bounded contexts, we assume $\|X_i(t)\|_2 \leq x_{\max}$ for all $i \in [K], t \in [T]$.*

**Discussion of Assumption 2** The bounded context assumption is widely used in the literature [1, 7, 8, 21, 22, 28]. For unbounded contexts, our assumption of $\psi_1$ boundedness is notably weaker than the sub-Gaussianity (or $\psi_2$) assumption commonly used in statistical regression literature to handle random design covariates [14, 39]. If $\ell_2$ boundedness holds, it automatically implies boundedness of the $\psi_1$ norm. However, as the analysis differs slightly depending on whether the support is restricted to a bounded ball or is unbounded, we address these cases separately.

## 5.2 Regret Bound of LinGreedy for LAC Distribution

We first introduce our main result, the regret bound of the greedy algorithm under the LAC condition.

**Theorem 1 (Regret bound of `LinGreedy`)** *Suppose $\mathbf{X}(t)$ satisfies the LAC condition with the polynomial function $\mathcal{L}$ and also satisfies Assumptions 1 and 2 for all $t$. Then, the cumulative expected regret of `LinGreedy` (Algorithm 1) is bounded by*

$$\mathbf{Reg}(T) \leq \mathcal{O}(\mathrm{poly}\log T),$$

*where $\mathcal{O}$ concerns only the dependency on $T$. Considering the dependency on $d$ and $K$, for unbounded contexts, we have*

$$\mathbf{Reg}(T) \leq \widetilde{\mathcal{O}}(d^{2.5}).$$

*For bounded contexts, refer to Appendix H for explicit results, as we consider several cases.*

**Discussion of Theorem 1.** Theorem 1 states that if the contexts are drawn from a distribution in the LAC class, the regret scales as $\mathcal{O}(\mathrm{poly}\log T)$. While our primary objective is not solely to achieve the sharpest regret bounds, attaining poly-logarithmic regret is highly favorable. Our main goal is to demonstrate that a large class of context distributions satisfies the LAC condition. When they do, a simple greedy algorithm can suffice or even outperform exploration-based algorithms (see numerical experiments in Section 6). The worst-case dependence on $d$ and $K$ for bounded contexts is detailed in Appendix H, where dependencies remain at most polynomial in $d$ and $K$. As these dependencies vary across distributions, refer to Appendix H for precise information. Proofs for unbounded contexts are provided in Appendix G, with a proof sketch in Appendix D. Proofs for bounded contexts are included in Appendix I.

Theorem 1 is the first result to expand the class of admissible distributions for greedy bandit algorithms beyond Gaussian and uniform distributions. Our result demonstrates, for the first time, that distributions in the LAC class inherently exhibit margin behavior, achieving sharp poly-logarithmic regret without requiring an additional margin assumption. This finding is of independent interest beyond the analysis of greedy bandit algorithms.

## 5.3 Proof Sketch of Theorem 1

We first present our key results for addressing each of Challenges 1 and 2 stated in Section 4.3.

### 5.3.1 Ensuring the Positive Diversity Constant

In this section, we present our key result for estimating lower bounds on the diversity constant for densities that satisfy the LAC condition, therefore addressing Challenge 1. Theorem 2 is the analysis under the case of unbounded contexts, where contexts have full support. A similar result is presented in Appendix H for contexts with bounded or truncated support.

**Theorem 2 (Diversity constant for unbounded contexts)** *If unbounded contexts $\mathbf{X}(t)$ has the LAC condition with $\mathcal{L}(x) := A_1 + A_2 x^\alpha$ and satisfies Assumption 2 for all $t$,*

$$\lambda_\star(t) \geq c_1 \frac{1}{d} \cdot \frac{1}{(A_1 + A_2(R_1 + 2)^\alpha)^2}$$

*holds for $R_1 := c_2 x_{\max}(\log d + \log K + 2)$. Here, $c_1, c_2$ are absolute constants.*

**Discussion of Theorem 2.** Theorem 2 implies that $\lambda_\star(t) \geq \Omega(\frac{1}{d})$, hence ensuring the growth of the minimum eigenvalue of the adapted Gram matrix. In Appendix D.1, we discuss how $\frac{1}{\lambda_\star(t)}$ factors into the regret bounds. Note that $\alpha$ is generally small in many distributions. For example, for Gaussian distributions, $\alpha = 1$, and for exponential distributions, $\alpha = 0$.

### 5.3.2 Bounding Suboptimality Gap

Next, we present our result addressing Challenge 2 by computing the suboptimality gap constant $C_\Delta(t)$, which satisfies the inequality in Eq.(2) for every $\varepsilon > 0$. By combining this condition with the estimates for $\lambda_\star(t)$, we can obtain an $\mathcal{O}(\mathrm{poly}(\log T))$ regret bound in terms of $T$ by applying the analysis techniques of linear contextual bandits with stochastic contexts [7, 8, 19] to our setting.

**Theorem 3 (Suboptimality gap for unbounded contexts)** *In the same setup as Theorem 2,*

$$C_\Delta(t) \leq c_3\sqrt{d}(A_1 + A_2 3^\alpha R_2^\alpha)\frac{1}{\|\theta^\star\|_2}$$

*holds for $R_2 = c_4 x_{\max}(1 + \log K + \log d + \frac{1}{2}\log T) + 1$ and absolute constants $c_3, c_4 > 0$.*

**Discussion of Theorem 3.** Note that the suboptimality gap constant $C_\Delta(t)$ is multiplied linearly in regret bounds [5, 7, 19]. For bounded contexts, we present a similar result in the Appendix H.

### 5.3.3 Proof Intuitions

We provide a high-level proof overview of Theorem 1 in Appendix D. Note that once Challenges 1 and 2 are satisfied, achieving logarithmic regret becomes straightforward, as detailed in Appendix J.

The remaining task is to address these two challenges, with a particular focus on bounding the constants $\lambda_\star(t)$ and $C_\Delta(t)$. A key implication of the LAC condition is that the density decays slowly at every point. Another useful property is that the LAC condition is preserved under conditioning, meaning that $\mathbf{X}(t) \mid \{\mathbf{X}(t) \in A\}$ also satisfies the LAC condition with the same function for any set $A \subset \mathbb{R}^{K \times d}$. This can be verified using the fact that $\log f_{\mathbf{X}(t)\mid\{\mathbf{X}(t)\in A\}}(x) = \log f_{\mathbf{X}(t)}(x) - \log \mathbb{P}[A]$, where $\mathbb{P}[A]$ is a constant (see Appendix C.3 for details).

The main challenge of analyzing the statistical concentration in greedy linear contextual bandits lies in the fact that the distribution of selected contexts $X_{a(t)}(t)$ differs significantly from the distribution of the overall (pre-selected) contexts $\mathbf{X}(t)$. The preservation of LAC under conditioning ensures that LAC still holds when conditioning on the event of selecting arm $i$, enabling our analysis.

The full proofs for unbounded contexts are provided in Appendix G. For results on bounded contexts, see Appendix H (and their proofs in Appendix I).

## 5.4 $\sqrt{t}$-Consistency of Estimator

In addition to achieving logarithmic regret, an independently valuable result is obtained: the $\ell_2$-consistency of the estimator $\hat{\theta}_t$. This is a property that even typical sublinear-regret algorithms, such as UCB and TS, do not generally guarantee. Under the same setup as Theorem 1, we achieve $\|\hat{\theta}_t - \theta^\star\|_2 \leq \widetilde{\mathcal{O}}\left(\frac{d}{\sqrt{t}}\right)$ with high probability (see Corollaries 6 and 7). This additional result may also facilitate analysis of sample complexity, such as PAC bounds.

## 6 Experiments

To validate our theoretical findings numerically, we conducted experiments using various context distributions: Gaussian, Laplace, uniform, and truncated Cauchy distributions. We compared the performance of `LinGreedy` with the LinUCB and LinTS algorithms. The results showed that `LinGreedy` exhibited significantly superior regret performance compared to the other exploration-based algorithms, achieving a logarithmic scale of regret. Detailed experimental results are provided in Appendix M.

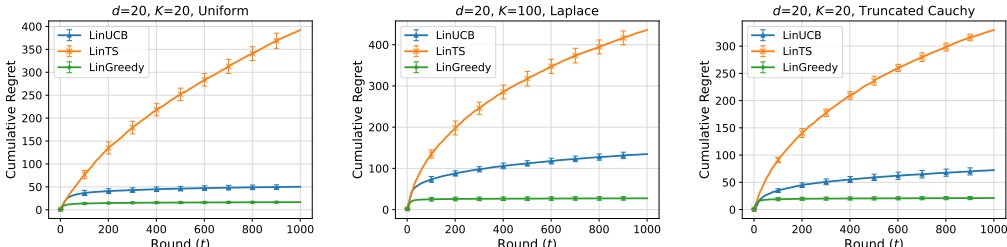

Figure 1: The cumulative regret plots of the numerical experiments. The full results are available in Appendix M.

## Acknowledgements

This work was supported by the National Research Foundation of Korea(NRF) grant funded by the Korea government(MSIT) (No. 2022R1C1C1006859, 2022R1A4A1030579, and RS-2023-00222663) and by AI-Bio Research Grant through Seoul National University. The authors thank H. Choi for helpful discussions.

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

# Appendix

## Contents

# A  Additional Notations

We define additional notation. For $v \in \mathbb{R}^d$, define

$$P_v := \{x \mid x^\top v = 0\}.$$

For $S \subset \mathbb{R}^n$, we define

$$\|S\|_\infty := \sup\{\|x\|_\infty, \ x \in S\}.$$

For the intervals $I \subset \mathbb{R}$ and $v \in \mathbb{R}^d$, define

$$I_v := \{x \mid x^\top v \in I\}.$$

For instance, $[-1, 1]_v := \{x \mid -1 \leq x^\top v \leq 1\}$. We define $\pi_V(\cdot)$ as the projection onto the subspace $V$. Also, in the appendix, we write $\mathbb{B}_R = \mathbb{B}_R^d = \{x \in \mathbb{R}^d \mid \|x\|_2 \leq R\}$ with slight abuse of notation. For a random variable $X$, we define $\mathrm{supp}(X)$ as the support of $X$. Recall that we define the expected regret in round $t$ as $\mathrm{reg}(t)$ and the unexpected regret as $\mathrm{reg}'(t)$. Also, for $\Omega = \bigcup_{i \in I} E_i$ with disjoint events $E_i$, we use the following notation

$$\mathbb{E}\left[\mathbb{E}[X \mid E_i]\right] := \sum_{i \in I} \mathbb{P}[E_i]\mathbb{E}[X \mid E_i].$$

For an event $A$, we define $\mathbb{P}[X; A]$ and $\mathbb{E}[X; A]$ as $\mathbb{P}[X \cap A]$ and $\mathbb{E}[X \cap A]$, respectively, following Durrett [15]'s notation.

# B  $\|\theta^\star\|_2$ Dependency of Suboptimality Gap

Throughout the proof, we first bound the margin constant $C_\Delta$ by assuming $\|\theta^\star\|_2 = 1$. If we obtain $C_\Delta$ for $\|\theta^\star\|_2 = 1$, then in the general case where $\|\theta^\star\|_2 = \kappa$, observe that

$$\mathbb{P}\left[X_{a^\star(t)}(t)^\top \theta^\star - \max_{j \neq a^\star(t)} X_j(t)^\top \theta^\star \leq \varepsilon\right] = \mathbb{P}\left[\frac{X_{a^\star(t)}(t)^\top \theta^\star}{\kappa} - \max_{j \neq a^\star(t)} \frac{X_j(t)^\top \theta^\star}{\kappa} \leq \frac{\varepsilon}{\kappa}\right] \quad (3)$$

$$= \frac{C_\Delta}{\kappa}\varepsilon + \frac{1}{\sqrt{T}} \quad (4)$$

holds. Therefore, from now on, we assume $\|\theta^\star\|_2 = 1$ and adjust for the general case later. Additionally, we highlight that we only need to bound $C_\Delta$ for $\varepsilon < \frac{1}{2}$, since for $\varepsilon > \frac{1}{2}$, it holds that $C_\Delta = 2$.

# C  More Details of LAC: Distributions, Conditioning, & Truncation

In this section, we provide the omitted details of LAC and its properties.

## C.1  LAC of Various Distributions

We first provide more details on the LAC of various distributions, as presented in Section 3.2.

**LAC of Gaussian contexts.** For density $f(x) = C \exp\left(-(x-\mu)^\top V(x-\mu)\right)$, $x \in \mathbb{R}^n$, where $V = \frac{1}{2}\Sigma^{-1}$,

$$\nabla \log f = 2V^\top (x - \mu).$$

First, when $V$ is diagonal, after taking the log and gradient, we get

$$\|\nabla \log f(x)\|_\infty \leq 2\lambda_{\max}(V)\|x - \mu\|_\infty \leq 2\lambda_{\max}(V)\left(\|x\|_\infty + \|\mu\|_\infty\right).$$

Since $V = \frac{1}{2}\Sigma^{-1}$, we obtain the desired results. For general $V$, we can see that

$$\|\nabla \log f(x)\|_\infty \leq 2\left(\max_i \|V^i\|_1\right)\|x - \mu\|_\infty,$$

where $V^i$ is the $i$-th row of $V$.

**LAC of bounded support contexts.** If the support of the contexts is bounded within a compact set, $\nabla \log f$ is bounded provided it is continuous. Therefore, in this case, LAC can be a constant function. Additionally, if $\|x\|_2 \leq R$, we have

$$\|\nabla \log f(x)\|_\infty \leq \mathcal{L}(R)$$

by the monotonicity of $\mathcal{L}(\cdot)$.

**LAC and context correlations.** We claim that the LAC function $\mathcal{L}(\cdot)$ of a random vector $x = (x_1, \ldots, x_n)$ is related to the correlation structure of $x_i, x_j$ rather than the dimension $n$ itself. Consider $f(x) \propto \exp(-V(x))$. Then,

$$\nabla \log f(x) = -\nabla V(x)$$

and

$$\partial_i \log f(x) = -\partial_i V(x)$$

hold. Since $\mathcal{L}$ is defined in the sense of the supremum norm, to investigate the LAC function, the maximum value $\max_i \|\partial_i V(x)\|_\infty$ is important. We claim that it is a dimension-free property, and the correlation structure is more important than the dimension $n$ itself. For instance, if $x = (x_1, \ldots, x_n)$ is coordinate-wise independent, we have $V(x) = V_1(x_1) + V_2(x_2) + \cdots + V_n(x_n)$. Hence,

$$\|\nabla V(x)\|_\infty = \max_i \|V_i'(x_i)\|_\infty$$

and we can see it is a dimension-free value.

**LAC with shifted mean distribution.** Let the mean-zero contexts $X$ have density $f(x)$ with LAC function $\mathcal{L}(\cdot)$. Then, the shifted contexts with mean $\mu$ have density $g(x) = f(x + \mu)$, and we observe that

$$\begin{aligned}
\|\nabla \log g(x)\|_\infty &= \|\nabla \log f(x + \mu)\|_\infty \\
&\leq \mathcal{L}\left(\|x + \mu\|_\infty\right) \\
&\leq \mathcal{L}\left(\|x\|_\infty + \|\mu\|_\infty\right).
\end{aligned}$$

Hence, the density $g(x)$ has LAC with function $\mathcal{L}'(x) = \mathcal{L}(x + \|\mu\|_\infty)$ and when $\|\mu\|_\infty = O(1)$, it has the same rate. Thus, LAC does not require contexts to be mean-zero and can be defined for distributions with a general mean.

## C.2 Proof of Proposition 1

If $\mathbf{X} = (X_1^\top, X_2^\top)$ and $X_1, X_2$ are independent, the density $f_\mathbf{X}$ of $\mathbf{X}$ can be decomposed as

$$f_\mathbf{X}((x_1, x_2)) = f_{X_1}(x_1) f_{X_2}(x_2).$$

Then,

$$\nabla \log f_\mathbf{X}((x_1, x_2)) = \nabla(\log f_{X_1}(x_1) + \log f_{X_2}(x_2)) = (\nabla_{x_1} \log f_{X_1}(x_1), \nabla_{x_2} \log f_{X_2}(x_2))$$

holds, and taking the supremum norm, we get

$$\begin{aligned}
\|\nabla \log f_\mathbf{X}((x_1, x_2))\|_\infty &= \max\left(\|\nabla_{x_1} \log f_{X_1}(x_1)\|_\infty, \|\nabla_{x_2} \log f_{X_2}(x_2)\|_\infty\right) \\
&\leq \max\left(\mathcal{L}_1(\|x_1\|_\infty), \mathcal{L}_2(\|x_2\|_\infty)\right) \\
&\leq \max(\mathcal{L}_1(\|(x_1, x_2)\|_\infty), \mathcal{L}_2(\|(x_1, x_2)\|_\infty))
\end{aligned}$$

where the third inequality holds by the monotonicity of $\mathcal{L}(\cdot)$. By the definition of the LAC function, we finally obtain the desired result: $f$ has LAC with the function $\mathcal{L}(x) = \max(\mathcal{L}_1(x), \mathcal{L}_2(x))$ for $x \in \mathbb{R}$.

∎

## C.3 LAC and Conditioning

First, we observe that the LAC condition is preserved under conditioning.

Let the density of $\mathbf{X} = (X_1^\top, \dots, X_K^\top) \in \mathbb{R}^{dK}$ be $\mathbf{f}(\cdot)$.

For the event $E := \{\mathbf{X} \in A\}$ where $A \subset \mathbb{R}^{dK}$, the conditional density of $\mathbf{X}$ given $E$ is formulated as $\mathbf{f}_{|E}(\mathbf{x}) = \frac{\mathbf{f}(\mathbf{x})}{\mathbb{P}[E]} \in \mathbb{R}^d$ for $\mathbf{x} \in \mathbb{R}^{dK}$.

Then, the following holds:

$$\nabla \log \mathbf{f}_{|E}(\mathbf{x}) = \nabla \log \mathbf{f}(\mathbf{x}) - \nabla \mathbb{P}[E] = \nabla \log \mathbf{f}(\mathbf{x}).$$

This means $\mathbf{X} \mid E$ also satisfies the LAC condition with the same function, hence LAC is robust under conditioning.

Especially, if the event $E$ has the form $E = \{X_i \in D \text{ and } X_j = x_j \text{ for } j \neq i\}$ for some $D \subset \mathbb{R}^d$, we define the density of $X_i \mid E$ as

$$f_{i,E}(x) = \frac{\mathbf{f}(x_1, \dots, x_{i-1}, x, x_{i+1}, \dots, x_K)}{\mathbb{P}[E]} \quad \text{for } x \in \mathbb{R}^d.$$

Then,

$$\begin{aligned}
\nabla \log f_{i,E}(x) &= \nabla \log f(x_1, \dots, x_{i-1}, x, x_{i+1}, \dots, x_K) - \nabla \mathbb{P}[E] \\
&= \nabla \log f(x_1, \dots, x_{i-1}, x, x_{i+1}, \dots, x_K)
\end{aligned}$$

holds. We have

$$\|\nabla_x \log f_{i,E}(x)\|_\infty \leq \mathcal{L}(\|(x_1, \dots, x_{i-1}, x, x_{i+1}, \dots, x_K)\|_\infty)$$

which indicates that the conditional density also satisfies the LAC property with $\mathcal{L}(\cdot)$.

We summarize the above observations in the following lemma, noting that $\mathcal{L}(\cdot)$ is a non-decreasing function.

**Lemma 1 (LAC of conditional contexts)** *For a random vector $\mathbf{X} = (X_1^\top, \dots, X_K^\top) \in \mathbb{R}^{dK}$, let its density satisfy the LAC condition with function $\mathcal{L}(\cdot)$. Then, for any event $E := \{\mathbf{X} \in A\}$ where $A \subset \mathbb{R}^{dK}$ and $\|A\|_\infty \leq R$, the conditional random vector $\mathbf{X} \mid E$ satisfies the LAC condition with the constant function $\mathcal{L}(R)$. Specifically, if $E$ has the form $E = \{X_i \in D \text{ and } X_j = x_j \text{ for } j \neq i\}$ for some $D \subset \mathbb{R}^d$ with $\|D\|_\infty \leq R$, then the conditional random vector $X_i \mid E$ satisfies the LAC condition with $\mathcal{L}(R)$.*

## C.4 LAC and Truncation

We introduce the property that can compute the LAC of truncated contexts. Truncating the contexts $X_i(t)$ to a $d$-dimensional ball $\mathbb{B}_R \subset \mathbb{R}^d$ for every $i \in [K]$ still satisfies the LAC condition with a constant function, $\mathcal{L}(R)$.

**Lemma 2 (LAC with truncation)** *Suppose the density of contexts $\mathbf{X} \in \mathbb{R}^{dK}$ satisfies the LAC condition with the function $\mathcal{L}(\cdot)$. Consider the case we truncate $\mathbf{X}$ into the region $(\mathbb{B}_R)^K$ and define truncated contexts as $\overline{\mathbf{X}}$. Then, $\overline{\mathbf{X}}$ satisfies the LAC condition with constant function $\mathcal{L}(R)$.*

**Proof** The density of truncated contexts is calculated as

$$f_{\overline{\mathbf{X}}}(\mathbf{x}) = \frac{f_{\mathbf{X}}(\mathbf{x})}{\mathbb{P}[(\mathbb{B}_R)^K]}.$$

Then, when taking the log and gradient, we get

$$\nabla \log f_{\overline{\mathbf{X}}}(\mathbf{x}) = \nabla \log f_{\mathbf{X}}(\mathbf{x}).$$

Since $\|\mathbf{x}\| \leq R$ for $\mathbf{x} \in (\mathbb{B}_R)^K$, and $\mathcal{L}$ is defined by a non-decreasing function, we get

$$\nabla \log f_{\overline{\mathbf{X}}}(\mathbf{x}) \leq \mathcal{L}(R).$$

∎

## C.5 LAC and Decay Rate of Density

Next, we rigorously define the decay rate of a density.

**Definition 3 (Decay rate)** *For a density $f(x)$, $x \in \mathbb{R}^d$, and for a set $A \subset \mathbb{R}^d$, we say it has decay rate $M > 0$ if*

$$\frac{f(x_1)}{f(x_2)} \geq \exp(-M|x_1 - x_2|)$$

*for all $x_1, x_2 \in A$.*

We next present a lemma that states a density with a bounded LAC function $\mathcal{L}$ has a bounded decay rate in every direction. This property is especially useful throughout the paper.

**Lemma 3 (Decay rate and LAC)** *Suppose a density $f$ is defined on a domain $D \subset \mathbb{R}^d$ and satisfies the LAC condition with a constant function $\mathcal{L}$. Then, the decay rate of $f$ in $D$ is bounded by $\sqrt{d}\mathcal{L}$.*

**Proof** By using the Cauchy-Schwarz inequality, we can bound the directional derivative for any $v \in \mathbb{S}^{d-1}$ as

$$
\begin{aligned}
\frac{\partial_v f(x)}{f(x)} &= v^\top \frac{\nabla f(x)}{f(x)} \\
&\leq \|v\|_2 \left\| \frac{\nabla f(x)}{f(x)} \right\|_2 \\
&= \left\| \frac{\nabla f(x)}{f(x)} \right\|_2 \\
&\leq \sqrt{d} \| \frac{\nabla f(x)}{f(x)} \|_\infty \\
&\leq \sqrt{d}\mathcal{L}.
\end{aligned}
$$

Next, by applying the Gronwall inequality (Lemma 22), we obtain

$$\frac{f(x + hv)}{f(x)} \geq \exp(-\sqrt{d}\mathcal{L}h)$$

for any $x, x + hv \in D$. Since $h$ and $v$ are arbitrary, we achieve the desired result. ∎

For a univariate function, we define a one-sided decay rate, which is a weaker condition than the standard decay rate.

**Definition 4 (One-sided decay rate)** *For a univariate function $g$, we say it has a one-sided decay rate $M$ on set $A$ if for all $y, y' \in A$ with $y < y'$,*

$$\frac{g(y')}{g(y)} \geq \exp(-M(y' - y)).$$

Using these observations, we can see that the LAC condition provides an upper bound on the decay rate of the density. For example, in the one-dimensional case, a density with $f(x) \propto \exp(-Mx)$ has a decay rate $M$. We can then bound the lower bound of the variance as $\frac{1}{M^2}$ and the maximum density is bounded by $M$. Since Challenge 1 is related to the variance lower bound and Challenge 2 is related to the maximum density (of the suboptimality gap), we aim to investigate the decay rate of the contexts' density.

We first present our key lemma, which provides a relationship between the one-sided decay rate and the variance lower bound. In the following parts, we present the rigorous relationships between LAC, (one-sided) decay rate, variance lower bound, and maximum density.

**Lemma 4 (One-sided decay rate and variance lower bound)** *Consider the density $g(\cdot)$ of a random variable $Y \in \mathbb{R}$ with support $I = [a, b]$ where $b > \frac{1}{M}$ for some $M > 0$. Suppose $g$ satisfies*

$$\frac{g(y')}{g(y)} \geq \exp(-M|y - y'|)$$

*for all $y, y' \in I \cap \left[-\frac{1}{M}, \frac{1}{M}\right]$ with $y < y'$. Then, we have*

$$\mathbb{E}[Y^2] \geq c\frac{1}{M^2}$$

*for some absolute constant $c > 0$.*

**Proof** Using Lemma 5, we find that the density $g$ is bounded by $3M$ in the interval $\left[-\frac{1}{2M}, \frac{1}{2M}\right] \cap I$. By applying Lemma 6, we obtain the desired result. $\blacksquare$

Next, we present another key lemma that provides a relationship between the one-sided decay rate and the maximum density.

**Lemma 5 (One-sided decay rate leads maximum density)** *Suppose a real-valued random variable $Y$ has density $g$ and for $y_0 \in \mathbb{R}$, $[y_0, y_0 + \frac{1}{2M}] \subset \text{supp}(Y)$ for some $M > 0$. If $g$ satisfies*

$$\frac{g(y_0 + h)}{g(y_0)} \geq \exp(-Mh)$$

*for any $0 \leq h < \frac{1}{2M}$, then $g(y_0) \leq 3M$.*

**Proof** Since the integral of the density is 1, we have

$$
\begin{aligned}
1 &= \int_{y \in \text{supp}(g)} g(y)dy \\
&\geq g(y_0) \int_{y_0}^{y_0 + \frac{1}{2M}} \frac{g(y)}{g(y_0)}dy \\
&\geq g(y_0) \int_{y_0}^{y_0 + \frac{1}{2M}} \exp(-M(y - y_0))dy \\
&= g(y_0) \left[-\frac{1}{M}\exp(-M(y - y_0))\right]_{y_0}^{y_0 + \frac{1}{2M}} \\
&= g(y_0) \left(\frac{1}{M}\left(1 - \exp\left(-\frac{1}{2}\right)\right)\right) \\
&\geq g(y_0)\frac{1}{3M}.
\end{aligned}
$$

Therefore,

$$g(y_0) \leq 3M.$$

$\blacksquare$

**Lemma 6 (Maximum density leads to variance lower bound)** *Let $Y$ be a univariate random variable with support $A = \text{supp}(Y)$. For an interval $I = \left[-\frac{1}{3M}, \frac{1}{3M}\right]$, if the density of $Y$ in $I \cap A$ is bounded by $M$, then we have*

$$\mathbb{E}[Y^2] \geq c\frac{1}{M^2}.$$

*If $I \cap A = \emptyset$, above inequality still holds.*

**Proof** First, since the density is bounded by $M$,

$$\mathbb{P}[Y \in A \cap I] \leq \frac{2}{3M} \times M = \frac{2}{3}.$$

Therefore, with a probability greater than $\frac{1}{3}$, $Y \in A \cap \{|Y| > \frac{1}{3M}\}$. Thus,

$$\mathbb{E}[Y^2] \geq \frac{1}{3}\left(\frac{1}{3M}\right)^2 = \frac{1}{27M^2}.$$

If $I \cap A = \emptyset$, it means that $|Y| \geq \frac{1}{3M}$ almost surely, hence we get the wanted result. ∎

# D  High-Level Proof of Theorem 1 (Unbounded Contexts)

In this section, we briefly outline the high-level proof of Theorem 1 for unbounded contexts. This overview provides a summary and high-level sketch of the proofs, with full details presented in Appendices F and G.

We begin by explaining how overcoming the two primary challenges (discussed in Section 4) leads to a logarithmic regret bound in the proof of Theorem 1. Next, we provide a sketch of the proofs for the two key theorems: Theorem 2 for lower-bounding the diversity constant and Theorem 3 for upper-bounding the suboptimality gap.

For bounded contexts, the result statements are given in Appendix H, and the full proofs are in Appendix I. Since the proof for bounded contexts follows a similar approach to that of unbounded contexts, we start with a proof sketch for the unbounded case.

## D.1  Regret Analysis: Overcoming Two Challenges

In this section, we briefly present our proof sketch for the regret bounds. We describe how we can achieve logarithmic regret by addressing the two challenges: Challenges 1 and 2.

Addressing Challenge 1, we can obtain the $\mathcal{O}\left(\frac{1}{\sqrt{t}}\right)$ rate $\ell_2$ bound of the parameter $\hat{\theta}_t$ and $\theta^\star$ by using the combination of self-normalized concentration [1] and properties that $\Sigma_t \succeq \frac{1}{4}\lambda_\star t$ holds with high probability (by using Corollary 9). The details are in Appendix J. Hence, by tackling Challenge 1, $\|\hat{\theta}_t - \theta^\star\|_2 \leq c\frac{\sqrt{d \log T}}{\sqrt{\lambda_\star t}}$ holds with high probability. The following part describes how we can achieve logarithmic regret by addressing Challenge 2. First, simply assume the contexts are bounded, such as $\|X_i(t)\|_2 \leq x_{\max}$. Later, in our main proof, we also modify it to the relaxed condition $\|X_i(t)\|_{\psi_1} \leq x_{\max}$.

**Challenge 1: $\sqrt{t}$-rate $\ell_2$ concentration.** Ensuring the first Challenge 1 can lead to the $\ell_2$ statistical resolution of the estimator. Hence, we can get

$$|X_{a(t)}^\top(\hat{\theta}_{t-1} - \theta^\star)| \leq cx_{\max}\frac{\sqrt{d \log T}}{\sqrt{\lambda_\star \times (t-1)}},$$

$$|X_{a^\star(t)}^\top(\hat{\theta}_{t-1} - \theta^\star)| \leq cx_{\max}\frac{\sqrt{d \log T}}{\sqrt{\lambda_\star \times (t-1)}}$$

holds with high probability. Details are in Appendix J.1.

However, this resolution is insufficient for logarithmic regret; it can only achieve an $O(\sqrt{T})$ regret bound.

**Challenge 2: Towards logarithmic regret.** Furthermore, addressing Challenge 2 (margin condition), we can obtain a logarithmic expected regret upper bound. When the greedy policy selects $a(t)$, it means that

$$X_{a(t)}(t)^\top\hat{\theta}_{t-1} \geq X_{a^\star(t)}^\top\hat{\theta}_{t-1}$$

and by the definition of the optimal arm,

$$X_{a(t)}(t)^\top \theta^\star \le X_{a^\star(t)}^\top \theta^\star$$

holds. Ensuring Challenge 1, we get

$$\text{reg}'(t) := X_{a^\star(t)}(t)^\top \theta^\star - X_{a(t)}(t)^\top \theta^\star \le 2cx_{\max} \frac{\sqrt{d \log T}}{\sqrt{\lambda_\star \times (t-1)}}.$$

Next, we define the event $E$ as the event where $\text{reg}'(t) > 0$. Under the event $E$, the suboptimality gap of $\mathbf{X}(t)$ satisfies

$$\Delta(\mathbf{X}(t)) := X_{a^\star}^\top \theta^\star - \max_{j \ne a^\star} X_j^\top \theta^\star \le 2cx_{\max} \frac{\sqrt{d \log T}}{\sqrt{\lambda_\star \times (t-1)}}$$

and by overcoming Challenge 2, we get

$$\mathbb{P}_{\mathbf{X}(t)}[\text{reg}'(t) > 0] \le C_\Delta \times 2cx_{\max} \frac{\sqrt{d \log T}}{\sqrt{\lambda_\star \times (t-1)}} + \frac{1}{\sqrt{T}}$$

$$\le 3cx_{\max} C_\Delta \frac{\sqrt{d \log T}}{\sqrt{\lambda_\star \times (t-1)}}.$$

By combining these results, we can bound the expected regret as

$$\mathbf{E}_{\mathbf{X}(t)}[\text{reg}'(t)] \le 3cx_{\max} C_\Delta \frac{\sqrt{d \log T}}{\sqrt{\lambda_\star \times (t-1)}} \times 2cx_{\max} \frac{\sqrt{d \log T}}{\sqrt{\lambda_\star \times (t-1)}}$$

$$= 6c^2 x_{\max}^2 C_\Delta \frac{d \log T}{\lambda_\star \times (t-1)}.$$

This observation enables us to achieve a logarithmic regret bound. Using this argument, our only remaining goal is to bound the two constants in Challenges 1 and 2. We summarize our above observations in the following lemma.

**Lemma 7** *Assume that $\|X_i(t)\|_2 \le R$ for some $R > 0$ for all $i \in [K]$. Also assume that the estimator $\|\hat{\theta}_{t-1} - \theta^\star\| \le A\frac{1}{\sqrt{t-1}}$ holds for some constant $A > 0$. Then the (unexpected) regret in round $t$ is bounded by $\text{reg}'(t) \le 2AR\frac{1}{\sqrt{t-1}}$. Furthermore, under the margin condition (Challenge 2), we have $\mathbb{E}_{\mathbf{X}(t)}[\text{reg}'(t)] \le 6R^2 A^2 C_\Delta \frac{1}{t-1}$.*

Using the above observation, we can achieve a logarithmic regret bound when $\|X_i(t)\|_2$ is bounded. For the case where $\|X_i(t)\|_{\psi_1}$ is bounded, we use the peeling technique. Given the history $\mathcal{H}_{t-1}$, $\hat{\theta}_{t-1} - \theta^\star$ is a fixed vector, and using the results from Appendix J.2, we can bound $|X_i(t)^\top v|$, $v \in \{\hat{\theta}_{t-1} - \theta^\star, \theta^\star\}$ with high probability. Hence, we can apply similar arguments, and details are presented in Appendix J.

### D.2 Tackling Two Challenges and Fixed History Arguments

Now, we summarize our proof strategy to prove diversity (Challenge 1) and suboptimality gap (Challenge 2). First, we consider the problem with a fixed history setup, which involves analysis under the given history $\mathcal{H}_{t-1}$. We set history-conditioned contexts as $\mathbf{X} := \mathbf{X}(t) \mid \mathcal{H}_{t-1}$ and $\mathbf{X} = (X_1^\top, \ldots, X_K^\top)$, where $X_i \in \mathbb{R}^d$. We describe more about this history fixing in the next appendix F. Under the given event $\mathcal{H}_{t-1}$, $\hat{\theta}_{t-1}$ is a deterministic value and no longer random. Thus, the policy $a(t)$, conditioned on $\mathcal{H}_{t-1}$, is a greedy policy with respect to $\hat{\theta}_{t-1}$, making it deterministic as well. To address any $\hat{\theta}_{t-1}$, we propose an analysis that applies to any $\theta \in \mathbb{R}^d$. In Appendix F.1, we argue that it suffices to bound the variance of selected contexts of the greedy policy with respect to any fixed $\theta$. Since we fix $\theta$, which corresponds to the value of $\hat{\theta}_{t-1}$ under the given history $\mathcal{H}_{t-1}$, we define the policy-selected arm $a = \arg\max X_i^\top \theta$. To address Challenge 1, our first goal is to find the lower bound of $\mathbb{E}[X_a X_a^\top]$. Next, recall that we defined $\Delta(\mathbf{X}) := X_{a^\star}^\top \theta^\star - \max_{i \ne a^\star} X_i^\top \theta^\star$ and we aim to bound this suboptimality gap. Next, we introduce our two important goals to achieve the two challenges.

**Goal 1.** For any $\theta, v \in \mathbb{S}^{d-1}$ and a greedy policy with respect to $\theta$, we aim to find the lower bound of

$$\mathbb{E}[v^\top X_a X_a^\top v].$$

**Goal 2.** For a fixed $\theta^\star$, our aim is to find the upper bound of $C_\Delta$ that satisfies

$$\mathbb{P}[\Delta(\mathbf{X}) \leq \varepsilon] \leq C_\Delta \varepsilon + \frac{1}{\sqrt{T}}.$$

It is straightforward that when we achieve Goal 1, then Challenge 1 can be achieved easily since Goal 1 prepares for all greedy policies with $\theta$.

### D.3 Starting with Truncation

Before we tackle the two above goals, we start by truncating our contexts $\mathbf{X}$ to high-probability regions. Since $X_i$ has a bounded $\psi_1$ norm as $x_{\max}$, roughly speaking, there exists a set $D \subset \mathbb{R}^d$ with $\|D\|_\infty = \widetilde{\mathcal{O}}(1)$ and

$$\mathbb{P}[\mathbf{X} \in D^K] \approx 1.$$

We can view $\mathbf{X}$ as a mixture of $\mathbf{X} \mid \{X \in D^K\}$ and $\mathbf{X} \mid \{\mathbf{X} \in (D^K)^c\}$ and with high probability, $\mathbf{X}$ is sampled from the distribution $\mathbf{X} \mid \{X \in D^K\}$. In this section (Appendix D), since we are providing a proof sketch, we simply regard $\mathbf{X}$ as sampled from the distribution with bounded support $D^K$ where $\|D\|_\infty = \widetilde{\mathcal{O}}(1)$. In Appendix C.3, we observed that $\mathbf{X} \mid \{\mathbf{X} \in D^K\}$ also has LAC, and since $\|D\|_\infty = \widetilde{\mathcal{O}}(1)$, it has a constant LAC with $\mathcal{L}(\|D\|_\infty) = \mathcal{L}(\widetilde{\mathcal{O}}(1)) = \widetilde{\mathcal{O}}(1)$ since $\mathcal{L}$ is polynomial.

Therefore, for this section only, we assume that $\mathbf{X}$ has bounded support $D^K$ and **it has a bounded constant LAC function** $\mathcal{L} = \widetilde{\mathcal{O}}(1)$. For a rigorous proof and justification, we provide them throughout Appendix F and Appendix G.

### D.4 Event Decompositions

Next, we start to bound the constants in the two challenges. Before that, we present event decompositions for our analysis.

**Definition 5 (Event decomposition 1)** *We define the event* $\{a = i\}$ *as* $\Omega_i$ *and* $\Omega_i(\{x_j\}_{j \neq i}) := \{a = i\} \cap \{X_j = x_j \text{ for all } j \neq i\}$.

Pick any $v \in \mathbb{S}^{d-1}$. We can decompose the variance term as

$$\begin{aligned}
\mathbb{E}[v^\top X_a X_a^\top v] &= \mathbb{E}\left[\mathbb{E}\left[v^\top X_a X_a^\top v \mid \Omega_i\right]\right] \\
&= \mathbb{E}\left[\mathbb{E}\left[v^\top X_i X_i^\top v \mid \Omega_i\right]\right] \\
&= \mathbb{E}\left[\mathbb{E}\left[v^\top X_i X_i^\top v \mid \Omega_i(\{x_j\}_{j \neq i})\right]\right].
\end{aligned}$$

For notations $\mathbb{E}\left[\mathbb{E}\left[\cdot \mid \cdot\right]\right]$, please refer to Appendix A. From now on, we aim to bound

$$\mathbb{E}\left[v^\top X_i X_i^\top v \mid \Omega_i(\{x_j\}_{j \neq i})\right] \tag{5}$$

for any $i, \{x_j\}_{j \neq i}$. Hence, we investigate the property of the conditional density

$$X_i \mid \Omega_i(\{x_j\}_{j \neq i})$$

and especially we are interested in the density of projected conditional contexts

$$X_i^\top v \mid \Omega_i(\{x_j\}_{j \neq i}). \tag{6}$$

We define new event decompositions for bounding the suboptimality gap. Under the fixed history, set $a^\star := \arg\max_{i \in [K]} X_i^\top \theta^\star$.

**Definition 6 (Event decomposition 2)** *We define the event* $\{a^\star = i\}$ *as* $\Omega_i^\star$ *and* $\Omega_i^\star(\{x_j\}_{j\neq i}) := \{a^\star = i\} \cap \{X_j = x_j \text{ for all } j \neq i\}$.

Next, we aim to bound $C_\Delta$, which is another key constant in Challenge 2. Define the optimal arm $a^\star = \arg\max X_i^\top \theta^\star$ and the suboptimal arm $a' = \arg\max_{i\neq a^\star} X_i^\top \theta^\star$. By definition, for any $\varepsilon > 0$, we have to calculate

$$
\begin{aligned}
\mathbb{P}[\Delta(\mathbf{X}) \leq \varepsilon] &= \mathbb{P}[X_{a^\star}^\top \theta^\star - X_{a'}^\top \theta^\star \leq \varepsilon] \\
&= \mathbb{E}\left[\mathbb{P}\left[X_{a^\star}^\top \theta^\star - X_{a'}^\top \theta^\star \leq \varepsilon \mid \Omega_i^\star\right]\right] \\
&= \mathbb{E}\left[\mathbb{P}\left[X_{a^\star}^\top \theta^\star - X_{a'}^\top \theta^\star \leq \varepsilon \mid \Omega_i^\star(\{x_j\}_{j\neq i})\right]\right] \\
&= \mathbb{E}\left[\mathbb{P}\left[X_i^\top \theta^\star - \max_{j\neq i} x_j^\top \theta^\star \leq \varepsilon \mid \Omega_i^\star(\{x_j\}_{j\neq i})\right]\right].
\end{aligned}
$$

Then, we aim to bound

$$
\mathbb{P}\left[\max_{j\neq i} x_j^\top \theta^\star \leq X_i^\top \theta^\star \leq \max_{j\neq i} x_j^\top \theta^\star + \varepsilon \mid \Omega_i^\star(\{x_j\}_{j\neq i})\right]
$$

and hence we investigate the properties of the conditional density

$$
X_i^\top \theta^\star \mid \Omega_i^\star(\{x_j\}_{j\neq i}). \tag{7}
$$

It is sufficient to bound the maximum density of $X_i^\top \theta^\star \mid \Omega_i^\star(\{x_j\}_{j\neq i})$ since if it is bounded by $U$, we can see that

$$
\mathbb{P}\left[\max_{j\neq i} x_j^\top \theta^\star \leq X_i^\top \theta^\star \leq \max_{j\neq i} x_j^\top \theta^\star + \varepsilon \mid \Omega_i^\star(\{x_j\}_{j\neq i})\right] \leq U\varepsilon
$$

holds, and we can set $C_\Delta = U$.

## D.5 Conditional Contexts are Still in LAC Class

In the previous Appendix C.3 and Lemma 1, we saw that LAC is preserved under conditioning. Our conditioned contexts of interest, $X_i \mid \Omega_i(\{x_j\})$ and $X_i \mid \Omega_i^\star(\{x_j\})$, involve conditioning on events such as $X_i \mid \{X_j = x_j \text{ for all } j \neq i, X_i^\top \theta \geq \max_{j\neq i} x_j^\top \theta\}$, which meet the conditions of Lemma 1. Then, we can apply Lemma 1 and conclude that these two conditional densities

$$
X_i \mid \Omega_i(\{x_j\}_{j\neq i}), \quad X_i \mid \Omega_i^\star(\{x_j\})
$$

also **have LAC with a bounded constant function** $\mathcal{L}$.

## D.6 Remaining Goal: Bounding Decay Rate of Projected Contexts

Lemma 4 tells us that a bounded one-sided decay rate of a density guarantees a lower bound on the variance. Lemma 5 tells us that a bounded one-sided decay rate of a density guarantees a bound on the maximum density.

In summary, we need to find the lower bound of equation (6) and the maximum density of (7). We present Lemma 4 and 5, which tells us that if we can bound the one-sided decay rate, then we can bound the variance of (6) and the maximum density of (7). By the property that LAC is preserved under conditioning, we can bound the decay rate of $X_i \mid \Omega_i(\{x_j\}_{j\neq i})$ and $X_i \mid \Omega_i^\star(\{x_j\}_{j\neq i})$ by $\sqrt{d}\mathcal{L}$, by combining Lemma 3 and the LAC property of conditional contexts. However, our interests are projected contexts, defined as

$$
X_i^\top v \mid \Omega_i(\{x_j\}_{j\neq i}),
$$

and

$$
X_i^\top \theta^\star \mid \Omega_i^\star(\{x_j\}_{j\neq i}).
$$

These are *projected contexts* with some direction $v$ and $\theta^\star$. In the remaining part, we aim to bound the **one-sided decay rate** of these projected random variables' densities to apply Lemma 4 and Lemma 5.

**Support of conditional densities.** We first observe the support of the conditional density $X_i \mid \Omega_i(\{x_j\}_{j \neq i})$. By the definition of $\Omega_i$, the arm $i$ should be optimal under the greedy policy $\theta$ in this event. Therefore, its support is restricted to $\{x \in D \mid x^\top \theta \geq \max_{j \neq i} x_j^\top \theta\}$.

Next, we consider the density of projected contexts, $X_i^\top v \mid \Omega_i(\{x_j\}_{j \neq i})$. Since the support of $X_i \mid \Omega_i(\{x_j\}_{j \neq i})$ is of the form $\{x \in D \mid x^\top \theta \geq \max_{j \neq i} x_j^\top \theta\}$, our projected density is an integrated form:

$$\mathbb{P}\left[X_i^\top v = y \mid \Omega_i(\{x_j\}_{j \neq i})\right] = \int_{\{x \in D \mid x^\top \theta \geq \max_{j \neq i} x_j^\top \theta\} \cap \{x \mid x^\top v = y\}} \mathbb{P}\left[X_i = x \mid \Omega_i(\{x_j\}_{j \neq i})\right] dx.$$

Geometrically, this represents the total density at the intersection of the hyperplane $\{x \mid x^\top v = y\}$ and the set $\{x \in D \mid x^\top \theta \geq \max_{j \neq i} x_j^\top \theta\}$. We refer to these as section densities, as they represent the total density of sections sliced by the hyperplane $\{x \mid x^\top v = y\}$. Further discussion is provided in Appendix E.

**Conclusion.** Later in Appendix F and Appendix G, since $X_i \mid \Omega_i(\{x_j\}_{j \neq i})$ has a bounded decay rate $\sqrt{d}\mathcal{L} = \widetilde{\mathcal{O}}(\sqrt{d})$, we prove that the projected densities also have a bounded one-sided decay rate of $\widetilde{\mathcal{O}}(\sqrt{d})$. Lastly, by applying Lemma 4, we can prove the quantity

$$\mathbb{E}\left[v^\top X_i X_i^\top v \mid \Omega_i(\{x_j\}_{j \neq i})\right] \geq c\frac{1}{d}$$

for some $c = \widetilde{\mathcal{O}}(1)$. Also, using Lemma 5, we can prove that the density of (7) has a bounded density $\sqrt{d}\mathcal{L}$ and we can prove that $C_\Delta = \widetilde{\mathcal{O}}(\sqrt{d})$.

To summarize, we proceed with the following steps:

1. Event decomposition by conditioning.
2. We know $X_i \mid \Omega_i(\{x_j\}_{j \neq i})$ and $X_i \mid \Omega_i^\star(\{x_j\}_{j \neq i})$ have a bounded decay rate $\sqrt{d}\mathcal{L} = \widetilde{\mathcal{O}}(\sqrt{d})$.
3. We aim to bound the decay rate of the projected densities $X_i^\top v \mid \Omega_i(\{x_j\}_{j \neq i})$ and $X_i^\top \theta^\star \mid \Omega_i^\star(\{x_j\}_{j \neq i})$.
4. If we are able to bound the decay rate of the above projected densities, we can bound the two key constants in the two challenges using Lemma 4 and Lemma 5.

For step 3, the remaining important task is to bound the decay rate of the projected densities, which we address in Appendix E, Appendix F, and Appendix G.

## E   Sections, Section Densities and Decay Rate

In this section, we define sections and section densities and investigate their properties. Previously in Appendix D, we discussed decomposition by conditioning, and our interest became the properties of conditioned contexts, the form of $X_i \mid \Omega_i(\{x_j\}_{j \neq i})$ and $X_i \mid \Omega_i^\star(\{x_j\}_{j \neq i})$. Our first goal is to bound the variance of projected contexts, such as $X_i^\top v \mid \Omega_i(\{x_j\}_{j \neq i})$ for all $v \in \mathbb{R}^d$. To do that, we investigate the projected density of $X_i^\top v = y \mid \Omega_i(\{x_j\}_{j \neq i})$. To apply Lemma 4 and Lemma 5, we only need to bound the one-sided decay rate of that projected density. We saw that its density is the total density of intersections with $\{x \mid x^\top \theta \geq \max_{j \neq i} x_j^\top \theta\}$ and hyperplane $\{x \mid x^\top v = y\}$. We can view it as sections sliced by hyperplanes, and we provide some theory related to sections and section densities.

### E.1   Motivation

In the proof strategy (Appendix D), we aim to bound the one-sided decay rate and maximum density of projected conditional density, $X_i^\top v \mid \Omega_i(\{x_j\}_{j \neq i})$ and $X_i^\top \theta^\star \mid \Omega_i^\star(\{x_j\}_{j \neq i})$. Recall that we define one-sided decay rate of univariate density $g$ (Definition 4) at Appendix D as a constant $M$ satisfying

$$\frac{g(y')}{g(y)} \geq \exp(-M(y' - y))$$

for all $y < y'$.

Then, how to bound the (one-sided) decay rate of section density? We present a technique to bound the (one-sided) decay rate of section densities defined in equal and expanding sections.

## E.2 Sections and Section Densities

Our remaining goal is to bound the one-sided decay rate of section density. To build some general theory, we first define sections and section densities explicitly.

**Definition 7 (Sections)** *For the set $A \subset \mathbb{R}^d$ and $v \in \mathbb{S}^{d-1}$, we define sections as*

$$\mathrm{Sec}(A, v, y) := A \cap \{x \mid x^\top v = y\}$$

*for $y \in \mathbb{R}$.*

This means that a section is the intersection of region $A$ and hyperplane $\{x \in \mathbb{R}^d \mid x^\top v = y\}$.

**Definition 8 (Section density)** *We define the section density of the region $A$. Let $f$ be the density defined in the region $A \subset \mathbb{R}^d$. Then we define $g(y)$ as the section density of $\mathrm{Sec}(A, v, y)$ as*

$$g(y) = \int_{x \in \mathrm{Sec}(A,v,y)} f(x) \, dx.$$

**Remark 1** *If the density of $X \in \mathbb{R}^d$ is $f$, the section density corresponds to the density of $X^\top v$ for $v \in \mathbb{S}^{d-1}$.*

Next, we define the areas with special section structures: *equal sections* and *expanding sections*. First, we define equal sections, the area whose section with direction $v$ is equal.

**Definition 9 (Equal sections)** *We define the set $A$ to have equal sections with $v \in \mathbb{S}^{d-1}$ when $\mathrm{Sec}(A, v, y)$ is congruent in shape for $y > 0$.*

Next, we define expanding sections, where the sections of $A$ with direction $v$ expand when $y$ increases.

**Definition 10 (Expanding sections)** *We define the sections of $A$ with direction $v$ as expanding sections when*

$$\mathrm{Sec}(A, v, y) + hv \subset \mathrm{Sec}(A, v, y + h)$$

*for every $h > 0$. Technically, equal sections are included in expanding sections.*

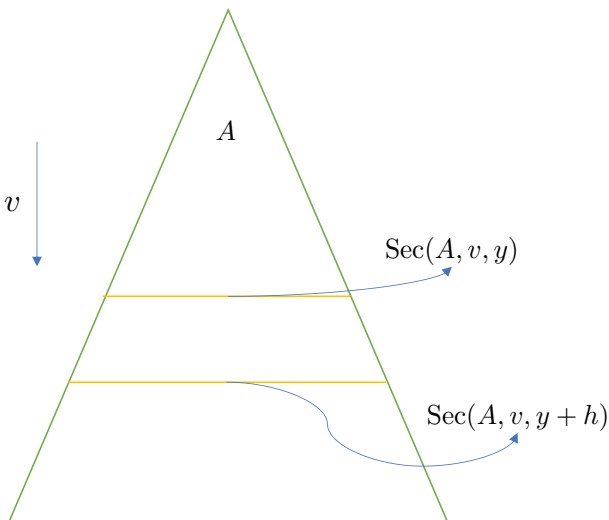

Figure 2: Illustration of expanding section's example. The section with direction $v$ is expanding when $y$ increases: $y$ to $y + h$.

## E.3 Decay Rate of Section Density

Recall that we define the one-sided decay rate of univariate density $g$ (Definition 4) at Appendix D as a constant $M$ satisfying

$$\frac{g(y')}{g(y)} \geq \exp(-M(y' - y))$$

for all $y < y'$.

Then, how to bound the (one-sided) decay rate of section density? We present a technique to bound the (one-sided) decay rate of section densities defined in equal and expanding sections.

**Lemma 8 (One-sided decay rate: equal and expanding sections)** *For $A \subset \mathbb{R}^d$, $v \in \mathbb{R}^d$, and $y \in \mathbb{R}$, a density $f$ has support $A$ and has a bounded decay rate $M$. Assume that the sections of $A$ with direction $v$ are expanding sections. We define the section density of $\mathrm{Sec}(A, v, y)$ as $g(y)$, then we have for any $y_1 < y_2$:*

$$\frac{g(y_1)}{g(y_2)} \geq \exp(-M|y_1 - y_2|).$$

**Proof** Since the section is expanding with the direction $v$, we can observe for any $h > 0$,

$$
\begin{aligned}
\frac{g(y + h)}{g(y)} &= \frac{\int_{\mathrm{Sec}(A,v,y+h)} f(x) \, dx}{\int_{\mathrm{Sec}(A,v,y)} f(x) \, dx} \\
&\geq \frac{\int_{\mathrm{Sec}(A,v,y)} f(x) \exp(-Mh) \, dx}{h \int_{\mathrm{Sec}(A,v,y)} f(x) \, dx} \\
&\geq \exp(-Mh).
\end{aligned}
$$

The second inequality holds since the section is expanding and using Lemma 3. Since equal sections are also expanding sections, we end the proof. ∎

**Lemma 9 (Maximum density: equal and expanding sections)** *The density $f(\cdot)$ has support $A \subset \mathbb{R}^d$ and has a bounded decay rate $M$. Furthermore, sections with direction $v$ are expanding sections and define the section density of $\mathrm{Sec}(A, v, y)$ as $g(y)$. If the support of $g(\cdot)$ contains an interval $[a, b]$, then $g(y) \leq 3M$ for $y \in [a, b - \frac{1}{2M}]$.*

**Proof** This can be obtained directly by applying the result of the previous Lemma 8 and Lemma 5. ∎

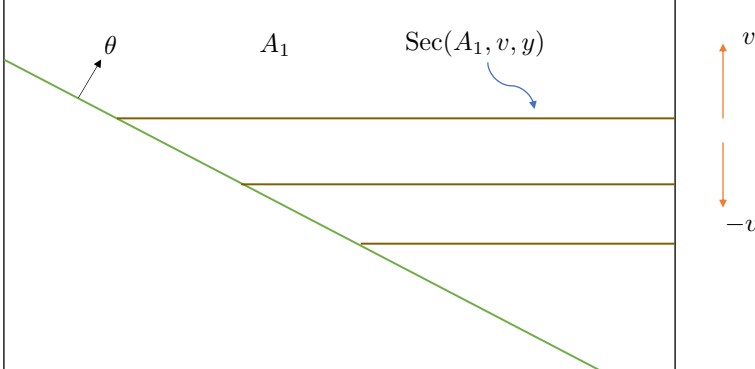

Figure 3: Illustration of $A_1$ and expanding sections of $v$ or $-v$. $A_1$ is the area above the green line. In this case, sections with direction $+v$ are expanding! If a cylindrical set is cut by some hyperplane (which is $A_1$), **at least one direction** makes expanding sections.

# F  Two Key Propositions in Fixed Histories (Unbounded Contexts)

This section aims to provide key propositions to prove Theorem 2 and Theorem 3. Our two challenges aim to bound

$$\mathbb{E}[X_{a(t)}(t)X_{a(t)}(t)]$$

and

$$\mathbb{P}[\Delta(\mathbf{X}(t)) \leq \varepsilon]. \tag{8}$$

We define $\mathbf{X}(t) = (X_1(t)^\top, \ldots, X_K(t)^\top) \in \mathbb{R}^{dK}$. Under the given event $\mathcal{H}_{t-1}$, $\hat{\theta}_{t-1}$ is a deterministic value and is no longer random.

## F.1  Details for Diversity Constant (Challenge 1)

In the definition of the diversity constant, we recall that the expectation is taken with respect to $\mathbf{X}(t)$ and the history $\mathcal{H}_{t-1}$. In round $t$, for any fixed history $\mathcal{H}_{t-1}$, we perform the greedy policy with the estimator $\hat{\theta}_{t-1}$. Hence, when the entire set of contexts $\mathbf{X}(t) = (X_1(t)^\top, \ldots, X_K(t)^\top)$ is revealed, $a(t)$ is determined immediately, given the history $\mathcal{H}_{t-1}$. Then the exact statement is:

"In round $t$, for any given history $\mathcal{H}_{t-1}$,

$$\mathbb{E}_{\mathbf{X}(t)}[X_{a(t)}(t)X_{a(t)}(t)^\top] \succeq \lambda_\star I_d$$

holds with some $\lambda_\star > 0$."

In the proof, we proved a stronger statement:

"For any greedy policy with any $\theta \in \mathbb{R}^d$ and selected arm $a(t)$ with that policy,

$$\mathbb{E}_{\mathbf{X}(t)}\left[X_{a(t)}(t)X_{a(t)}(t)^\top\right] \succeq \lambda_\star I_d$$

holds with some $\lambda_\star > 0$."

We prove a stronger quantity (second argument). For any given parameter $\theta \in \mathbb{R}^d$, define $a_\theta(\mathbf{X}(t)) := \operatorname{argmax}_{i \in [K]} X_i(t)^\top \theta$ and define

$$\lambda_\star(t) := \min_{\|\theta\|_2 = 1} \lambda_{\min}\left(\mathbb{E}_t[X_{a_\theta(\mathbf{X}(t))}X_{a_\theta(\mathbf{X}(t))}^\top]\right).$$

Since $\hat{\theta}_{t-1}$ can be an arbitrary value, we prepare for all greedy policies w.r.t. $\theta \in \mathbb{S}^{d-1}$. Then the newly defined $\lambda_\star(t)$ satisfies equation (1), and we discuss this in Appendix F.

Thus, the policy $a(t)$, conditioned on $\mathcal{H}_{t-1}$, is a greedy policy with respect to $\hat{\theta}_{t-1}$, making it deterministic as well. To address any $\hat{\theta}_{t-1}$, we propose an analysis that applies to any $\theta \in \mathbb{R}^d$, aiming to bound the variance of the selected contexts. In Appendix F.1, we argue that it suffices to bound this variance for any fixed $\theta$.

$$\min_{\|\theta\|_2 = 1} \lambda_{\min}\left(\mathbb{E}[X_{a_\theta(\mathbf{X}(t))}(t)X_{a_\theta(\mathbf{X}(t))}(t)^\top]\right)$$

Note that $a_\theta(\mathbf{X}(t))$ is defined as $\arg\max_{i \in [K]} X_i(t)^\top \theta$. This quantity is equal to

$$\min_{\|\theta\|_2 = 1, \|v\|_2 = 1} \mathbb{E}_t\left[|v^\top X_{a_\theta(\mathbf{X}(t))}(t)|^2\right]$$

and we prove this lower bound for any $\theta$ and $v$. In summary, our aim is to prove that for any history $\mathcal{H}_{t-1}$ and $\theta, v$, we can bound

$$\mathbb{E}\left[|v^\top X_{a_\theta(\mathbf{X}(t))}(t)|^2\right]. \tag{9}$$

To bound the margin constant, our aim is to prepare for all $\theta^\star$, since the true model parameter $\theta^\star$ can also be arbitrary. For simplicity, when contexts are clear, we write $a_\theta(\mathbf{X}) = a$ (when we fix $\theta$).

### F.2 The Setup of Fixed History Analysis

We aim to find a class of densities $\mathcal{F}$ such that if $\mathbf{X}(t) \in \mathcal{F}$, then equation (9) is lower bounded for any $\theta, v$. Similarly, we aim to find a class of densities $\mathcal{G}$, such that if $\mathbf{X}(t) \in \mathcal{G}$, then equation (8) can be upper bounded for any $\theta^\star \in \mathbb{R}^d$.

1. We first prove that for a fixed $\theta, v$, for densities with LAC and supports satisfying certain geometric conditions, we can bound equation (9).

2. We also prove that for a fixed $\theta^\star$, for densities with LAC and supports satisfying certain geometric conditions, we can bound equation (8).

3. Next, we prove that the densities with LAC and bounded $\psi_1$ norm, for any given $\theta, v$, can be truncated (with high probability) to densities contained in class 1.

4. Lastly, we prove that for densities with LAC and bounded $\psi_1$ norm, for any given $\theta^\star$, we can truncate it (with high probability) to densities contained in class 2.

In this section, we provide the results of 1 and 2. In the next Appendix G, we proceed to 3 and 4.

Now, we build the theory with the assumptions that $\theta, v$ and $\mathcal{H}_{t-1}$ are fixed. We assume that random vectors $\mathbf{Z} = (Z_1^\top \ldots Z_K^\top)$, where $Z_i \in \mathbb{R}^d$, arise from some distribution with LAC $\mathcal{L}(\cdot)$. We fix an arbitrary $\theta \in \mathbb{R}^d$ and define the random index variable

$$a = \underset{i}{\operatorname{argmin}}\, Z_i^\top \theta.$$

We also fix $v \in \mathbb{S}^{d-1}$ and first aim to bound $\mathbb{E}[v^\top Z_a Z_a^\top v]$. Our goal is to find the lower bound of $\mathbb{E}[v^\top Z_a Z_a^\top v]$ for any $v \in \mathbb{S}^{d-1}$; from now on, we fix an arbitrary $v \in \mathbb{S}^{d-1}$. We then aim to bound

$$\mathbb{E}[v^\top Z_a Z_a^\top v]$$

for fixed $\theta$ and $v$. Proposition 2 below shows that when the support of $\mathbf{Z}$ satisfies certain geometric conditions, $\mathbb{E}[v^\top Z_a Z_a^\top v]$ can be effectively bounded.

We next define $C_\Delta$ as the margin constant of $\mathbf{Z}$ with the parameter $\theta^\star$, which satisfies

$$\mathbb{P}[\Delta(\mathbf{Z}) \leq \varepsilon] \leq C_\Delta \varepsilon + \frac{1}{\sqrt{T}}.$$

where $\Delta(\mathbf{Z})$ is a suboptimality gap, defined similarly in Section 4. Proposition 3 below shows that when the support of $\mathbf{Z}$ satisfies certain geometric conditions, the margin constant of $\mathbf{Z}$ can be effectively bounded.

Same as previous definitions, we define the event

$$\Omega_i := \{a = i\}$$

and

$$\Omega_i(\{z_j\}_{j \neq i}) := \{a = i\} \cap \{Z_j = z_j \text{ for all } j \neq i\}.$$

Simiarly, we define

$$\Omega_i^\star := \{a^\star = i\}$$

and

$$\Omega_i^\star(\{z_j\}_{j \neq i}) := \{a^\star = i\} \cap \{Z_j = z_j \text{ for all } j \neq i\}.$$

### F.3 Key Proposition for Diversity Constant

Recall that we define $\left[-1, 1\right]_v := \{x \mid -1 \leq x^\top v \leq 1\}$ for any $v \in \mathbb{R}^d$.

**Proposition 2 (Key proposition for diversity constant)** *The random vector* $\mathbf{Z} = (Z_1^\top \ldots Z_K^\top)$, $Z_i \in \mathbb{R}^d$, *satisfies the following conditions:*

1. $\mathbf{Z}$ *has LAC with* $\mathcal{L}(\cdot)$ *and* $\|\operatorname{supp}(\mathbf{Z})\|_\infty \leq R$.

2. *For all* $i \in [K]$, $\operatorname{supp}(Z_i)$ *is identical for some subset* $A \subset \mathbb{R}^d$. *That is,* $\operatorname{supp}(\mathbf{Z}) = A^K$ *for some* $A \subset \mathbb{R}^d$.

3. *The support $A \cap [-1, 1]_v$ has equal sections with direction $v$.*

*Then there exists an absolute constant $c > 0$ such that*

$$\mathbb{E}[v^\top Z_a Z_a^\top v] \geq \frac{c}{d\mathcal{L}(R)^2}.$$

**Remark 2** *For unbounded contexts $\mathbf{X}(t)$ in the original bandit problem, we truncate to some region $\mathbf{C}_1$ with positive probability and make the truncated contexts satisfy the condition of the above proposition.*

### F.3.1 Proof of Proposition 2

Since $\operatorname{supp}(\mathbf{Z}) \leq R$, it has bounded decay rate $\sqrt{d}\mathcal{L}(R)$ by Lemma 3. By the established event decomposition, we see

$$\Omega_i = \bigcup_{\{z_j\}_{j \neq i}} \Omega_i(\{z_j\}_{j \neq i}),$$

and by applying the tower property, we get

$$\mathbb{E}[v^\top Z_a Z_a^\top v] = \mathbb{E}\big[\mathbb{E}[Z_a Z_a^\top \mid \Omega_i(\{z_j\}_{j \neq i})]\big]$$
$$= \mathbb{E}\big[\mathbb{E}[Z_i Z_i^\top \mid \Omega_i(\{z_j\}_{j \neq i})]\big].$$

Hence, we only need to bound

$$\mathbb{E}[v^\top Z_i Z_i^\top v \mid \Omega_i(\{z_j\}_{j \neq i})]. \tag{10}$$

for every $\Omega_i(\{z_j\}_{j \neq i})$. Recall that we defined $\big[-1, 1\big]_v := \{x \mid x^\top v \in [-1, 1]\}$. Then we decompose it as

$$\mathbb{E}[v^\top Z_i Z_i^\top v \mid \Omega_i(\{z_j\}_{j \neq i})]$$
$$= \mathbb{E}\big[v^\top Z_i Z_i^\top v \mid \Omega_i(\{z_j\}_{j \neq i}) \cap \{Z_i \in [-1, 1]_v\}\big] \, \mathbb{P}\big[Z_i \in [-1, 1]_v \mid \Omega_i(\{z_j\}_{j \neq i})\big]$$
$$+ \mathbb{E}\big[v^\top Z_i Z_i^\top v \mid \Omega_i(\{z_j\}_{j \neq i}) \cap \{Z_i \in ([-1, 1]_v)^c\}\big] \, \mathbb{P}\big[Z_i \in ([-1, 1]_v)^c \mid \Omega_i(\{z_j\}_{j \neq i})\big]$$
$$\geq \mathbb{E}\big[v^\top Z_i Z_i^\top v \mid \Omega_i(\{z_j\}_{j \neq i}) \cap \{Z_i \in [-1, 1]_v\}\big].$$

The last inequality holds because $\mathbb{E}\big[v^\top Z_i Z_i^\top v \mid \Omega_i(\{z_j\}_{j \neq i}) \cap \{Z_i \in ([-1, 1]_v)^c\}\big] \geq 1$ and $\mathbb{E}\big[v^\top Z_i Z_i^\top v \mid \Omega_i(\{z_j\}_{j \neq i}) \cap \{Z_i \in [-1, 1]_v\}\big] \leq 1$.

From now on, we focus on the conditional density of $Z_i \mid \Omega_i(\{z_j\}_{j \neq i}) \cap \{Z_i \in [-1, 1]_v\}$ and its projected density $v^\top Z_i \mid \Omega_i(\{z_j\}_{j \neq i}) \cap \{Z_i \in [-1, 1]_v\}$. For simplicity, we set the conditional density of $Z_i \mid \Omega_i(\{z_j\}_{j \neq i}) \cap \{Z_i \in [-1, 1]_v\}$ as $f_1(\cdot)$ and the projected density of $v^\top Z_i \mid \Omega_i(\{z_j\}_{j \neq i}) \cap \{Z_i \in [-1, 1]_v\}$ as $g_1(\cdot)$. Using Lemma 1, the density $f_1(\cdot)$ has LAC with constant function $\mathcal{L}(R)$, and it has bounded decay rate $\sqrt{d}\mathcal{L}(R)$ by Lemma 3.

**[1] Investigating the support of $f_1$ and $g_1$.** First, we examine the support of $f_1$. Since arm $i$ is optimal with estimator $\theta$, all $z$ in the support of $f_1$ must satisfy $z^\top \theta \geq \max_{j \neq i} z_j^\top \theta$. Also, because $-1 \leq z^\top v \leq 1$ holds, this support is the intersection of these two areas. Define

$$A_1 := \operatorname{supp}(Z_i \mid \Omega_i(\{z_j\}_{j \neq i}) \cap \{Z_i \in [-1, 1]_v\}) = \{z \in A \mid z^\top \theta \geq \max_{j \neq i} z_j^\top \theta, \, -1 \leq z^\top v \leq 1\}.$$

Recall that $A := \operatorname{supp}(Z_i)$ is designed to have equal sections with direction $v$. Hence $A \cap [-1, 1]_v$ has equal sections with $v$. Therefore, the support of $Z_i^\top v \mid \Omega_i(\{z_j\}_{j \neq i}) \cap \{Z_i \in [-1, 1]_v\}$ is an interval by our design of $A$.

**[2] Bounding one-side decay rate of section density in $[-1, 1]$.** We aim to apply Lemma 4 to bound the variance $\mathbb{E}\big[v^\top Z_i Z_i^\top v \mid \Omega_i(\{z_j\}_{j \neq i}) \cap \{Z_i \in [-1, 1]_v\}\big]$. It tells us that it is enough to bound the one-side decay rate of the section density $g_1(\cdot)$ in the interval $[-1/2, 1/2]$.

**Claim 1** *One of $\operatorname{Sec}(A_1, v, y)$ or $\operatorname{Sec}(A_1, -v, y)$ consists of expanding sections when $y$ increases for $y \in [-1, 1]$.*

**Proof of Claim 1**  Choose one of $v, -v$ such that $\langle \cdot, \theta \rangle \geq 0$. We want to use the result of Lemma 10. Since $[-1, 1]_v$ has equal sections with direction $v$, by rotating the axis, we can satisfy the condition of Lemma 10. Then Lemma 10 implies that at least one direction among $v$ or $-v$ yields expanding sections. See Figure 5 for intuition. ∎

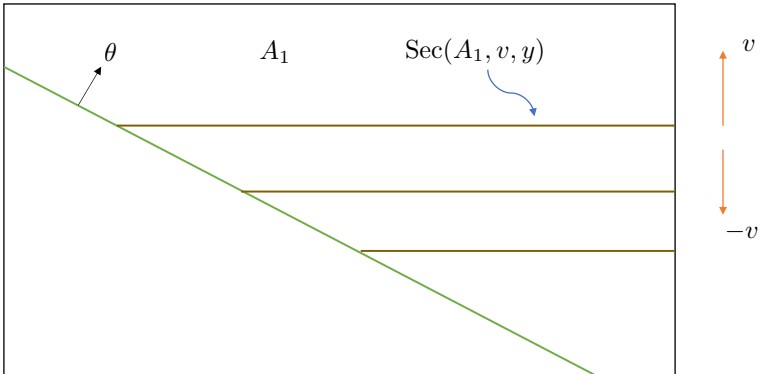

Figure 4: Illustration of $A_1$ and expanding sections of $v$ or $-v$. $A_1$ is the area above the green line. In this case, sections with direction $+v$ are expanding. If a cylindrical set is cut by some hyperplane, **at least one direction** produces expanding sections.

Claim 1 tells us that we can apply Lemma 8, and thus bound the one-side decay rate of the section density $g_1(\cdot)$. Without loss of generality, assume $\mathrm{Sec}(A_1, v, y)$ is expanding when $y$ increases. Then the support of $Z_i^\top v \mid \Omega_i(\{z_j\}_{j \neq i}) \cap \{Z_i \in [-1, 1]_v\}$ is an interval of the form $[c, 1]$ with some $-1 < c < 1$.

**Claim 2** *The one-side decay rate of $\mathbb{P}[v^\top Z_i = y \mid \Omega_i(\{z_j\}_{j \neq i}) \cap [-1, 1]_v]$ is bounded by $\sqrt{d}\mathcal{L}(R)$ for $y \in [-1, 1]$.*

**Proof of Claim 2**  To bound the one-side decay rate, we apply Lemma 8. By Claim 1, without loss of generality, sections with direction $v$, i.e., $\mathrm{Sec}(A, v, y)$, form an expanding section. By Lemma 8, we obtain the desired result. ∎

**[3] Bounding the desired variance.**  We now see that the support of $g_1(\cdot)$ has the form $[c, 1]$ for some $c < 1$. Hence, by Lemma 4, we conclude

$$\mathbb{E}[v^\top Z_i Z_i^\top v \mid \Omega_i(\{z_j\}_{j \neq i}) \cap \{Z_i \in [-1, 1]_v\}] \geq \frac{c}{d\mathcal{L}(R)^2}$$

for some absolute constant $c > 0$. ∎

Below, we prove Lemma 10, which is used in the proof.

**Lemma 10** *Consider the (cylindrical-shaped) set $S = U \times I$ for $U \subset \mathbb{R}^{d-1}$ and an interval $I \subset \mathbb{R}$. For $\theta \in \mathbb{R}^d$ with $\theta^\top e_n \geq 0$ and any $b \in \mathbb{R}$, define the set $S' := S \cap \{x \mid x^\top \theta \geq b\}$. Then $S'$ yields expanding sections with direction $e_n$.*

**Proof**  To prove that they are expanding sections, we need to show that for any $h > 0$, $x_0 + he_n \in \mathrm{Sec}(S', e_n, y + h)$ for any $x_0 \in \mathrm{Sec}(S', e_n, y)$. By the definition of sections, clearly $x_0 + he_n \in \{x \in \mathbb{R}^d \mid x^\top e_n = y + h\}$. Next, we must show $x_0 + he_n \in \{x \mid x^\top \theta \geq b\}$. Since $e_n^\top \theta \geq 0$,

$$(x_0 + he_n)^\top \theta \geq x_0^\top \theta \geq b$$

holds. ∎

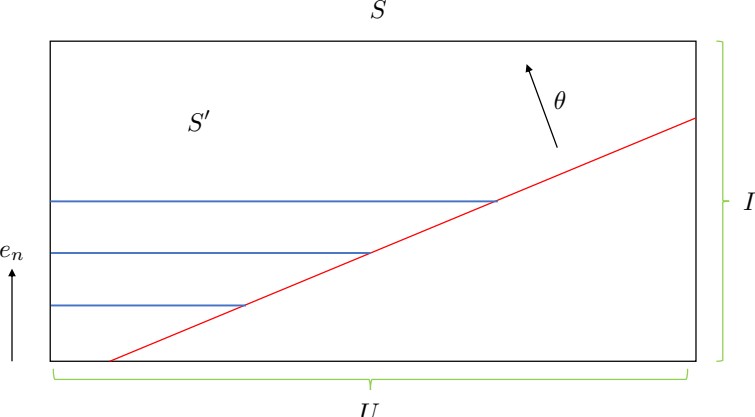

Figure 5: Illustration of $S$ and expanding sections with direction $e_n$. Blue lines are sections $\mathrm{Sec}(S', e_n, y)$. If a cylindrical-shaped set is sliced by some hyperplane, **at least one direction** forms expanding sections.

### F.4 Key Proposition for Suboptimality Gap

Next, we provide a fixed-history analysis to bound the margin constant of Challenge 2.

Before we start, we define the cylindrical set used to characterize the support of densities.

**Definition 11 (Cylindrical region)** *We define $A \subset \mathbb{R}^d$ as a cylindrical region with direction $v \in \mathbb{S}^{d-1}$ and length $2H$ if*

$$A = \{B + tv \mid -H \leq t \leq H\}$$

*for some subset $B \subset \{x \mid x^\top v = 0\}$. We denote this set by $\mathrm{cyl}(B, v, H)$.*

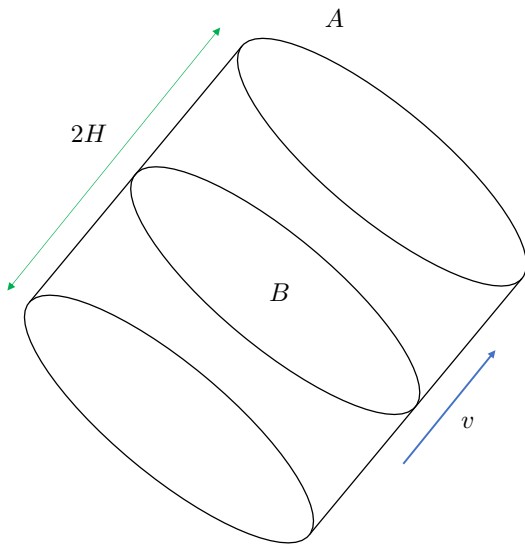

Figure 6: Illustration of a cylindrical region $A$.

Next, we show that if the support of each context $Z_i$ is a cylindrical set, we can bound the margin constant. We state our key result on bounding the margin constant in the fixed-history setup.

**Proposition 3 (Key proposition: suboptimality gap)** *Assume that the random vector $\mathbf{Z} = (Z_1^\top \ldots Z_K^\top)$, $Z_i \in \mathbb{R}^d$, satisfies LAC with function $\mathcal{L}(\cdot)$ and $\|\mathrm{supp}(\mathbf{Z})\|_\infty \leq R$. Additionally, suppose it satisfies the following conditions:*

1. *For all $i \in [K]$, $\mathrm{supp}(Z_i) \subset \mathbb{R}^d$ is identical for some set $A \subset \mathbb{R}^d$. That is, $\mathrm{supp}(\mathbf{Z}) = A^K$.*

2. *For some $H > 0$, $Z_i$'s support $\mathrm{supp}(Z_i) = A = \mathrm{cyl}(B, \theta^\star, H + 1)$ for all $i \in [K]$, and $\mathbb{P}[\mathbf{Z} \in (\mathrm{cyl}(B, \theta^\star, H))^K] \geq 1 - \delta$.*

*Then*

$$\mathbb{P}[\Delta(\mathbf{Z}) \leq \varepsilon] \leq 3\sqrt{d}\mathcal{L}(R)\varepsilon + \delta.$$

**Remark 3** *For the original bandit problem with contexts $\mathbf{X}(t)$, we truncate to a high-probability region $\mathbf{C}_2$ and set the truncated contexts to satisfy the conditions of the above proposition. We provide the analysis in the next Appendix G.*

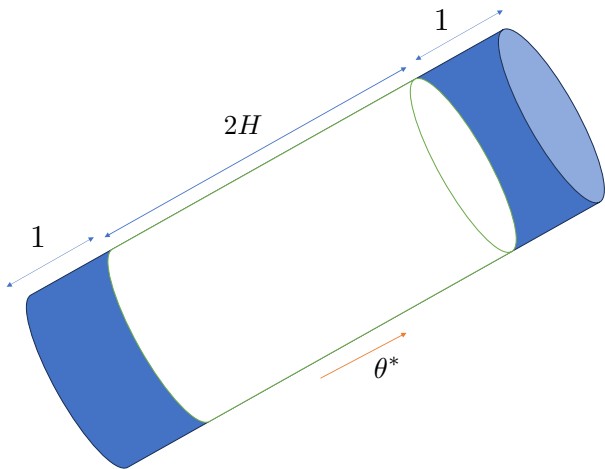

Figure 7: Illustration of $\mathrm{supp}(Z_i)$. It has equal sections with $\theta^\star$ and $\|Z_i\|_\infty \leq R$.

### F.4.1   Proof of Proposition 3

Define $Z_i^\top \theta^\star = U_i$ for all $i \in [K]$. With a slight abuse of notation, define the optimal arm $a^\star = \arg\max_i Z_i^\top \theta^\star$ and the suboptimal arm $a^\dagger = \arg\max_{i \neq a^\star} Z_i^\top \theta^\star$.

**[1] Decomposition by conditioning.**   We can bound the margin probability by conditioning as

$$
\begin{aligned}
&\mathbb{P}[U_{a^\star} - U_{a^\dagger} \leq \varepsilon]\\
&= \mathbb{E}[\mathbf{1}(U_{a^\star} - U_{a^\dagger} \leq \varepsilon)]\\
&= \mathbb{E}[\mathbf{1}(U_{a^\star} - U_{a^\dagger} \leq \varepsilon) \mid U_{a^\dagger} \leq H]\mathbb{P}[U_{a^\dagger} \leq H] + \mathbb{E}[\mathbf{1}(U_{a^\star} - U_{a^\dagger} \leq \varepsilon) \mid U_{a^\dagger} > H]\mathbb{P}[U_{a^\dagger} > H]\\
&\leq \mathbb{E}[\mathbf{1}(U_{a^\star} - U_{a^\dagger} \leq \varepsilon) \mid U_{a^\dagger} \leq H]\mathbb{P}[U_{a^\dagger} \leq H] + \delta \quad \text{(By the second condition of the proposition)}\\
&\leq \mathbb{E}[\mathbf{1}(U_{a^\star} - U_{a^\dagger} \leq \varepsilon) \mid U_{a^\dagger} \leq H] + \delta\\
&\leq \mathbb{E}\big[\mathbb{E}[\mathbf{1}(U_{a^\star} - U_{a^\dagger} \leq \varepsilon) \mid \{U_{a^\dagger} \leq H\} \cap \Omega_i^\star(\{z_j\}_{j \neq i})]\big] + \delta\\
&\leq \mathbb{E}\big[\mathbb{E}[\mathbf{1}(0 \leq U_i - \max_{j \neq i} z_j^\top \theta^\star \leq \varepsilon) \mid \{\max_{j \neq i} z_j^\top \theta^\star \leq H\} \cap \Omega_i^\star(\{z_j\}_{j \neq i})]\big] + \delta\\
&= \mathbb{E}\big[\mathbb{E}[\mathbf{1}(\max_{j \neq i} z_j^\top \theta^\star \leq Z_i^\top \theta^\star \leq \varepsilon + \max_{j \neq i} z_j^\top \theta^\star) \mid \{\max_{j \neq i} z_j^\top \theta^\star \leq H\} \cap \Omega_i^\star(\{z_j\}_{j \neq i})]\big] + \delta.
\end{aligned}
$$

We set the density of $Z_i \mid \{\max_{j \neq i} z_j^\top \theta^\star \leq H\} \cap \Omega_i^\star(\{z_j\}_{j \neq i})$ and $Z_i^\top \theta^\star \mid \{\max_{j \neq i} z_j^\top \theta^\star \leq H\} \cap \Omega_i^\star(\{z_j\}_{j \neq i})$ as $f_2$ and $g_2$.

Our goal is to bound the maximum density of $g_2$, which is a density of

$$Z_i^\top \theta^\star \mid \{\max_{j \neq i} z_j^\top \theta^\star \leq H\} \cap \Omega_i^\star(\{z_j\}_{j \neq i}).$$

**[2] Support of $f_2(\cdot)$, $g_2(\cdot)$ and their geometry.** We examine the support of the conditional density $f_2$, which is

$$Z_i = z \mid \{\max_{j \neq i} z_j^\top \theta^\star \leq H\} \cap \Omega_i^\star(\{z_j\}_{j \neq i}).$$

Under the event $\Omega_i^\star(\{z_j\}_{j \neq i})$, we have $z^\top \theta^\star \geq \max_{j \neq i} z_j^\top \theta^\star$ by the definition of $\Omega_i^\star(\{z_j\}_{j \neq i})$. Since $\max_{j \neq i} z_j^\top \theta^\star = b \leq H$, the support of $f_2$ becomes

$$\{z \in A \mid z^\top \theta^\star \geq b\}.$$

Recall $A := \mathrm{supp}(Z_i)$ and it has equal sections with direction $\theta^\star$. Using Lemma 1, $f_2(\cdot)$ has LAC with $\mathcal{L}(R)$, and it has bounded decay rate $\sqrt{d}\mathcal{L}(R)$ by Lemma 3.

**[3] Bounding one-side decay rate.** Next, we aim to bound the one-side decay rate of $Z_i^\top \theta^\star = y \mid \{\max_{j \neq i} z_j^\top \theta^\star \leq H\} \cap \Omega_i^\star(\{z_j\})$. Define the (conditional) density of

$$Z_i^\top \theta^\star = y \mid \{\max_{j \neq i} z_j^\top \theta^\star \leq H\} \cap \Omega_i^\star(\{z_j\})$$

as $g_2(y)$. Its support is restricted to the region $\{y \mid b \leq y \leq H + 1\}$.

**Claim 3** *The one-side decay rate of*

$$Z_i^\top \theta^\star = y \mid \{\max_{j \neq i} z_j^\top \theta^\star \leq H\} \cap \Omega_i^\star(\{z_j\})$$

*is bounded by $\sqrt{d}\mathcal{L}(R)$ in the interval $[b, b + \frac{1}{2}]$.*

**Proof of Claim** First, we observe that the support of

$$Z_i \mid \{\max_{j \neq i} z_j^\top \theta^\star \leq H\} \cap \Omega_i^\star(\{z_j\})$$

is an interval $[b, H + 1]$. This follows from the definition of $\Omega_i^\star(\{z_j\})$ and our design of $A$. Moreover, this support has equal sections with direction $\theta^\star$. Since $b \leq H$, Lemma 8 applies, yielding the desired result. ∎

**3) Bounding maximum density of $g_2$ by applying Corollary 5.** From Lemma 5, the maximum density is bounded by $3\sqrt{d}\mathcal{L}(R)$ in the interval $[b, b + \frac{1}{2}]$. Hence,

$$\mathbb{E}[\mathbf{1}(\max_{j \neq i} z_j^\top \theta^\star \leq Z_i^\top \theta^\star \leq \max_{j \neq i} z_j^\top \theta^\star + \varepsilon) \mid \{\max_{j \neq i} z_j^\top \theta^\star \leq H\} \cap \Omega_i(\{z_j\})] \leq 3\sqrt{d}\mathcal{L}(R)\varepsilon$$

for any $\varepsilon < \frac{1}{2}$.

∎

# G   Proofs of Results for Unbounded Contexts

In this section, we prove our main theorems stated in Section 4.3. We aim to apply the key propositions, Proposition 2 and 3. To apply them, we first truncate our contexts to an appropriate region to fulfill the conditions of the propositions. First, we present ways to construct a truncation set, which is used to prove Theorems 2 and 3. After that, we prove the two Theorems 2 and 3 by applying Proposition 2 and 3.

## G.1   Constructing Truncation Sets

This section presents operations to construct truncation sets that will be used in the proof of our main results. In the proof, we first truncate our contexts $\mathbf{X} \in \mathbb{R}^{dK}$ to a truncation set $\mathbf{C} = \prod_{i=1}^{K} D$, where $D \subset \mathbb{R}^d$, and work with the truncated contexts. If the supremum norm of the set is bounded, by combining it with LAC, we can bound the decay rate of the truncated contexts.

First, we define the directional completion of $v$. For a set $A$ and a vector $v$, we define the process of filling $A$ in the direction of $v$. This process expands $A$ so that its cross-section remains the same when cut by a hyperplane orthogonal to $v$. This procedure ensures that all sections of $A$ along the direction $v$ are equal. Recall that $\pi_S(A)$ denotes the projection of $A$ onto the subspace $S$. If $S$ is the subspace spanned by a vector $v$, we write $\pi_v(A)$, which is a subset of a straight line.

**Definition 12 (Set completion)** *For $A \subset \mathbb{R}^d$ and $v \in \mathbb{S}^{d-1}$, we define*

$$\mathcal{C}[A, v] := \pi_{\langle v \rangle^\perp}(A) + v \, \pi_v(A).$$

After this completion, the section sliced by the normal hyperplane of $v$ is the same.

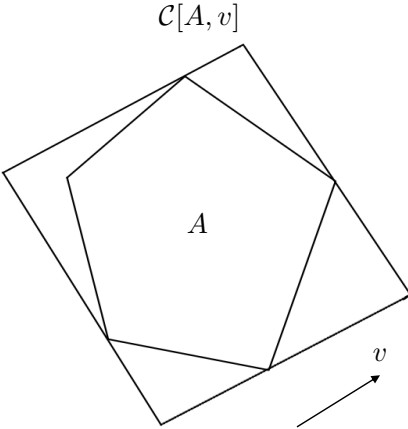

Figure 8: Illustration of $\mathcal{C}(A, v)$. This is the operation of filling the area $A$ in the $v$-direction. Then, *all sections in the $v$-direction become equal*.

**Lemma 11** *Suppose a set $A \in \mathbb{R}^d$ and a unit vector $v$ satisfy $\pi_v(A)$ is an interval with length $\ell$. Then,*

$$\|\mathcal{C}[A, v]\|_\infty \leq \|A\|_\infty + \ell.$$

**Proof** For all $p \in \mathcal{C}[A, v]$, there exists $q \in A$ and some $h$ with $|h| < \ell$ such that $p = q + hv$. Hence,

$$\|p\|_\infty \leq \|q\|_\infty + \|hv\|_\infty \leq \|A\|_\infty + \ell.$$

∎

Next, we define partial completion, which makes the section with direction $v$ in the region of $\mathcal{C}(A, v) \cap [-1, 1]_v$ equal.

**Definition 13 (Partial completion)** *For a set $A \subset \mathbb{R}^d$ and a unit vector $v \in \mathbb{R}^d$, define*

$$\mathcal{P}(A, v) := \mathcal{C}[A \cap \{x \mid |x \cdot v| \leq 1\}, v] \cup A.$$

Recall that $\pi_v(\cdot)$ is a projection onto the direction $v$.

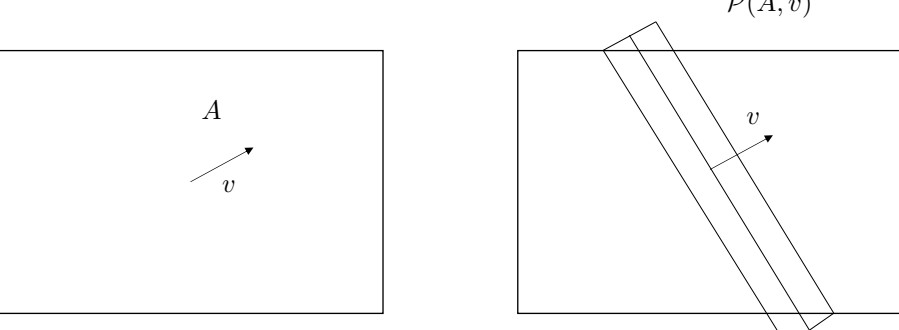

Figure 9: Illustration of $\mathcal{P}(A, v)$.

$\mathcal{P}(A, v)$ has the same section with the direction $v$ within the region $[-1, 1]_v$. Equivalently,

$$\mathcal{P}(A, v) \cap \{x \mid x^\top v = u_1\} \equiv \mathcal{P}(A, v) \cap \{x \mid x^\top v = u_2\}$$

for every $u_1 \neq u_2 \in [-1, 1]$. The partial completion operator has a bounded supremum norm, as shown in the following lemma.

**Lemma 12 (Sup norm bound of $\mathcal{P}(A, v)$)** *For any $A \subset \mathbb{R}^d$ and $v \in \mathbb{S}^{d-1}$,*

$$\|\mathcal{P}(A, v)\|_\infty \leq \|A\|_\infty + 2$$

*holds.*

**Proof** For any $y \in \mathcal{P}(A, v)$, there is $y' \in A$ such that $y = y' + tv$ for some $|t| \leq 2$. Hence,

$$\|y\|_\infty \leq \|y'\|_\infty + \|tv\|_\infty \leq \|A\|_\infty + 2.$$

∎

## G.2 Proof of Theorem 2

Our goal is to prove

$$\mathbb{E}[X_{a_\theta(\mathbf{X}(t))}(t) X_{a_\theta(\mathbf{X}(t))}(t)^\top] \geq \lambda_\star(t) > 0$$

for any $t$ and $\theta$. Fix an arbitrary history $\mathcal{H}_{t-1}$ and any $\theta \in \mathbb{S}^d$. For simplicity, define $(X_1^\top, \ldots, X_K^\top)$ as the conditioned random variable of $(X_1(t)^\top, \ldots, X_K(t)^\top) \mid \mathcal{H}_{t-1}$. Under the history $\mathcal{H}_{t-1}$, we aim to apply Proposition 2 for any $\theta$ and $v$. We also set $a = \arg\max X_i^\top \theta$. For any fixed $v \in \mathbb{R}^d$ with $\|v\| = 1$, our goal is to calculate the lower bound of

$$\mathbb{E}[v^\top X_a X_a^\top v].$$

We view it as a *fixed history random variable*, and we aim to use arguments developed in Section F.

To use Proposition 2, we first truncate our contexts $(X_1^\top, \ldots, X_K^\top)$ into some region $\mathbf{C}_1^v$, then apply Proposition 2 to the truncated contexts. To satisfy the geometric conditions of Proposition 2, we construct the truncation as follows.

### G.2.1 Constructing Truncation Sets

We define the truncation set as follows:

1. Define $R_1 = c_0 x_{\max}(2 + \log dK)$.

2. Define $D := [-R_1, R_1]^d$.

3. Define $D^v := \mathcal{P}[D, v]$.

4. Set the truncation set $\mathbf{C}_1^v := (D^v)^K$.

5. Then $\|\mathbf{C}_1^v\|_\infty \leq R_1 + 2$ holds (by Lemma 12).

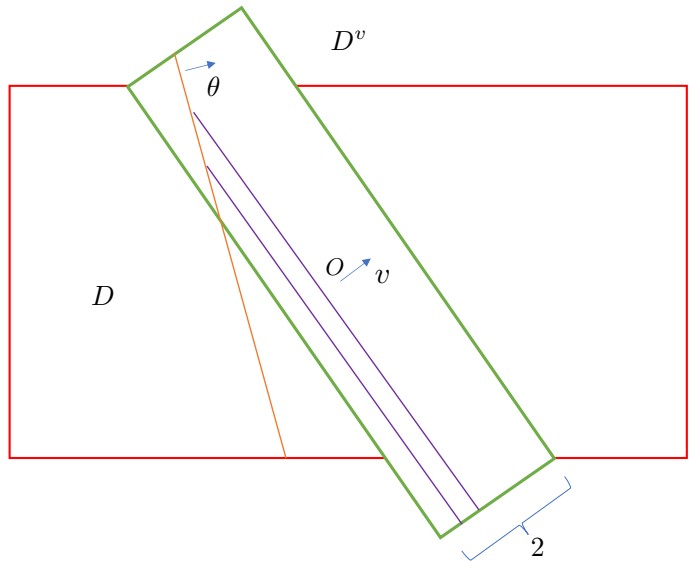

Figure 10: Illustration of the set $D^v$. The red rectangle is $D$, and $D^v$ is the union with the green boundary rectangle and $D$. We force the sections in the direction $v$ within $[-1, 1]_v$ to be equal. Hence, we can apply the previous results from Proposition 2.

**Claim 4** *For any $\mathbf{X}$ satisfying Assumption 2, $\mathbb{P}[\mathbf{X} \in \mathbf{C}_1^v] \geq \frac{1}{2}$.*

**Proof** First, we show $\mathbb{P}[\mathbf{X} \in D^K] \geq \frac{1}{2}$. Since $\mathbf{C}_1^v \supset D^K$, it suffices to prove the latter. Using (14), we get

$$\mathbb{P}[\mathbf{X} \in (D^K)^c] \ \leq \ \sum_{i=1}^{K} \mathbb{P}[X_i \in D^c] \ \leq \ \sum_{i=1}^{K}\sum_{j=1}^{d} \mathbb{P}[|X_{ij}| > R_1] \ \leq \ dK \times \frac{1}{2dK} = \frac{1}{2}.$$

∎

Define the truncated contexts $\mathbf{W} = \mathbf{X} \mid \{\mathbf{X} \in \mathbf{C}_1^v\} := (W_1^\top, \dots, W_K^\top)$.

**Claim 5** *The truncated contexts $\mathbf{W}$ satisfy*

$$\mathbb{E}[v^\top X_a X_a^\top v] \ \geq \ \frac{1}{2}\mathbb{E}[v^\top W_a (W_a)^\top v].$$

**Proof** Recall $a = \arg\max_{i \in [K]} X_i^\top \theta$. Note that

$$\mathbb{E}[v^\top X_a X_a^\top v] \ \geq \ \mathbb{E}[v^\top X_a X_a^\top v \, \mathbb{I}_{\{\mathbf{X}=\mathbf{W}\}}] \ + \ \mathbb{E}[v^\top X_a X_a^\top v \, \mathbb{I}_{\{\mathbf{X}\neq\mathbf{W}\}}].$$

Since $\mathbb{I}_{\{\mathbf{X}\neq\mathbf{W}\}} \leq 1$, and $\mathbb{P}[\mathbf{X} = \mathbf{W}] \geq \frac{1}{2}$, we get

$$\mathbb{E}[v^\top X_a X_a^\top v] \ \geq \ \mathbb{E}[v^\top X_a X_a^\top v \, \mathbb{I}_{\{\mathbf{X}=\mathbf{W}\}}] \ \geq \ \mathbb{E}[v^\top W_a W_a^\top v \mid \mathbf{X} = \mathbf{W}] \, \mathbb{P}[\mathbf{X} = \mathbf{W}] \ \geq \ \frac{1}{2}\mathbb{E}[v^\top W_a W_a^\top v].$$

Hence, we only need to bound the diversity of the truncated contexts $\mathbf{W} = (W_1^\top, \dots, W_K^\top)$. ∎

### G.2.2  Properties of Truncated Contexts $\mathbf{W}$

We need to check two conditions: (i) The sections of $\mathrm{supp}(W_i)$ in the direction $v$ within $[-1, 1]_v$ are the same for all $i$. (ii) $\mathbf{W}$ has a bounded decay rate $\sqrt{d}\,\mathcal{L}(R_1 + 2)$.

Since $\mathbf{X}$ satisfies LAC with functions $\mathcal{L}$ and $\|\mathbf{C}_1\|_\infty \leq R_1 + 2$, it follows from Lemma 3 that $\mathbf{X}$ has a bounded decay rate of $\sqrt{d}\,\mathcal{L}(R_1 + 2)$.

**Claim 6** *The truncated contexts* $\mathbf{W}$ *have LAC with* $\mathcal{L}(\cdot)$*, and* $\|\operatorname{supp}(\mathbf{W})\|_\infty \leq R_1 + 2$.

**Proof** By Appendix C.3, conditioning on the event $\{\mathbf{X} \in \mathbf{C}_1^v\}$ preserves LAC with the same function $\mathcal{L}$. Also, by construction of $\mathbf{C}_1^v$ and Lemma 11, $\|\operatorname{supp}(\mathbf{W})\|_\infty \leq R_1 + 2$. ∎

**Claim 7** *For every* $i \in [K]$*, the support* $\operatorname{supp}(W_i)$ *is identical for all* $i$*, and* $\operatorname{supp}(W_i) \cap [-1, 1]_v$ *has equal sections in the* $v$*-direction.*

**Proof** Since we set $D^v$ as the partial completion of $D$ with direction $v$, the sections in $[-1, 1]_v$ become equal by construction. ∎

### G.2.3 Applying Proposition 2

We have verified that our truncated contexts $\mathbf{W}$ satisfy the conditions of Proposition 2. Hence, by Proposition 2, for some absolute constant $c' > 0$,

$$\mathbb{E}[W_a W_a^\top] \succeq \frac{c'}{d\left(A_1 + A_2(R_1 + 2)^\alpha\right)^2} I_d.$$

Combining with Claim 5, we get

$$\mathbb{E}[X_a X_a^\top] \succeq \frac{c}{d\left(A_1 + A_2(R_1 + 2)^\alpha\right)^2} I_d$$

for some absolute constant $c > 0$ and it ends proof.

### G.3 Proof of Theorem 3

We now use the result of Proposition 3. First consider the case $\|\theta^\star\|_2 = 1$. For the general case, we can adjust via the argument in Appendix B.

Truncate our contexts $\mathbf{X} = (X_1^\top \ldots X_K^\top)$ to some region $\mathbf{C}_2 = D^K$, where $D \subset \mathbb{R}^d$, with a slight abuse of notation. Define the truncated contexts as $\mathbf{W} = (W_1^\top, \ldots, W_K^\top)$. Note that the support of $W_i$ is $D$. To satisfy the conditions of Proposition 3, we want the sections of $\operatorname{supp}(W_i)$ in the direction $\theta^\star$ to be the same.

#### G.3.1 Constructing Truncation Sets

We construct the truncation set $\mathbf{C}_2$ as follows. First, set $R_3 = c_0 \, x_{\max}(1 + \log dK + \frac{1}{2}\log T)$ and $R_2 = R_3 + 1$.

- Set $D_1 := ([-R_2, R_2])^d \subset \mathbb{R}^d$.
- Set $D_2 := \{x \in \mathbb{R}^d \mid -R_2 \leq x^\top \theta^\star \leq R_2\} \subset \mathbb{R}^d$.
- Set $D_3 := D_1 \cap D_2$.
- Set $D := \mathcal{C}[D_3, \theta^\star]$.
- The final truncation set is $\mathbf{C}_2 = D^K$.

**Claim 8** *The sup norm of the truncation set satisfies* $\|D\|_\infty \leq 3R_2$. *Also,* $\mathbf{W}$ *has bounded LAC with* $\mathcal{L}(3R_2)$.

**Proof** By Lemma 1, the LAC of $\mathbf{W}$ is the same as the original, $\mathcal{L}(\cdot)$. By Lemma 11, $\|D\|_\infty \leq 3R_2$ since $\pi_{\theta^\star}(D_2)$ is an interval with length $2R_2$. Since $\mathcal{L}(\cdot)$ is increasing and $\|D\|_\infty \leq 3R_2$, $\mathbf{W}$ has bounded LAC with $\mathcal{L}(3R_2)$. ∎

**Claim 9** *The region* $\mathbf{C}_2$ *has equal sections in the direction* $\theta^\star$ *and* $\mathbb{P}[\mathbf{X} \in \mathbf{C}_2] \geq 1 - \frac{1}{\sqrt{T}}$.

**Proof** By Assumption 2, $\mathbb{P}[|X_{ij}| \leq R_2] \geq 1 - \frac{1}{2dK\sqrt{T}}$ holds for all $i, j$. Hence, $\mathbb{P}[\mathbf{X} \in D_1^K] \geq 1 - \frac{1}{2\sqrt{T}}$. By the same assumption, $\mathbb{P}[\mathbf{X} \in D_2^K] \geq 1 - \frac{1}{2\sqrt{T}}$. Thus, $\mathbb{P}[\mathbf{X} \in D_3^K] \geq 1 - \frac{1}{\sqrt{T}}$. Since $D \supset D_3$, $\mathbb{P}[\mathbf{X} \in \mathbf{C}_2] \geq 1 - \frac{1}{\sqrt{T}}$. ∎

### G.3.2 Truncation with a High-Probability Region

We truncate $\mathbf{X}$ into the high-probability region $\mathbf{C}_2$ and define the truncated contexts $\mathbf{W} = (W_1^\top, \ldots, W_K^\top)$. We then work with $\mathbf{W}$.

The following calculation shows that it suffices to bound the suboptimality gap of $\mathbf{W}$:

$$\mathbb{P}[\Delta(\mathbf{X}) \leq \varepsilon] = \int_{\mathbb{R}^{Kd}} \mathbb{I}_{\{\Delta(\mathbf{x}) \leq \varepsilon\}} \mathbf{f}(\mathbf{x})\, d\mathbf{x} = \int_{\mathbf{C}_2} \mathbb{I}_{\{\Delta(\mathbf{x}) \leq \varepsilon\}} \mathbf{f}(\mathbf{x})\, d\mathbf{x} + \int_{\mathbf{C}_2^c} \mathbb{I}_{\{\Delta(\mathbf{x}) \leq \varepsilon\}} \mathbf{f}(\mathbf{x})\, d\mathbf{x}$$

$$\leq \int_{\mathbf{C}_2} \mathbb{I}_{\{\Delta(\mathbf{x}) \leq \varepsilon\}} \mathbf{f}(\mathbf{x})\, d\mathbf{x} \; + \; \mathbb{P}[\mathbf{X} \in \mathbf{C}_2^c]$$

$$= \mathbb{P}[\mathbf{X} \in \mathbf{C}_2] \int_{\mathbf{C}_2} \mathbb{I}_{\{\Delta(\mathbf{x}) \leq \varepsilon\}} \frac{\mathbf{f}(\mathbf{x})}{\mathbb{P}[\mathbf{X} \in \mathbf{C}_2]}\, d\mathbf{x} \; + \; \frac{1}{\sqrt{T}}$$

$$= \mathbb{P}[\mathbf{X} \in \mathbf{C}_2]\, \mathbb{P}[\Delta(\mathbf{W}) \leq \varepsilon] \; + \; \frac{1}{\sqrt{T}}$$

$$\leq \mathbb{P}[\Delta(\mathbf{W}) \leq \varepsilon] + \frac{1}{\sqrt{T}}. \tag{11}$$

Therefore, if we find the constant $C_\Delta'$ satisfying

$$\mathbb{P}[\Delta(\mathbf{W}) \leq \varepsilon] \; \leq \; C_\Delta'\, \varepsilon, \tag{12}$$

we get $C_\Delta \leq C_\Delta'$.

**Lemma 13** *The truncated contexts* $\mathbf{W} = \mathbf{X} \mid \{\mathbf{X} \in \mathbf{C}_2\}$ *meet the condition of Proposition 3 with* $H = R_2 + 1$ *and* $\delta = \frac{1}{\sqrt{T}}$.

**Proof** Let $D = \mathrm{cyl}(B, \theta^\star, R_2)$ for some set $B \subset P_{\theta^\star}$, and define $D' = \mathrm{cyl}(B, \theta^\star, R_2 - 1)$. By a similar argument to Claim 9, $\mathbb{P}[\mathbf{X} \in D'] \geq 1 - \frac{1}{\sqrt{T}}$. Hence,

$$\mathbb{P}[\mathbf{W} \in D'] = \frac{\mathbb{P}[\mathbf{X} \in D' \cap \{\mathbf{X} \in \mathbf{C}_2\}]}{\mathbb{P}[\mathbf{X} \in \mathbf{C}_2]} \; \geq \; \frac{\mathbb{P}[\mathbf{X} \in D']}{\mathbb{P}[\mathbf{X} \in \mathbf{C}_2]} \; \geq \; \frac{1 - \frac{1}{\sqrt{T}}}{1} = 1 - \frac{1}{\sqrt{T}}.$$

Thus, the condition of Proposition 3 is satisfied with $\delta = \frac{1}{\sqrt{T}}$. ∎

### G.3.3 Applying Proposition 3

By Proposition 3, we can ensure that (12) holds for

$$C_\Delta' = 3\sqrt{d}\, \mathcal{L}(R_2) = 3\sqrt{d}\, \big(A_1 + A_2(3R_2)^\alpha\big).$$

Hence,

$$C_\Delta \; \leq \; 3\sqrt{d}\, \big(A_1 + A_2(3R_2)^\alpha\big) \; = \; \widetilde{\mathcal{O}}(\sqrt{d}).$$

### G.4 Proof of Theorem 1: Unbounded Contexts Case

Combining Proposition 9 with our results from Theorems 2 and 3, we get

$$\mathbf{Reg}(T) \; \leq \; c\, C_\Delta\, d\, x_{\max}^2\, \frac{1}{\lambda_\star}\, (\log T)^4.$$

By substituting our bounds on $C_\Delta$ and $\lambda_\star$, we have

$$\mathbf{Reg}(T) \; \leq \; c\, C_\Delta\, d^{2.5} x_{\max}^2 \big(A_1 + A_2(R_1 + 2)^\alpha\big)^2 \big(A_1 + A_2\, 2^\alpha R_2^\alpha\big) (\log T)^4 \; \leq \; \widetilde{\mathcal{O}}\big(d^{2.5}\big).$$

# H  Results for Bounded Contexts

Now, we present our main result for bounded contexts.

## H.1  Two Cases of Bounded Contexts

In the linear contextual bandit setting, $\ell_2$ boundedness is widely used. However, for light-tailed distributions (such as Gaussian or exponential), the $\ell_2$ norm is unbounded. Therefore, we divide our analysis into two cases: unbounded and bounded contexts. While our previous focus was on unbounded contexts, here we summarize and present results for bounded contexts.

We classify the analysis into two cases: the first is for truncated contexts, and the second is for naturally bounded contexts, which include cases where the uniform distribution on the ball $\mathbb{B}_R$ or a distribution with bounded density on the ball.

**1) Truncated contexts.**   Truncated contexts refer to cases where the context distribution is truncated from the original distribution. Examples include truncated Gaussian and truncated exponential distributions.

**2) Naturally bounded contexts.**   Naturally bounded contexts include uniform distributions on a ball or distributions with bounded density defined on the ball.

## H.2  Regret Bounds for Bounded Contexts

Next, we present our result for the regret bound. The first case is when we receive truncated contexts, generated by truncating unbounded contexts to $(\mathbb{B}_R)^K$, where $\mathbb{B}_R$ is a $d$-dimensional ball of radius $R$. For the $\ell_2$-bounded case, $R$ may depend on the dimension $d$ when we choose a large $R$, so we do not hide the $R$ term in our main regret bound. Corollary 1 and 2 gives the regret bound for truncated contexts. Corollary 3 gives the regret bound for naturally bounded contexts.

**Corollary 1 (Regret bound: truncated contexts)** *Let* $\mathbf{X}(t)$ *be unbounded contexts with the same condition as Theorem 1. For* $R' > 0$ *with* $\mathbb{P}[\mathbf{X}(t) \in (\mathbb{B}_{R'})^K] = p > 0$, *we receive truncated contexts* $\mathbf{X}(t) \mid (\mathbb{B}_{R'+r})^K$ *for some* $r > 0$ *each round* $t \geq 1$. *In this case, for* $R = R' + r$, *the regret of Algorithm 1 is bounded by*

$$\mathbf{Reg}(T) \leq \widetilde{O}(d^{2.5} R^2 \mathcal{L}(R)^2 \frac{1}{p}).$$

*when* $r \asymp R'$.

**Corollary 2 (Regret bound: high-prob truncated contexts)** *Let* $\mathbf{X}(t)$ *be unbounded contexts with the same condition as Theorem 1. We receive truncated contexts* $\mathbf{X}(t) \mid (\mathbb{B}_L)^K$ *for* $L \gtrsim \sqrt{d}x_{\max}(1 + \log d + \log K)$ *each round* $t$. *In this case, the regret of Algorithm 1 is bounded by*

$$\mathbf{Reg}(T) \leq \widetilde{O}(d^{2.5}).$$

**Discussion.**   For Corollary 1, when we receive truncated light-tailed distributions (e.g., Gaussian, exponential, Laplace), this theorem states that they enjoy a logarithmic regret bound. It has a $\mathrm{poly}(\log T)$ regret bound for $T$ and depends on $R$, the truncation radius. Corollary 2 states that when we choose a sufficiently large truncation radius, the bound becomes radius-free, matching the unbounded case (Theorem 1). Proofs for these corollaries are presented in Appendix I.

Next, we introduce the regret bound results for naturally bounded contexts. Here, new parameters $c_\star$ and $p_\star$ are introduced, along with Condition 1. This condition will be discussed in Appendix H.3, where we clarify that both uniform distributions and bounded-density distributions satisfy it. We will provide detailed explanations and state the range of $c_\star, p_\star$ for several distributions, including the uniform distribution.

**Corollary 3 (Regret bound: naturally bounded contexts)** *Let naturally bounded contexts* $\mathbf{X}(t) \in (\mathbb{B}_R)^K$ *satisfy the LAC condition with function* $\mathcal{L}(\cdot)$ *and Condition 1 holds with concentration*

*parameters $c_\star, p_\star$ each round $t$. Under Assumption 1 and 2, the regret of Algorithm 1 is bounded by*

$$\mathbf{Reg}(T) \leq \widetilde{O}\left(Kd^{2.5}R^2\mathcal{L}(R)^3 \frac{1}{p_\star(1-c_\star)^2}\right).$$

**Discussion.** Corollary 3 applies to the uniform distribution and any distribution with bounded density on the ball. It also covers the truncation of heavy-tailed distributions. For Corollary 3, the polynomial dependency on $K$ is due to boundedness, and it matches the asymptotic result of the uniform distribution studied in [12] when $d = 1$. Combining with Lemma 15, for a uniform distribution or a bounded density distribution in the ball, we have $\mathbf{Reg}(T) \leq \widetilde{\mathcal{O}}(K^{\frac{d+5}{d+1}}d^{2.5})$. For Condition 1 and parameters $c_\star$ and $p_\star$, we deeply discuss it in the following part. Proofs for these corollaries are presented in Appendix I.

**Comparison with known results.** For truncated contexts, to the best of our knowledge, there are no known results for greedy bandits. For naturally bounded contexts, Oh et al. [28] studied the case of a uniform distribution on the sphere, where each context $X_i$ for arm $i$ is independent across arms, and they only considered the case $K \leq d$. Their work assumes the minimum eigenvalue of the context covariance matrix is constant; however, for a uniform distribution, it scales as $\asymp \frac{1}{d}$. Taking this into account, their regret bound is $\widetilde{\mathcal{O}}(d^3\sqrt{T})$ when $K \leq d$. When $K \leq d$, our result has the bound $\widetilde{\mathcal{O}}(d^{2.5+\frac{d+5}{d+1}})$ for the uniform distribution. Moreover, for the multi-parameter shared context setup, Bastani and Bayati [7], Bastani et al. [8] studied the uniform context case, and their worst-case regret bound is $\widetilde{\mathcal{O}}(K^4d^4)$, considering that the minimum eigenvalue of the context covariance scales as $\frac{1}{d}$. In conclusion, our result is the sharpest among known previous results.

### H.3 Concentration Parameters for Bounded Contexts

We introduce a new condition that defines two concentration parameters measuring sufficient concentration for bounded contexts. We will show that truncated contexts satisfy this condition, as do naturally bounded contexts with bounded density. This condition does not need to be assumed for truncated contexts, as it is automatically satisfied.

**Condition 1 (Concentration parameters for bounded contexts)** *For the random vectors (contexts)* $\mathbf{X} = (X_1^\top \ldots X_K^\top) \in \mathbb{R}^{dK}$ *where* $X_i \in \mathbb{B}_R$, *there exist* $0 < c_\star < 1$ *and* $0 < p_\star < 1$ *such that for every* $\eta$ *with* $\|\eta\|_2 = 1$,

$$\mathbb{P}\left[\max_{i\in[K]} X_i^\top \eta \leq c_\star R\right] \geq p_\star.$$

*We call $p_\star, c_\star$ the concentration parameters.*

**Discussion.** Two parameters, $c_\star$ and $p_\star$, must exist if $\mathbf{X}$ is random. Below, we discuss $1 - c_\star$ and $p_\star \asymp 1$ for truncated contexts truncated into the positive-probability region. Additionally, we show that a uniform distribution within the ball, as well as any distribution within the ball with bounded density, satisfies this condition, and we explicitly calculate the range of these two parameters. For our regret bound, we have a dependence on $\frac{p_\star}{(1-c_\star)^2}$.

**Example: truncated contexts.** The next lemma states that truncated contexts satisfy Condition 1 with $1 - c_\star, p_\star \asymp 1$. If contexts are generated by truncating the original distribution into a positive-probability region, then the parameters in Condition 1, $c_\star, p_\star$, are well-defined and do not harm the regret bound when we choose a sufficiently large truncation set.

**Lemma 14 (Concentration parameters for truncated contexts)** *Suppose the unbounded contexts* $\mathbf{X} \in \mathbb{R}^{dK}$ *satisfy*

$$\mathbb{P}[\mathbf{X} \in (\mathbb{B}_{R'})^K] = p > 0,$$

*for some $R' > 0$. Then, if we truncate each $X_i$ to $\mathbb{B}_{R'+r}$ for any $r > 0, i \in [K]$, the truncated contexts $\overline{\mathbf{X}} = (\overline{X}_1^\top, \ldots, \overline{X}_K^\top)$ satisfy Condition 1 with $p_\star = p, c_\star = \frac{R'}{R'+r}$.*

**Proof**    Recall that we truncate $\mathbf{X}$ to $(\mathbb{B}_{R'+r})^K$. If $\mathbb{P}[\mathbf{X} \in (\mathbb{B}_{R'})^K] = p > 0$ and we truncate to $\mathbb{B}_{R'+r}$,

$$
\begin{aligned}
\mathbb{P}\Big[\max_{i \in [K]} \overline{X}_i^\top \eta \le R'\Big] &= \frac{\mathbb{P}\big[\{\max X_i^\top \eta \le R'\} \cap \{\mathbf{X} \in (\mathbb{B}_{R'+r})^K\}\big]}{\mathbb{P}\big[\mathbf{X} \in (\mathbb{B}_{R'+r})^K\big]} \\
&\ge \frac{\mathbb{P}\big[\mathbf{X} \in (\mathbb{B}_{R'})^K\big]}{\mathbb{P}\big[\mathbf{X} \in (\mathbb{B}_{R'+r})^K\big]} \\
&\ge \mathbb{P}\big[\mathbf{X} \in (\mathbb{B}_{R'})^K\big] \\
&= p.
\end{aligned}
$$

Therefore, we can set $p_\star = p$ and $c_\star = \frac{R'}{R'+r}$. ∎

**Example: Bounded Density Distributions**    The lemma below tells us that a uniform distribution or a bounded density distribution on $\mathbb{B}_R$ satisfies Condition 1 and provides the range of the two parameters. Without loss of generality, we prove it for a distribution defined on the unit ball $\mathbb{B}_1$.

**Lemma 15** *Consider a random vector $X_i \in \mathbb{B}_1, i \in [K]$, each with density upper-bounded by $\frac{c_u}{\omega_d}$. Here, $\omega_d$ is the volume of the $d$-dimensional unit ball $\mathbb{B}_1$. (Recall that the density of the uniform distribution in $\mathbb{B}_1$ is $\frac{1}{\omega_d}$.)*

1. *When $X_1 = \cdots = X_K$ (strongly correlated), Condition 1 holds with*

$$
p_\star = \frac{1}{2}, \quad 1 - c_\star \gtrsim 1.
$$

2. *When $X_1, \ldots, X_K$ are independent (not correlated), we have*

$$
p_\star = \frac{1}{2}, \quad 1 - c_\star \ge c \, K^{-\frac{2}{d+1}}.
$$

**Proof**  We use the result of Claim 10 below.

**Correlated Case.**    If $X_1 = X_2 = \cdots = X_K$, then the $\frac{1}{2}$-quantile $c_\star$ of $X_i^\top \eta$ satisfies

$$
2(1 - c_\star) \ge \Big(\frac{1}{2c_u} \frac{\omega_d}{\omega_{d-1}}\Big)^{\frac{2}{d+1}}.
$$

Since $\big(\frac{\omega_d}{\omega_{d-1}}\big)^{\frac{2}{d+1}} \asymp 1$, it follows that

$$
1 - c_\star \gtrsim 1.
$$

**Independent Case.**    If $X_1^\top, \ldots, X_K^\top$ are independent, then for a certain $c_\star$ with

$$
2(1 - c_\star) \ge 1 - c_\star^2 = \Big(\frac{1}{Kc_u} \frac{\omega_d}{\omega_{d-1}}\Big)^{\frac{2}{d+1}} \asymp K^{-\frac{2}{d+1}},
$$

we have $\mathbb{P}[X_i^\top \eta \le c_\star R] \le 1 - \frac{1}{K}$ for all $i \in [K]$. By independence, $p_\star = (1 - \frac{1}{K})^K \le \frac{1}{2}$. ∎

**Claim 10** *Consider a random vector $X \in \mathbb{R}^d$ defined in $\mathbb{B}_1$ with density upper-bounded by $\frac{c_u}{\omega_d}$. For any $\eta \in \mathbb{S}^{d-1}$, let the $(1-p)$-quantile of $X^\top \eta$ be $\alpha$. Then*

$$
2(1 - \alpha) \ge 1 - \alpha^2 \ge \Big(\frac{p}{c_u} \frac{\omega_d}{\omega_{d-1}}\Big)^{\frac{2}{d+1}}.
$$

**Proof**  We show that $\alpha$ with $1 - \alpha^2 = \big(\frac{p}{c_u} \frac{\omega_d}{\omega_{d-1}}\big)^{\frac{2}{d+1}}$ satisfies

$$
\mathbb{P}[X^\top \eta \ge \alpha] \le p.
$$

Since the density of $X^\top \eta = r$ satisfies

$$\mathbb{P}[X^\top \eta = r] \leq \omega_{d-1}(1 - r^2)^{\frac{d-1}{2}} \frac{c_u}{\omega_d},$$

we have

$$\begin{aligned}
\mathbb{P}[X^\top \eta \geq \alpha] &\leq \int_\alpha^1 \omega_{d-1}(1 - r^2)^{\frac{d-1}{2}} c_u \frac{1}{\omega_d} \, \mathrm{d}r \\
&\leq c_u \frac{\omega_{d-1}}{\omega_d}(1 - \alpha)(1 - \alpha^2)^{\frac{d-1}{2}} \\
&\leq c_u \frac{\omega_{d-1}}{\omega_d}(1 - \alpha^2)^{\frac{d+1}{2}} \\
&\leq p.
\end{aligned}$$

$\blacksquare$

## H.4   Results for Two Challenges: Bounded Contexts

Now, we present results related to two challenges for bounded contexts. Proofs are provided in Appendix I.

**Proposition 4 (Diversity constant: naturally bounded contexts)** *Suppose $\mathbf{X}(t)$ is a random vector supported in $(\mathbb{B}_R)^K$, its density satisfies the LAC condition with constant function $\mathcal{L}(R)$, and Condition 1 with $p_\star, c_\star$ holds. Then*

$$\lambda_\star(t) \geq c \frac{p_\star}{d} \frac{1}{\left(\mathcal{L}(R) + \frac{1}{R(1 - c_\star)}\right)^2}$$

*for some absolute constant $c > 0$.*

**Proposition 5 (Diversity constant: high-prob truncated contexts)** *Suppose $\mathbf{X}(t) \in \mathbb{R}^{d \times K}$ satisfies LAC with $\mathcal{L}(\cdot)$ and Assumption 2. Then we truncate $\mathbf{X}(t)$ to $(\mathbb{B}_L)^K$ for some $L \geq c\sqrt{d}x_{\max}\left(1 + \log\left(\frac{dKx_{\max}}{\lambda_\star}\right)\right)$ and define these truncated contexts as $\overline{\mathbf{X}}(t) = (\overline{X}_1(t), \ldots, \overline{X}_K(t))$. Then $\overline{\mathbf{X}}(t)$ has a diversity constant (Challenge 1) with $\frac{1}{2}\lambda_\star(t)$, where $\lambda_\star(t)$ is the diversity constant of $\mathbf{X}(t)$.*

**Proposition 6 (Suboptimality gap: truncated contexts)** *Let $\overline{\mathbf{X}}(t)$ be a truncated random variable of $\mathbf{X}(t)$ into $(\mathbb{B}_R)^K$ with $\mathbb{P}[\mathbf{X} \in (\mathbb{B}_R)^K] \geq 1 - \delta$ for some $\delta > 0$. Let $C_\Delta(t)$ be the margin constant of the contexts before truncation. Then the margin constant of truncated contexts $\overline{\mathbf{X}}(t)$, denoted by $\overline{C}_\Delta(t)$, satisfies*

$$\overline{C}_\Delta(t) \leq \frac{1}{1 - \delta} C_\Delta(t).$$

**Proposition 7 (Suboptimality gap: naturally bounded contexts)** *For naturally bounded contexts $\mathbf{X}(t)$ with LAC function $\mathcal{L}(\cdot)$, the margin constant is bounded by*

$$C_\Delta(t) \leq cK\sqrt{d}\mathcal{L}(R)$$

*for some absolute constant $c > 0$.*

**Discussion of Proposition 7.**   It has a linear factor in $K$, which can worsen the regret bound compared to LinUCB and LinTS when the number of arms $K$ is large. Nevertheless, this factor arises in the logarithmic regret bound, which is still advantageous compared to algorithms that achieve $\mathcal{O}(\sqrt{T})$ regret when $T \gg K$. Also, if every $X_i(t)$ follows a uniform distribution independently, then extreme value theory [12] implies there should be dependence on $K$. Hence, our result matches the lower bound for the uniform distribution, which indicates it cannot be improved.

# I   Proofs of Results for Bounded Contexts

We provide proofs for Appendix H. For simplicity, we assume $R \geq 1$. Using the observation of Appendix C.1, any density defined in $\mathbb{B}_R$ with LAC $\mathcal{L}(\cdot)$ has a bounded decay rate $\sqrt{d}\mathcal{L}(R)$ by combining Lemma 3.

## I.1   Fixed History Arguments

We fix again the time step $t$ and the history $\mathcal{H}_{t-1}$ and derive the fixed history results for the diversity and suboptimality gap. We use the same fixed history arguments in Appendix F. We set the history-conditioned contexts as $\mathbf{X} := \mathbf{X}(t) \mid \mathcal{H}_{t-1}$, and $\mathbf{X} = (X_1^\top, \ldots, X_K^\top), X_i \in \mathbb{R}^d$. To address any greedy policy with respect to $\hat{\theta}_{t-1}$, we propose an analysis that applies to any greedy policy with an arbitrary $\theta \in \mathbb{R}^d$. In Appendix F.1, we already argued that it suffices to bound this variance for any fixed $\theta$. Since we fix $\theta$, which is the corresponding value of $\hat{\theta}_{t-1}$ under the given history $\mathcal{H}_{t-1}$, we define the policy-selected arm $a = \arg\max X_i^\top \theta$. Following the previous definitions, define the event $\Omega_i := \{a = i\}$ and $\Omega_i(\{x_j\}_{j \neq i}) := \{a = i\} \cap \{X_j = x_j \text{ for all } j \neq i\}$. Similarly, define $\Omega_i^\star := \{a^\star = i\}$ and $\Omega_i^\star(\{x_j\}_{j \neq i}) := \{a^\star = i\} \cap \{X_j = x_j \text{ for all } j \neq i\}$.

**Event decomposition and conditional density.**   We first define $b = \max_{i \neq a} X_i^\top \theta$. Similar to the unbounded contexts, we decompose our diversity as

$$\mathbb{E}[v^\top X_a X_a^\top v] = \mathbb{P}[b \leq c_\star R] \, \mathbb{E}[v^\top X_a X_a^\top v \mid b \leq c_\star R] + \mathbb{P}[b > c_\star R] \, \mathbb{E}[v^\top X_a X_a^\top v \mid b > c_\star R]$$
$$\geq p_* \mathbb{E}[v^\top X_a X_a^\top v \mid b \leq c_\star R]$$
$$\geq p_* \mathbb{E}\Big[\mathbb{E}\big[v^\top X_a X_a^\top v \mid \{b \leq c_\star R\} \cap \Omega_i(\{x_j\}_{j \neq i})\big]\Big].$$

Therefore, our next interest is the projected contexts, defined as

$$X_i^\top v \,\Big|\, \Omega_i(\{x_j\}_{j \neq i}) \,\cap\, \{b \leq c_\star R\}.$$

Let us define this projected context's density as $g(\cdot)$. We first investigate the support of the conditional random vector,

$$X_i \,\Big|\, \Omega_i(\{x_j\}_{j \neq i}) \cap \{b \leq c_\star R\}$$

and define its density as $g(\cdot)$. The projected context's density $g$ is the integration of $f$ within the section $\{x \in \mathbb{B}_R \mid x^\top \theta \geq b\} \cap \{x \in \mathbb{B}_R \mid x^\top v = y\}$. Hence, the density of $X_i^\top v \,\Big|\, \Omega_i(\{x_j\}) \cap \{b \leq c_\star R\}$ is a section density of $\{x \in \mathbb{B}_R \mid x^\top \theta \geq b\}$ with direction $v$.

## I.2   Sections of the Ball

Next, we aim to bound the one-side decay rate of the section density in the ball. Unlike the previous sections, the sections corresponding to $\Omega_i(\{x_j\}_{j \neq i})$ no longer make expanding sections due to the boundary of $\mathbb{B}_R$. To deal with sections for bounded contexts, we define several sections of the ball.

**Definition 14 (Sliced ball)** *We define the sliced ball $\mathbb{S}_R(v, y)$ as*

$$\mathbb{S}_R(v, y) := \{x \in \mathbb{B}_R \mid x^\top v = y\}.$$

*Also define the double sliced ball*

$$\mathbb{S}_R(\theta, b, v, y) := \{x \in \mathbb{B}_R \mid x^\top v = y, \, x^\top \theta \geq b\}.$$

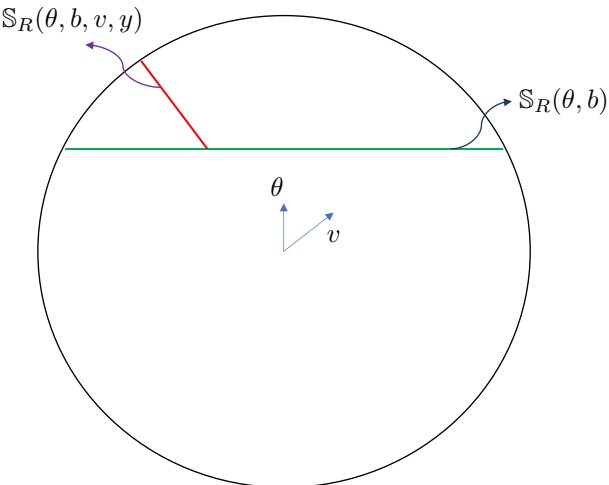

Figure 11: Illustrations of various sections in the ball $\mathbb{B}_R$. The red line is $\mathbb{S}_R(\theta, b, v, y)$ and the green line is $\mathbb{S}_R(\theta, b)$.

For bounded contexts, we need to bound the one-side decay rate of the section density, where the sections are sliced balls. Recall that we set $b = \max_{j \neq a} x_j^\top \theta$. We aim to obtain the lower bound of

$$\mathbb{E}\big[|v^\top X_i|^2 \mid \Omega_i(\{x_j\}_{j \neq i}) \cap \{b \leq c_\star R\}\big] \tag{13}$$

for any $\Omega_i(\{x_j\}_{j \neq i}), b$. Observe that the support of $v^\top X_i = y \mid \Omega_i(\{x_j\}_{j \neq i}) \cap \{b \leq c_\star R\}$ is the section of a double sliced ball, $\mathbb{S}_R(\theta, b, v, y)$. Here, we aim to bound the one-side decay rate of these section densities to apply Lemma 4.

### I.3 Linear Section Maps

To deal with varying sections, we define *linear section maps*, and using linear section maps, we can bound the one-side decay rate of the section densities. We first define a projection map, a projection to the center of similarity.

**Definition 15 (Projection map)** *We define the projection map between $A \subset \mathbb{R}^d$ and $P \in \mathbb{R}^d$ as a map $\Phi : A \to P$, which is an affine point projection with the center of similarity at $P$. Furthermore, we call $P$ the projection point.*

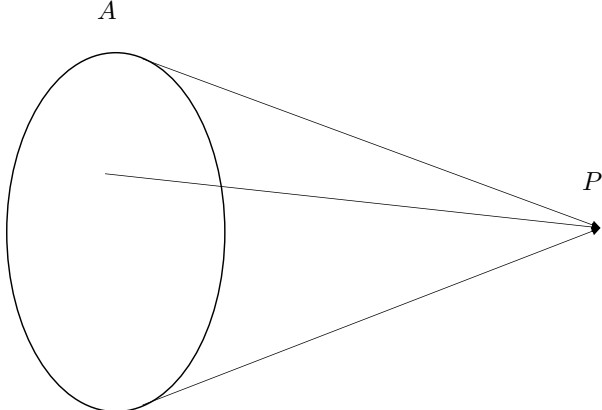

Figure 12: Illustration of a projection map.

**Remark 4** *A projection map means a homothety toward some point $P$.*

**Definition 16 (Linear section maps)** *We define linear section maps between sections. For two sections of A,* $\mathrm{Sec}(A, v, y)$ *and* $\mathrm{Sec}(A, v, y + h)$*, we define the linear section map* $\Phi_y^h$ *as*

$$\Phi_y^h(\cdot) : \mathrm{Sec}(A, v, y) \to \mathrm{Sec}(A, v, y + h)$$

*which satisfies* $\Phi_y^h(\mathrm{Sec}(A, v, y)) \subset \mathrm{Sec}(A, v, y + h)$ *and* $\Phi_y^h$ *is a part of some projection map with center* $P$*.*

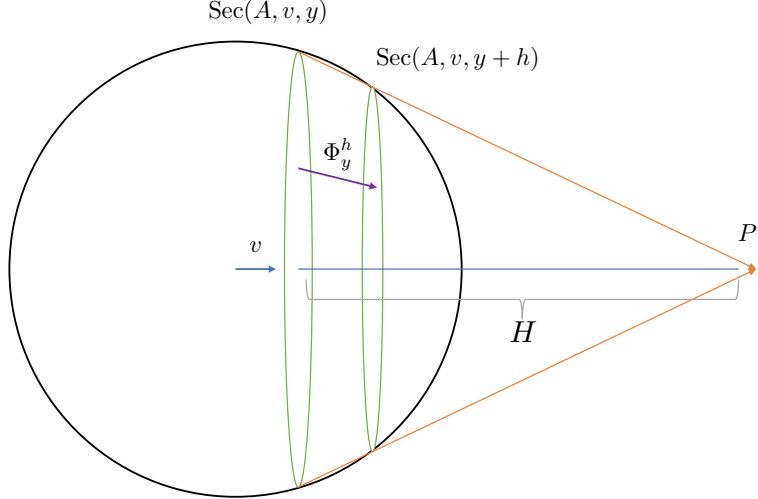

Figure 13: Illustration of linear section maps. A projection map of $\mathrm{Sec}(A, v, y)$ to $P$ induces a linear section map $\mathrm{Sec}(A, v, y)$ to $\mathrm{Sec}(A, v, y + h)$. We also use $H$ as the length between $\mathrm{Sec}(A, v, y)$ and $P$.

### I.4   One-side Decay Rate of Linear Section Maps

Our subgoal is to bound the one-side decay rate of the section density, and we use linear section maps to achieve this. Previously, we dealt with expanding or equal sections; hence section maps directly gave the lower bound of the density. However, for general linear section maps, they are no longer expanding sections due to boundaries. Assume the density $f$ is defined in $\mathbb{B}_R$ and it has a decay rate $M$. If $f$ has LAC with function $\mathcal{L}(\cdot)$, we already studied in Lemma 3 that it has a decay rate with $M = \sqrt{d}\mathcal{L}(R)$. We can set $\mathcal{L}(R) \geq 10$.

**Slope of linear section maps.**   We first bound the maximum length between $\Phi_y^h(x)$ and $x$. It is related to the slope of the section maps, and we define $s > 0$ such that

$$\|\Phi_y^h(x) - x\|_2 \leq s\,h$$

for any $x \in \mathrm{Sec}(A, v, y)$. We call $s$ the slope of the linear section map $\Phi_y^h(\cdot)$.

**Volume element.**   We define $|\det(\nabla\Phi_y^h(x))| := u_y(h)$. This value is the same for any $x \in \mathrm{Sec}(A, v, y)$, since we use linear section maps. Using that, we can calculate the lower bound of $g(y + h)$ as follows:

$$
\begin{aligned}
g(y + h) &= \int_{\mathrm{Sec}(A,v,y+h)} f(x)\,\mathrm{d}x \\
&\geq \int_{\mathrm{Sec}(A,v,y)} f\left(\Phi_y^h(x)\right) \left|\det\left(\nabla\Phi_y^h(x)\right)\right|\mathrm{d}x \\
&\geq \int_{\mathrm{Sec}(A,v,y)} f(x)\exp(-M\,s\,h)\,u_y(h)\,\mathrm{d}x \\
&= \exp(-M\,s\,h)\,u_y(h)\,g(y).
\end{aligned}
$$

The third inequality uses the Gronwall inequality, Lemma 22, applying that the decay rate is bounded by $M$ and the length between $x$ and $\Phi_y^h(x)$ is bounded by $s\,h$. This formula tells us that to bound the one-side decay rate of the section density, there are two quantities: slope $s$ and volume element $u_y(h)$.

**A way to bound** $u_y(h)$**.** Let us define $H$ as the distance between $\mathrm{Sec}(A, v, y)$ and $P$. The volume element can be calculated as

$$u_y(h) = \left(\frac{H - h}{H}\right)^d,$$

since $\frac{H-h}{H}$ is the ratio of similarity. Then,

$$\frac{g(y + h)}{g(y)} \geq \exp\left(-Msh - \left[-d\log\left(\frac{H - h}{H}\right)\right]\right).$$

When $h \leq \frac{H}{2}$, we have $\log\left(\frac{H-h}{H}\right) \geq 1 - \frac{2h}{H}$, so

$$\frac{g(y + h)}{g(y)} \geq \exp\left(-Msh - \frac{2d}{H}h\right).$$

This implies that when we can find the linear section map $\Phi_y^h$, we can bound the one-side decay rate of the section density by $Ms + \frac{2d}{H}$.

## I.5 Linear Section Maps between Sliced Balls, $\mathbb{S}_R(v, y)$

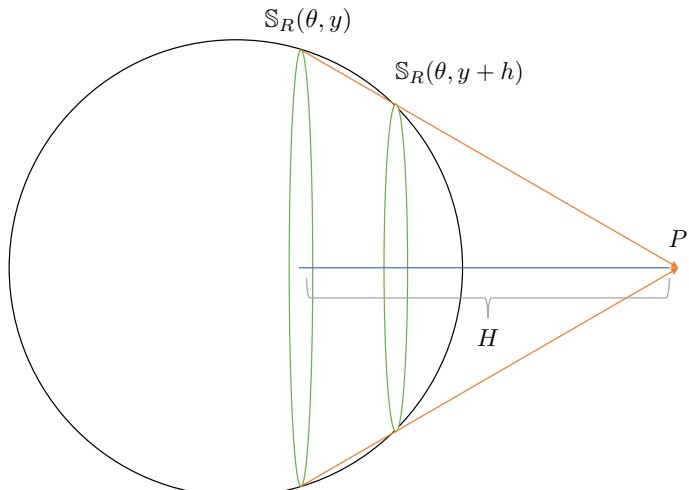

Figure 14: Illustration of the linear section map between two sections of the ball. The linear section map is constructed with the center $P$.

**Lemma 16 (One-side decay rate of $\mathbb{S}_R(v, y)$)** *The projected density $g(y)$ of sections $\mathbb{S}_R(v, y)$ satisfies*

$$\frac{g(y')}{g(y)} \geq \exp\left(-3\sqrt{d}\,\mathcal{L}(R)\,(y' - y)\right)$$

*for any* $-\frac{1}{\sqrt{d}} < y < y' < \frac{1}{\sqrt{d}}$.

**Proof** Let $y' = y + h \in [-\frac{1}{\sqrt{d}}, \frac{1}{\sqrt{d}}]$. Set $P$ as the similarity center of $\mathbb{S}_R(v, y)$ and $\mathbb{S}_R(v, y + h)$. We aim to apply the arguments of Appendix I.4. Using this $P$, we can form section maps $\Phi_y^h$. First, the slope of $\Phi_y^h$ is bounded by $s \leq \frac{R}{\sqrt{R^2 - (y+h)^2}} \leq 2$ for $|y| \leq \frac{1}{\sqrt{d}}$. The volume element of $\Phi_y^h$

can be interpreted as the ratio of similarity. Let $H$ be the distance of $P$ and $\mathbb{S}_R(v, y)$. Note that $H \geq \frac{R^2 - (y+h)^2}{y+h}$ and hence $H \geq 2h$ holds. By applying the result in Appendix I.4, its one-side decay rate in $\left[ -\frac{1}{\sqrt{d}}, \frac{1}{\sqrt{d}} \right]$ is bounded by

$$2\sqrt{d}\,\mathcal{L}(R) + \frac{2d}{H} \ \leq \ 2\sqrt{d}\,\mathcal{L}(R) \ + \ \frac{2d\,(y+h)}{R^2 - (y+h)^2} \ \leq \ 3\sqrt{d}\,\mathcal{L}(R).$$

This holds for all $-\frac{1}{\sqrt{d}} \leq y < y + h \leq \frac{1}{\sqrt{d}}$, since $\mathcal{L}(R) \geq 10$. ∎

## I.6  Linear Section Maps for Double Sliced Balls, $\mathbb{S}_R(\theta, b, v, y)$

**Goal.**  We investigate the one-sided decay rate of section densities with sections $\mathbb{S}_R(\theta, b, v, y)$ for fixed $\theta, v, b$. For density $f(x)$ defined in $\mathbb{B}_R$ with LAC function $\mathcal{L}(\cdot)$, we define the section density of $\mathbb{S}_R(\theta, b, v, y)$ as $g(y)$. We set $\mathbf{M} = 4\sqrt{d}\,\mathcal{L}(R) + \frac{16\sqrt{d}}{R^2(1 - c_\star)}$. We prove that the one-side decay rate of $g$ at $y \in \left[ -\frac{1}{\mathbf{M}}, \frac{1}{\mathbf{M}} \right]$ is bounded by $\mathbf{M}$. Also, we define a small angle $\tau_0 := \sin^{-1}\left( \frac{1}{\mathbf{M}R} \right)$.

### I.6.1  Case $v \perp \theta$.

For this case, the sections $\mathbb{S}_R(\theta, b, v, y)$ are no longer expanding sections when $\langle \theta, v \rangle$ is close to $\frac{\pi}{2}$. In that case, we can construct a linear section map, and the following figure illustrates the procedure. In each section $\mathbb{S}_R(\theta, b, v, y)$ and $\mathbb{S}_R(\theta, b, v, y + h)$, assume the highest coordinate with respect to direction $\theta$ is $Q$ and $Q_h$. Assume the directed half-line starting from connecting $Q$ to $Q_h$ and the hyperplane $\{x \mid x^\top \theta = b\}$ meets at some point $P$. Then we can set the intersection point $P$ as the center of the projection map and we can construct the linear section map using $P$. Please see Figure 15 for the intuition.

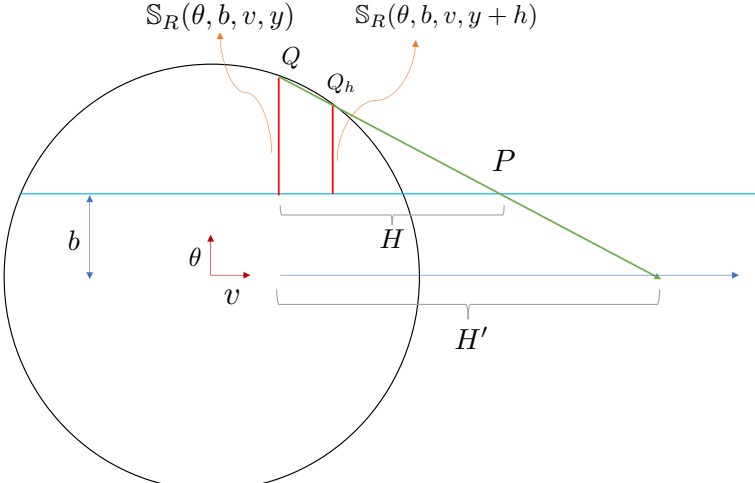

Figure 15: Illustration of section maps between $\mathbb{S}_R(\theta, b, v, y)$ and $\mathbb{S}_R(\theta, b, v, y + h)$. We connect two points $Q$ and $Q_h$. Let $P$ be the intersection with $\{x \mid x^\top \theta = b\}$. Then, using $P$, we can set the section maps between $\mathbb{S}_R(\theta, b, v, y)$ and $\mathbb{S}_R(\theta, b, v, y + h)$.

**Lemma 17 (One-side decay rate of double sliced ball)**  *Suppose $b \leq \alpha R$ for some $\alpha < 1$. If $v \perp \theta$, suppose we construct a linear section map by the sliced balls between $\mathbb{S}_R(\theta, b, v, y)$ and $\mathbb{S}_R(\theta, b, v, y + h)$ for $b \leq \alpha R$ in the way described above. By setting $M = 2\sqrt{d}\,\mathcal{L}(R) + \frac{8\sqrt{d}}{R^2(1-\alpha)}$, the one-side decay rate of the section density $g(\cdot)$ in the interval $-\frac{1}{M} < y < y + h < \frac{1}{M}$ is bounded by*

$$\frac{g(y + h)}{g(y)} \geq \exp(-M\,h).$$

**Proof** We aim to estimate $s$ and $H$ to apply the arguments in Appendix I.4. To bound $s$, we can easily see that $s \leq \frac{R}{\sqrt{R^2-(y+h)^2}} \leq 2$, since it is related to the slope of the tangent line at $y + h$.

To bound $H$, with some elementary calculations, we have

$$H = H' \times \left( \frac{\sqrt{R^2 - y^2} - c_\star R}{\sqrt{R^2 - y^2}} \right)$$

for

$$H' \geq \frac{R^2 - (y+h)^2}{y+h}.$$

$H'$ is defined in Figure 15. Then we see

$$
\begin{aligned}
\frac{d}{H} &\leq d \cdot \frac{y+h}{R^2 - (y+h)^2} \left( \frac{\sqrt{R^2 - y^2}}{\sqrt{R^2 - y^2} - \alpha R} \right) \\
&\leq \frac{2\,d\,y}{R^2 - y^2} \cdot \frac{2}{1-\alpha}.
\end{aligned}
$$

for any $y, y + h \in \left[ -\frac{1}{M}, \frac{1}{M} \right]$. Then,

$$2\sqrt{d}\,\mathcal{L}(R) + \frac{2d}{H} \lesssim 2\sqrt{d}\,\mathcal{L}(R) + \frac{8\,d\,y}{R^2(1-\alpha)} \leq M.$$

The remaining part is proved by the argument from Appendix I.4. ∎

### I.6.2   Case $\frac{\pi}{2} - \tau_0 \leq \angle(v, \theta) \leq \frac{\pi}{2}$

In this case, we define $\theta' := \frac{\theta - v(v^\top \theta)}{\|\theta - v(v^\top \theta)\|_2}$ and we aim to define section maps between $\mathbb{S}_R(\theta, b, v, y)$ using $v$ and $\theta'$. For new sections, $\mathbb{S}_R(\theta', b, v, y)$, we can make section maps as described in Lemma 17. We can see that these section maps also become section maps between $\mathbb{S}_R(\theta, b, v, y)$. By using Lemma 17 for $\alpha$ with $1 - \alpha = \frac{1-c_\star}{2}$ and $\theta', v$, we have

$$\frac{g(y+h)}{g(y)} \geq \exp(-\mathbf{M}\,h),$$

holding for $\mathbf{M} = 2\sqrt{d}\,\mathcal{L}(R) + \frac{16}{R^2(1-c^\star)}$ and any $-\frac{1}{\mathbf{M}} < y < y + h < \frac{1}{\mathbf{M}}$.

### I.6.3   Case $0 \leq \angle(v, \theta) \leq \frac{\pi}{2} - \tau_0$

In this case, the natural section map between $\mathbb{S}_R(v, y)$ and $\mathbb{S}_R(v, y+h)$ for any $-\frac{1}{\mathbf{M}} < y < y + h < \frac{1}{\mathbf{M}}$ can be an expanding section map for $y, y + h \in \left[ -\frac{1}{\mathbf{M}}, \frac{1}{\mathbf{M}} \right]$. We already proved that the one-side decay rate of the section density is bounded by

$$\frac{g(y+h)}{g(y)} \geq \exp(-\mathbf{M}\,h)$$

using Lemma 16.

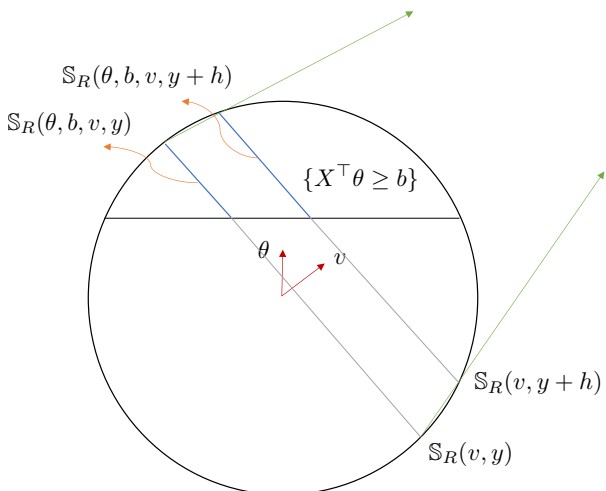

Figure 16: For the case where $\theta$ and $v$ do not form too large an angle, we first define a linear section map between $\mathbb{S}(\theta, y)$ and $\mathbb{S}(\theta, y + h)$. This section map can also be the linear section map between $\mathbb{S}(\theta, b, v, y)$ and $\mathbb{S}(\theta, b, v, y + h)$. Using this map, we can bound the one-side decay rate of section densities of $\mathbb{S}(\theta, b, v, y)$ by Lemma 16.

## I.7 Proof of Proposition 4

We first fix $\theta$ and $v$ with the same argument as in the proof of Theorem 2. For fixed history contexts $\mathbf{X} = (X_1^\top, \ldots, X_K^\top)$ and fixed $\theta$ and $v$, define $b$ as the second largest value of $(X_1^\top \theta, \ldots, X_K^\top \theta)$. By Condition 1, we first bound

$$
\begin{aligned}
\mathbb{E}[v^\top X_a X_a^\top v] &= \mathbb{P}[b \le c_\star R] \, \mathbb{E}[v^\top X_a X_a^\top v \mid b \le c_\star R] + \mathbb{P}[b > c_\star R] \, \mathbb{E}[v^\top X_a X_a^\top v \mid b > c_\star R] \\
&\ge p_\star \, \mathbb{E}[v^\top X_a X_a^\top v \mid b \le c_\star R] \\
&\ge p_\star \, \mathbb{E}\Big[\mathbb{E}\big[v^\top X_a X_a^\top v \mid \{b \le c_\star R\} \cap \Omega_i(\{x_j\}_{j \ne i})\big]\Big] \\
&\ge p_\star \, \mathbb{E}\Big[\mathbb{E}\big[v^\top X_i X_i^\top v \mid \{b \le c_\star R\} \cap \Omega_i(\{x_j\}_{j \ne i})\big]\Big].
\end{aligned}
$$

Then, we aim to bound

$$
\mathbb{E}[v^\top X_i X_i^\top v \mid \{b \le c_\star R\} \cap \Omega_i(\{x_j\}_{j \ne i})].
$$

Further observe the density of

$$
X_i^\top v = y \,\Big|\, \Omega_i(\{x_j\}_{j \ne i}) \cap \{b \le c_\star R\}.
$$

It is a section density of sections $\mathbb{S}_R(\theta, b, v, y)$.

**Case 1:** $\frac{\pi}{2} - \tau_0 \le \angle(\theta, v) \le \frac{\pi}{2}$. By applying results from Appendix I.6 and Lemma 4, we have

$$
\mathbb{E}[v^\top X_i X_i^\top v \mid \{b \le c_\star R\} \cap \Omega_i(x_j)] \ge c \, \frac{1}{\mathbf{M}^2}.
$$

**Case 2:** $0 \le \angle(\theta, v) \le \frac{\pi}{2} - \tau_0$. By applying results from Appendix I.6 and Lemma 4, we have

$$
\mathbb{E}[v^\top X_i X_i^\top v \mid \{b \le c_\star R\} \cap \Omega_i(x_j)] \ge c \, \frac{1}{\mathbf{M}^2}.
$$

**Case 3:** $\angle(\theta, v) \ge \frac{\pi}{2}$. When we replace $v$ with $-v$, it falls into Case 1 or 2.

$\blacksquare$

## I.8   Proof of Proposition 5

**Proof**  Using a similar argument, we use the fixed history arguments. Write $\mathbf{X} = (X_1, \ldots, X_K) = \mathbf{X}(t) \mid \mathcal{H}_{t-1}$ and we first fix $\theta$ and consider the greedy policy $a$ with estimator $\theta$. We define $\lambda_\star$ as a diversity constant of $\mathbf{X}$. Also, we define $L_n := c_0 \sqrt{d}\, x_{\max}\big(1 + \log dK + n \log \gamma\big)$ and set the event $\mathbf{B}_n := \{X_i \in \mathbb{B}(0, L_n) \text{ for all } i \in [K]\}$. We determine $\gamma > 0$ later. By Lemma 19, we have $\mathbb{P}[\mathbf{B}_n] \geq 1 - \frac{1}{\gamma^n}$. We apply the peeling technique to bound the truncated contexts:

$$\mathbb{E}[X_a X_a^\top] = \mathbb{E}[X_a X_a^\top;\, \mathbf{B}_1] + \sum_{n=1}^{\infty} \mathbb{E}[X_a X_a^\top;\, \mathbf{B}_{n+1} \setminus \mathbf{B}_n].$$

By setting $\alpha = c_0 \sqrt{d}\, x_{\max}$, $\beta = 1 + \log dK$, observe that

$$\sum_{n=1}^{\infty} \mathbb{E}[X_a X_a^\top;\, \mathbf{B}_{n+1} \setminus \mathbf{B}_n] \preceq \sum_{n=1}^{\infty} L_{n+1}^2 \frac{1}{\gamma^n}\, I_d$$

$$\preceq \sum_{n=1}^{\infty} \Big(\alpha\big(\beta + n \log \gamma\big)\Big)^2 \frac{1}{\gamma^n}\, I_d$$

$$\preceq \sum_{n=1}^{\infty} 2\alpha^2 \big(\beta^2 + n^2 \log^2 \gamma\big) \frac{1}{\gamma^n}\, I_d.$$

If $\gamma \geq 3\alpha^2 \beta^2 \frac{1}{\lambda_\star}$, we have

$$\sum_{n=1}^{\infty} \mathbb{E}[X_a X_a^\top;\, \mathbf{B}_{n+1} \setminus \mathbf{B}_n] \preceq \frac{1}{2} \lambda_\star I_d.$$

Using Theorem 2, we get

$$\lambda_\star I_d \;\leq\; \mathbb{E}[X_a X_a^\top;\, \mathbf{B}_1] \;+\; \frac{1}{2} \lambda_\star I_d,$$

and hence

$$\mathbb{E}[X_a X_a^\top;\, \mathbf{B}_1] \;\geq\; \frac{1}{2} \lambda_\star I_d.$$

Finally, we get

$$\mathbb{E}\big[X_a X_a^\top \mid \mathbf{B}_1\big] \;\geq\; \frac{1}{2} \lambda_\star.$$

∎

## I.9   Proof of Proposition 6

Define $\Delta(\overline{\mathbf{X}})$ as the suboptimality gap of $\overline{\mathbf{X}}$ and denote its density as $f_{\overline{\mathbf{X}}}(x)$, $x \in (\mathbb{R}^d)^K$. By direct expectation, we get

$$\mathbb{P}\big[\Delta(\overline{\mathbf{X}}) \leq \varepsilon\big] = \int_{\Delta(\overline{\mathbf{X}}) \leq \varepsilon} f_{\overline{\mathbf{X}}}(x)\, dx$$

$$= \int_{\{\Delta(\overline{\mathbf{X}}) \leq \varepsilon\} \cap D} \frac{f_{\mathbf{X}}(x)}{\mathbb{P}[\mathbf{X} \in D]}\, dx$$

$$\leq \int_{\{\Delta(\mathbf{X}) \leq \varepsilon\}} \frac{f_{\mathbf{X}}(x)}{\mathbb{P}[\mathbf{X} \in D]}\, dx$$

$$\leq \frac{1}{1 - \delta} \mathbb{P}\big[\Delta(\mathbf{X}) \leq \varepsilon\big],$$

which holds.

## I.10 Proof of Proposition 7

By conditioning on $\mathcal{H}_{t-1}$, we define fixed history contexts $\mathbf{X} = (X_1^\top, \ldots, X_K^\top)$, similar to previous proofs in Appendix G. We decompose the probability of the suboptimality gap as

$$
\mathbb{P}\big[\Delta(\mathbf{X}) \le \varepsilon\big] = \sum_{i=1}^{K} \mathbb{P}\big[\{\Delta(\mathbf{X}) \le \varepsilon\} \cap \Omega_i^\star\big]
$$

$$
= \sum_{i=1}^{K} \mathbb{E}\bigg[\mathbb{P}\big[\{\Delta(\mathbf{X}) \le \varepsilon\} \cap \Omega_i^\star \,\big|\, \{X_j = x_j\}_{j \ne i}\big]\bigg]
$$

$$
= \sum_{i=1}^{K} \mathbb{E}\bigg[\mathbb{P}\big[\{\max_{j \ne i} x_j^\top \theta^\star \le X_i^\top \theta^\star \le \max_{j \ne i} x_j^\top \theta^\star + \varepsilon\} \,\big|\, \{X_j = x_j\}_{j \ne i}\big]\bigg],
$$

the last equality holds by the definition of $\Omega_i^\star$. Next, we aim to bound

$$
\mathbb{P}\big[\{\max_{j \ne i} x_j^\top \theta^\star \le X_i^\top \theta^\star \le \max_{j \ne i} x_j^\top \theta^\star + \varepsilon\} \,\big|\, \{X_j = x_j\}_{j \ne i}\big].
$$

Similarly, we only need to bound the maximum density of

$$
\mathbb{P}\big[X_i^\top \theta^\star = y \,\big|\, \{X_j = x_j\}_{j \ne i}\big].
$$

and it is enough to bound its one-side decay rate by Lemma 5. Since the conditional density of

$$
X_i \,\big|\, \{X_j = x_j\}_{j \ne i},
$$

say $f_3(\cdot)$, has the same LAC with constant function $\mathcal{L}(R)$, it thus has a bounded decay rate $\sqrt{d}\,\mathcal{L}(R)$ by Lemma 3. Also, the density $\mathbb{P}[X_i^\top \theta^\star = y \,\big|\, \{X_j = x_j\}_{j \ne i}]$ is a section density of $\mathbb{S}_R(\theta^\star, y)$ for $y \in [-R, R]$. Then we observe that in at least one of the directions of $\theta^\star$ or $-\theta^\star$, the sections $\mathbb{S}_R(\theta^\star, y)$ are expanding sections in $\mathbb{B}_R$. By applying Lemma 8, we can bound the one-side decay rate of the section density $X_i^\top \theta^\star \,\big|\, \{X_j(t) = x_j\}_{j \ne i}$ by $\sqrt{d}\,\mathcal{L}(R)$, and finally we can bound the maximum density by $3\sqrt{d}\,\mathcal{L}(R)$ using Lemma 5. Therefore, by the decomposition above, we finally get

$$
\mathbb{P}\big[\Delta(\mathbf{X}(t)) \le \varepsilon\big] \le \sum_{i \in [K]} 3\sqrt{d}\,\mathcal{L}(R)
$$

$$
\le 3K\,\sqrt{d}\,\mathcal{L}(R).
$$

## I.11 Proof of Corollary 1

We have $\lambda_\star(t) \ge c\,\frac{p}{d}\,\frac{1}{\left(\mathcal{L}(R) + \frac{1}{R(1 - c_\star)}\right)^2}$ with $1 - c_\star = \frac{R'}{r} \asymp 1$ by using Proposition 4 and Lemma 14. Hence, we get $\lambda_\star(t) \ge c\,\frac{p}{d\,\mathcal{L}(R)^2} := \lambda_\star$. Also, using Proposition 6, we have the margin constant of truncated contexts $\bar{C}_\Delta$ bounded by $\frac{1}{1-p}\,C_\Delta$, where $C_\Delta$ is defined in Theorem 3. We can see that

$$
\mathbf{Reg}(T) \le c\,\sigma^2\,d\,R^2\,C_\Delta\,\frac{1}{\lambda_\star}\,(\log(T))^2
$$

$$
\le \widetilde{\mathcal{O}}\big(d^{2.5}\,R^2\,\mathcal{L}(R)^2\,\frac{1}{p}\big)
$$

holds.

■

## I.12 Proof of Corollary 2

If $L \ge \sqrt{d}\,c_0\,x_{\max}(3 + \log dK)$, we can use the same proof as in Theorem 2. For the diversity constant of truncated contexts, it is lower bounded by $\frac{1}{2}\,\lambda_\star(t)$, where $\lambda_\star(t)$ is defined in Theorem 2. Also, using Proposition 6, we have the margin constant of truncated contexts $\bar{C}_\Delta$ bounded by $\frac{1}{1-p}\,C_\Delta$, where $C_\Delta$ is defined in Theorem 3. Combining these observations, we can finally apply Proposition 9 since truncated contexts also have bounded $\psi_1$-norm by $x_{\max}$, and we get the desired result.

■

## I.13  Proof of Corollary 3

We can prove it directly by combining the results of Proposition 4 and Proposition 7. This can be obtained directly by combining those results with Proposition 8.

∎

# J  Analysis After Challenges 1 and 2 are Satisfied

We now present the results and proofs to obtain an exact regret bound after the two challenges are addressed. Recall that we define the (unexpected) regret as $\mathrm{reg}'(t) := X_{a^\star(t)}(t)^\top \theta^\star - X_{a(t)}(t)^\top \theta^\star$ and the expected regret as $\mathrm{reg}(t) = \mathbb{E}_{\mathcal{H}_{t-1}, \mathbf{X}(t)}[\mathrm{reg}'(t)]$. We can achieve logarithmic regret bounds if the contexts meet Challenges 1 and 2.

We first present our regret analysis for bounded contexts.

**Proposition 8 (Regret analysis: bounded contexts)** *For bounded contexts where $\|X_i(t)\|_2 \leq R$, suppose that $\mathbb{E}[X_{a(t)} X_{a(t)} \mid \mathcal{H}_{t-1}] \succeq \lambda_\star I$ and $C_{\Delta(\mathbf{X}(t))} \leq C_\Delta$ for all $t \in [T]$ and $\mathcal{H}_{t-1}$. Under Assumption 1, the expected regret of Algorithm 1 is bounded by*

$$\mathbf{Reg}(T) \leq c\sigma^2 R^2 d C_\Delta \frac{1}{\lambda_\star}(\log T)^2 = \widetilde{\mathcal{O}}(R^2 d \frac{C_\Delta}{\lambda_\star}).$$

Next, we present the regret bound result for unbounded contexts, under the satisfaction of the two challenges.

**Proposition 9 (Regret analysis: unbounded contexts)** *For unbounded contexts where $\|X_i(t)\|_{\psi_1} \leq x_{\max}$, suppose that $\mathbb{E}[X_{a(t)} X_{a(t)} \mid \mathcal{H}_{t-1}] \succeq \lambda_\star I$ and $C_{\Delta(\mathbf{X}(t))} \leq C_\Delta$ for all $t \in [T]$ and $\mathcal{H}_{t-1}$. Under Assumption 1, the expected regret of Algorithm 1 is bounded by*

$$\mathbf{Reg}(T) \leq c\sigma^2 x_{\max}^2 d C_\Delta \frac{1}{\lambda_\star}(\log T)^4 = \widetilde{\mathcal{O}}(x_{\max}^2 d \frac{C_\Delta}{\lambda_\star}).$$

## J.1  Proof of Proposition 8

Firstly, set $T_0 := \frac{1}{c_1 \lambda_\star} R(3 \log T + \log d)$ for the constant $c_1$ defined in Corollary 9. Before starting the proof, we define the *good events* that satisfy the sufficient concentration of the estimator $\hat{\theta}_t$.

**Definition 17** *From now on, in this section, we define the event $E_t$ by*

$$E_t := \{\lambda_{\min}(\Sigma(t)) \geq \frac{\lambda_\star}{4}t\}.$$

*This is the event that the Gram matrix has sufficiently large minimum eigenvalue. The event $E_t$ holds with high probability according to Corollary 9.*

**Corollary 4** *For $t \geq T_0$, the following holds:*

$$\mathbb{P}[E_t] \geq \frac{1}{2T^2}.$$

**Proof**  Using Corollary 9, with probability $1 - \frac{1}{2T^2}$, $E_t$ holds. ∎

**Definition 18 (Self-normalized bound of OLS estimator)** *Next, we define the event $F_t$ as*

$$F_t := \left\{ \|\hat{\theta}_t - \theta^\star\|_{\Sigma(t)} \leq 2\sigma\sqrt{d\log(T(1+tR^2))} + 1 \right\},$$

*which satisfies the self-normalized concentration of the estimator.*

**Lemma 18 (Concentration of OLS Estimator)** *For any $t > \frac{4}{\lambda_\star}$,*

$$\mathbb{P}[F_t \cap E_t] \geq 1 - \frac{1}{T^2}$$

*holds.*

**Proof** Under $E_t$, we have $\Sigma(t) \succeq \frac{1}{4}\lambda_\star t I_d$. Then the concentration of the OLS estimator satisfies

$$
\begin{aligned}
\|\hat{\theta}_t - \theta^\star\|_{\Sigma(t)} &= \big(\sum_{\tau=1}^t X_{a(\tau)}(\tau)\eta_\tau\big)(\Sigma(t))^{-1}\big(\sum_{\tau=1}^t X_{a(\tau)}(\tau)\eta_\tau\big) \\
&\leq \big(\sum_{\tau=1}^t X_{a(\tau)}(\tau)\eta_\tau\big)\big(\frac{1}{2}\Sigma(t) + I_d\big)^{-1}\big(\sum_{\tau=1}^t X_{a(\tau)}(\tau)\eta_\tau\big) \\
&\leq 2\underbrace{\big(\sum_{\tau=1}^t X_{a(\tau)}(\tau)\eta_\tau\big)(\Sigma(t) + I_d)^{-1}\big(\sum_{\tau=1}^t X_{a(\tau)}(\tau)\eta_\tau\big)}_{I}.
\end{aligned}
$$

To bound $I$, we use Lemma 24 from Abbasi-Yadkori et al. [1] and obtain the desired result. ∎

Next, we introduce an important lemma used to obtain the concentration of the minimum eigenvalue of the Gram matrix. We define $\mathbf{G}_t := \bigcap_{\tau=T_0}^t E_\tau \cap F_\tau$.

**Corollary 5 (Good events)** *For the event $\mathbf{G}_t$ defined above, for any $T_0 \leq t \leq T$,*

$$\mathbb{P}[\mathbf{G}_t] \geq 1 - \frac{1}{T}$$

*holds.*

**Proof** Straightforward from the previous observations. ∎

Then, under the event $\mathbf{G}_t$, we have the following corollary by the definition of event $E_t$ and $F_t$:

**Corollary 6 ($\ell_2$ concentration: bounded contexts)** *For any $t \geq T_0$, under the event $\mathbf{G}_t$, we have*

$$\|\hat{\theta}_t - \theta^\star\|_2 \leq c\sigma\frac{\sqrt{d\log T}}{\sqrt{\lambda_\star t}}$$

*for some absolute constant $c > 0$.*

**Proof of the Proposition.** We are now ready to prove Proposition 8. First, for time $t \geq T_0$, the history $\mathcal{H}_{t-1}$ is contained in $\mathbf{G}_{t-1}$ with probability $1 - \frac{1}{T}$. The expectation of $\mathrm{reg}'(t)$ is calculated with respect to the randomness of the whole history $\mathcal{H}_{t-1}$ and the distribution of contexts $\mathbf{X}(t)$. We can observe the following for $\gamma(d) := 4\sigma\sqrt{d\log(T + T^2R^2)}$:

$$
\begin{aligned}
\mathbb{E}[\mathrm{reg}'(t)] &= \mathbb{E}_{\mathcal{H}_{t-1}}\big[\mathbb{E}_{\mathbf{X}(t)}[\mathrm{reg}'(t) \mid \mathcal{H}_{t-1}]\big] \\
&= \mathbb{E}_{\mathbf{X}(t)}\big[\mathrm{reg}'(t) \mid \mathbf{G}_{t-1}\big]\mathbb{P}[\mathbf{G}_{t-1}] + R\mathbb{P}[\mathbf{G}_{t-1}^c] \\
&\leq 6C_\Delta\gamma(d)^2\frac{R^2}{(t-1)\lambda_\star} + R\mathbb{P}[\mathbf{G}_{t-1}^c] \quad \text{(by Lemma 7)} \\
&\leq 6C_\Delta\gamma(d)^2\frac{R^2}{(t-1)\lambda_\star} + \frac{R}{T}.
\end{aligned}
$$

By summing these inequalities until $T$, we obtain the desired result. ∎

## J.2 Concentration of Sub-exponential Contexts

We now provide regret analysis for unbounded contexts. To do that, we first provide some known facts for sub-exponential vectors. Under Assumption 2, $X_i(t)$ has $\psi_1$ norm with $x_{\max}$. Using a result from Wainwright [38], for any $v \in \mathbb{S}^{d-1}$, there exists an absolute constant $c_0 > 0$ such that

$$\mathbb{P}[|X_i(t)^\top v| > c_0 x_{\max}(1+u)] \leq \exp(-u) \tag{14}$$

holds for all $u > 0$. If we set $v = e_i = (0, 0, \dots, 1, \dots, 0)$, then we get the concentration for each coordinate,

$$\mathbb{P}[|X_{ij}(t)| > c_0 x_{\max}(1+u)] \leq \exp(-u).$$

First, we investigate the concentration of the $\ell_2$ norm of the contexts.

**Lemma 19 (High probability $\ell_2$ bound of the contexts)** *Suppose the contexts $\mathbf{X}(t)$ satisfy Assumption 2, then*

$$\max_{i \in [K]} \|X_i(t)\|_2 \leq c_0\sqrt{d}x_{\max}(1+\log(dK\frac{1}{\delta}))$$

*holds with probability at least $1 - \delta$.*

**Proof** For any $v \in \mathbb{S}^{d-1}$, we know

$$\mathbb{P}[|X_i(t)^\top v| \geq c_0 x_{\max}(1+u)] \leq \exp(-u)$$

by the result from Appendix J.2. Then, we get

$$\mathbb{P}\Big[\text{there exists } i \in [K] \text{ such that } \|X_i(t)\|_2 > \sqrt{d}\,\frac{1}{c_0}x_{\max}\big(1+\log(dK\frac{1}{\delta})\big)\Big]$$

$$\leq \sum_{i=1}^{K}\sum_{j=1}^{d} \mathbb{P}\Big[|X_{ij}(t)| > \frac{1}{c_0}x_{\max}\big(1+\log(dK\frac{1}{\delta})\big)\Big]$$

$$\leq dK \times \frac{\delta}{dK} = \delta.$$

$\blacksquare$

## J.3 Proof of Proposition 9

We now prove Proposition 9. It consists of three steps. First, we investigate the concentration of the Gram matrix for unbounded contexts. Next, we define several high-probability good events. Finally, we bound the regret using the peeling technique.

### J.3.1 Gram Matrix Concentration

For bounded contexts, we can apply Lemma 23 and Corollary 9 to ensure the linear growth of the Gram matrix. However, for unbounded contexts, we cannot apply Lemma 23 directly since it requires $\ell_2$ boundedness. We apply the same technique used in Kannan et al. [20], who also deal with Gaussian contexts, which are not bounded. They use a truncation technique to guarantee the growth of the Gram matrix: interpret it as the mixture of truncated contexts and large $\ell_2$ norm contexts. We apply similar arguments here. In this section, we set our truncation radius $L = c\sqrt{d}x_{\max}\big(1+3\log\big(\frac{dKT}{\lambda_\star}\big)\big)$ and define $T_1 := \frac{2}{c_1}\frac{L}{\lambda_\star}(2\log T + \log d)$.

**Lemma 20** *For any $t \geq T_1$, the following holds with probability $1 - \frac{2}{T^2}$:*

$$\Sigma(t) \succeq \frac{1}{8}\lambda_\star t.$$

**Proof** We prove it using a similar argument to Kannan et al. [20] to bound the Gram matrix. We view $X_i(t)$ given the history $\mathcal{H}_{t-1}$ as a mixture of $X_i(t) \mid \mathbb{B}_L$ and $X_i(t) \mid \mathbb{B}_L^c$. Consider an invisible

coin toss $c_s$ for every time $s \in [t]$, and if $c_s = 1$, the contexts are drawn in $\mathbf{X}(t) \mid (\mathbb{B}_L)^K$; if $c_s = 0$, they are drawn from $\mathbf{X}(t) \mid (\mathbb{B}_L^c)^K$. By our choice of high-probability region radius $L$, for any $c_s$, $\mathbb{P}[c_s = 1] \geq 1 - \frac{1}{T^2}$. Define $\overline{\Sigma}(t)$ as the Gram matrix of the contexts sampled from the truncated distribution for all $1 \leq s \leq t$. For truncated contexts, Proposition 5 tells us that it has diversity constant $\frac{1}{2}\lambda_\star$ under our choice of $L$. Then, by our independence assumption and Lemma 5,

$$\mathbb{P}\big[\lambda_{\min}(\Sigma(t)) > \tfrac{1}{8}\lambda_\star t\big] \leq \mathbb{P}\big[c_i = 0 \text{ for some } i \in [t]\big] + \mathbb{P}\big[\lambda_{\min}(\overline{\Sigma}(t)) > \tfrac{1}{8}\lambda_\star t\big]$$

$$\leq \frac{1}{T^3} \times T + d \exp\big(-c_1 \frac{\lambda_\star t}{2L}\big).$$

Hence, if we choose $t \geq T_1$, we get the desired result. ∎

Recall that $L = c\sqrt{d}x_{\max}\big(1 + 3\log\big(dKT\frac{1}{\lambda_\star}\big)\big)$.

### J.3.2   Good Events

We present 4 concentrations and define good events that satisfy the concentrations.

**Concentration 1.**   For any $t \geq 1$, with probability $1 - \frac{1}{T^2}$, $\|X_i(t)\|_2 \leq L$ holds for all $i \in [K]$. Then, under that event,

$$\det(\Sigma(t)) \leq \big(1 + \frac{TL^2}{d}\big)^d$$

and

$$\log\det(\Sigma(t)) \leq d\log\big(1 + \frac{TL^2}{d}\big).$$

**Concentration 2.**   For any $1 \leq t \leq T$, with probability $1 - \frac{1}{T^2}$, using Lemma 24, we get

$$\sqrt{(\sum_{\tau=1}^{t} X_{a(\tau)}(\tau)\eta_\tau)(\Sigma(t) + I_d)^{-1}(\sum_{\tau=1}^{t} X_{a(\tau)}(\tau)\eta_\tau)} \leq \sigma\sqrt{2\log\big(T^2\det(\Sigma(t))^{1/2}\big)}.$$

**Concentration 3.**   In the previous section, we showed that for any $t \geq T_1$,

$$\Sigma(t) \succeq \frac{1}{8}\lambda_\star t$$

holds with probability $1 - \frac{2}{T^2}$.

Combining these three concentrations, we get the following result: with probability $1 - \frac{4}{T^2}$ for any $t \geq T_1$,

$$\|\hat{\theta}_t - \theta^\star\|_{\Sigma(t)} = \sqrt{\big(\sum_{\tau=1}^{t} X_{a(\tau)}(\tau)\eta_\tau\big)\Sigma(t)^{-1}\big(\sum_{\tau=1}^{t} X_{a(\tau)}(\tau)\eta_\tau\big)}$$

$$\leq \sqrt{2\big(\sum_{\tau=1}^{t} X_{a(\tau)}(\tau)\eta_\tau\big)\big(\Sigma(t) + I_d\big)^{-1}\big(\sum_{\tau=1}^{t} X_{a(\tau)}(\tau)\eta_\tau\big)}$$

$$\leq \sigma\sqrt{2\log\big(T^2\det(\Sigma(t))^{1/2}\big)}$$

$$\leq 2\sigma\sqrt{\log\big(\det(\Sigma(t))^{1/2}\big) + \log T}$$

$$\leq 2\sigma\sqrt{\frac{d}{2}\log\big(1 + TL^2\big) + \log T}$$

$$\leq c\sigma\sqrt{d\log T}.$$

The second inequality holds since, under Concentration 3, $\Sigma(t) \succeq \frac{1}{2}(\Sigma(t) + I_d)$. We finally obtain the following $\ell_2$-concentration result:

**Corollary 7 ($\ell_2$ concentration: unbounded contexts)** *For any $t \geq T_1$, we have*

$$\|\hat{\theta}_t - \theta^\star\|_2 \leq c\sigma \frac{1}{\sqrt{\lambda_\star t}} \sqrt{d \log T}$$

*for some absolute constant $c$ with probability $1 - \frac{4}{T^2}$.*

We define event $G_t$ as the event in which Corollary 7 holds and set $\mathbf{G}_t = \bigcup_{\tau=T_1}^{t} G_\tau$.

Finally, we present a key analysis to bound the regret. Recall that we set $T_1 := \frac{1}{c_1} \frac{L \log d}{\lambda_\star} (1 + 2 \log T)$.

**Corollary 8 (Good events)** *For the event $\mathbf{G}_t$ defined above, for any $T_1 \leq t \leq T$,*

$$\mathbb{P}[\mathbf{G}_t] \geq 1 - \frac{4}{T}$$

*holds.*

**Proof** Straightforward by the previous arguments. ∎

### J.3.3 Bounding Regret by the Peeling Technique

**Lemma 21** *For $t \geq T_1$, under the good event $\mathbf{G}_{t-1}$, and if $\|X_i(t)\|_{\psi_1} \leq x_{\max}$, then*

$$\mathbb{E}_{\mathbf{X}(t)}[\mathrm{reg}'(t)] \leq c \, d \, x_{\max}^2 \frac{C_\Delta}{(t-1)\lambda_\star} (\log T)^3$$

*holds for some absolute constant $c > 0$.*

**Proof** Under any history contained in $\mathbf{G}_{t-1}$, by using (14), with probability $1 - \frac{1}{T^2}$ we have

$$\max_{i \in [K]} |X_i(t)^\top (\hat{\theta}_{t-1} - \theta^\star)| \leq c\sigma x_{\max} \frac{\sqrt{d \log T}}{\sqrt{(t-1)\lambda_\star}} (1 + \log KT) := e$$

for some absolute constant $c > 0$. We define this event as $K_t$. Then, the (unexpected) regret is bounded by

$$\mathrm{reg}'(t) \leq 2e.$$

We now bound the *expected* regret:

$$
\begin{aligned}
\mathbb{E}[\mathrm{reg}'(t); \mathbf{G}_{t-1}] &= \mathbb{E}[\mathrm{reg}'(t); \mathbf{G}_{t-1} \cap K_t] + \mathbb{E}[\mathrm{reg}'(t); \mathbf{G}_{t-1} \cap K_t^c] \\
&\leq 2e \, \mathbb{P}[\mathrm{reg}'(t) > 0; \mathbf{G}_{t-1} \cap K_t] + \mathbb{E}[\mathrm{reg}'(t); \mathbf{G}_{t-1} \cap K_t^c] \\
&\leq 2e \, \mathbb{P}[\Delta(\mathbf{X}(t)) \leq 2e] + \mathbb{E}[\mathrm{reg}'(t); \mathbf{G}_{t-1} \cap K_t^c] \\
&\leq 6C_\Delta e^2 + \mathbb{E}[\mathrm{reg}'(t); \mathbf{G}_{t-1} \cap K_t^c].
\end{aligned}
$$

In the last inequality, we use the definition of Challenge 2 and Lemma 7.

Peeling technique for tail events. Next, we bound $\mathbb{E}[\mathrm{reg}'(t) \mid \mathbf{G}_{t-1} \cap K_t^c]$ using the peeling technique. Let $L_n = c_0 \sqrt{d} x_{\max} (1 + \log dK + n \log \gamma)$ for $n \geq 1$, where we will determine $\gamma > 0$ later. Then, by results from Appendix J.2,

$$\mathbb{P}\Big[\max_{i \in [K]} |X_i(t)^\top \theta^\star| \geq L_n\Big] \leq \sum_{i \in [K]} \mathbb{P}\big[|X_i(t)^\top \theta^\star| \geq L_n\big] \leq \frac{1}{\gamma^n}.$$

We define the event $V_n$ as $\{\max_{i\in[K]} |X_i(t)^\top \theta^\star| \in [L_n, L_{n+1}]\}$. Then $\mathbb{P}[V_n] \le \frac{1}{\gamma^n}$ and $\mathbb{E}[\text{reg}'(t); V_n] \le 2L_{n+1}$. Set $\alpha = c_0\sqrt{d}x_{\max}, \beta = 1 + \log dK$. The regret can be decomposed as

$$\mathbb{E}[\text{reg}'(t); \mathbf{G}_{t-1} \cap K_t^c] \le \mathbb{E}[\text{reg}'(t); \mathbf{G}_{t-1} \cap K_t^c \cap V_1] + \sum_{n=1}^\infty \mathbb{E}[\text{reg}'(t); K_t^c \cap (V_{n+1} \setminus V_n)]$$

$$\le 2L_1 \mathbb{P}[\mathbf{G}_{t-1} \cap K_t^c] + \sum_{n=1}^\infty \mathbb{E}[\text{reg}'(t); K_t^c \cap (V_{n+1} \setminus V_n)]$$

$$\le 2L_1 \frac{1}{T} + 2\sum_{n=1}^\infty L_{n+1} \frac{1}{\gamma^n}$$

$$\le 2L_1 \frac{1}{T} + 2\sum_{n=1}^\infty \alpha(\beta + n\log\gamma)\frac{1}{\gamma^n}$$

$$\le 3L_1 \frac{1}{T}$$

when $\gamma \ge (\alpha\beta)^3 + T^2$.

Now set $\gamma = (\alpha\beta)^3 + T^2$. Then we have

$$\mathbb{E}[\text{reg}'(t); \mathbf{G}_{t-1}] \le cC_\Delta \big(\sigma x_{\max} \frac{\sqrt{d\log T}}{\sqrt{(t-1)\lambda_\star}} \log(KT)\big)^2 \le c\sigma^2 C_\Delta d x_{\max}^2 \frac{1}{\lambda_\star(t-1)}(\log T)^3.$$

∎

**Main proof of Proposition 9** We apply the result of Lemma 21. For $t \ge T_1$, we get

$$\mathbb{E}[\text{reg}'(t)] \le cC_\Delta d x_{\max}^2 \frac{1}{\lambda_\star(t-1)}(\log T)^3 + \mathbb{E}[\text{reg}'(t); \mathbf{G}_{t-1}^c]$$

$$\le cC_\Delta d x_{\max}^2 \frac{1}{\lambda_\star(t-1)}(\log T)^3 + \mathbb{E}[\text{reg}'(t); \mathbf{G}_{t-1}^c].$$

Next, we examine $\mathbb{E}[\text{reg}'(t); \mathbf{G}_{t-1}^c]$. We also bound this using the peeling technique. We define the same $L_n$ and $V_n$ as in the previous Lemma 21:

$$\mathbb{E}[\text{reg}'(t); \mathbf{G}_{t-1}^c] \le \mathbb{E}[\text{reg}'(t); V_1 \cap \mathbf{G}_{t-1}^c] + \sum_{n=1}^\infty \mathbb{E}[\text{reg}'(t); (V_{n+1} \setminus V_n) \cap \mathbf{G}_{t-1}^c]$$

$$\le 2L_1 \mathbb{P}[V_1 \cap \mathbf{G}_{t-1}^c] + \sum_{n=1}^\infty \mathbb{E}[\text{reg}'(t); (V_{n+1} \setminus V_n) \cap \mathbf{G}_{t-1}^c]$$

$$\le 2L_1 \frac{4}{T} + 2\sum_{n=1}^\infty L_{n+1} \frac{1}{\gamma^n}$$

$$\le c\frac{L_1}{T}.$$

for some absolute constant $c > 0$. Thus,

$$\mathbb{E}[\text{reg}'(t)] \le cC_\Delta d x_{\max}^2 \frac{1}{\lambda_\star(t-1)}(\log T)^3.$$

By summing up, we get the desired result. For $t \le T_1$, using a peeling technique shows $\mathbb{E}[\text{reg}(t)] \le cL_1$. Hence

$$\mathbf{Reg}(T_1) \le cL_1 T_1 \le c\sigma^2 C_\Delta d x_{\max}^2 \frac{1}{\lambda_\star}(\log T)^4.$$

Then we can get wanted result easily.

∎

# K  Discussion on Discrete-Supported Contexts

In this paper, we considered only context distributions with differentiable densities.

**Single-Parameter Linear Contextual Bandits.**    To the best of our knowledge, no study has addressed the greedy algorithm for linear contextual bandits with discrete-supported stochastic contexts.

**Multiple-Parameter Linear Contextual Bandits.**    For multi-parameter linear contextual bandits, Bastani et al. [8] proved that the Gibbs distribution, which has discrete support, satisfies the margin condition and their diversity condition, achieving logarithmic regret for the greedy algorithm. However, their proof applies only to the two-arm case.

We assert that, for the $K$-armed multi-parameter linear contextual bandit with $K \geq 3$, Gibbs distribution can fail under the greedy policy. For example, the two-dimensional Gibbs distribution has support points $(1, 1), (1, -1), (-1, 1), (-1, -1)$. However, if three parameters are given as $\beta_1^\star = (1, 1), \beta_2^\star = (1, -1), \beta_3^\star = (-1, 1)$, the diversity assumption of Bastani et al. [8] is violated.

We further claim that for multiple-parameter linear contextual bandits, the effectiveness of discrete-supported contextual bandits with an arbitrary number of arms $K$ has not yet been thoroughly studied.

# L  Dicussions, Limitations and Further Ideas

- Since our LAC class primarily includes differentiable densities, examining the performance of the greedy bandit with discrete valued contexts would be a valuable future direction. For discrete valued contexts, the only existing result by Bastani et al. [8] establishes performance for a Gibbs distribution in a 2-arm (shared-context) bandit, but this generally fails when $K \geq 3$ [2] and also it differs somewhat from our setup. In our setup, the linear contextual bandit, no results are currently known for discrete contexts. Given that Bastani et al. [8] studied a shared contexts setup, our work represents the largest class of distributions for which the greedy bandit shows efficient performance in the linear contextual bandit problem.

- To derive the concentration of minimum eigenvalue, the boundedness of contexts (random variables) is required. However, for heavy tail contexts, the upper bound of the contexts' norm can be large, so it leads to poor concentration. Dealing with non-truncated heavy tail contexts can be another interesting problem.

# M  Numerical Experiments

We conducted numerical experiments to evaluate the performance of the greedy algorithm and compare it with existing bandit algorithms, LinUCB from Abbasi-Yadkori et al. [1] and LinTS from Agrawal and Goyal [4]. We conducted experiments for three cases with varying parameters: $d = 20, K = 20, T \leq 1000, d = 100, K = 20, T \leq 1000$, and $d = 20, K = 100, T \leq 1000$, and five different distributions of contexts: Uniform in a ball, truncated Student's t, Laplace, Gaussian, and exponential. The experiments were repeated 10 times for each case, and the deviation was also displayed on the graph.

We note that the results were obtained as $\sqrt{d}$ times larger than the actual size because of the absence of dimensional correction. For the uniform distribution, we used $\mathrm{Unif}(\mathbb{B}(0, \sqrt{d}))$. The Laplace contexts were generated by independently sampling each component of a Laplace distribution with parameters $\mu = 0, b = 1$. The Gaussian context was created so that each element of the feature vectors was drawn from a multivariate Gaussian distribution with a covariance matrix $V$ with $V_{i,i} = 1$ and $V_{i,j} = 0.7$ for any $i \neq j$. The truncated Cauchy contexts were generated by independently sampling each component from a truncated Cauchy with loc 0, scale 1 and truncation range $[-5, 5]$.

In most cases, the greedy algorithm produced the best results. Our theory predicted a polynomial scale dependency on dimension for the regret of the greedy algorithm, and the experimental results

---

[2]Further details on discrete contexts are in Appendix K.

confirmed good performance even with an increased dimension. This discrepancy is due to the fact that we considered the worst case. The experimental results for the three cases are listed. [3]

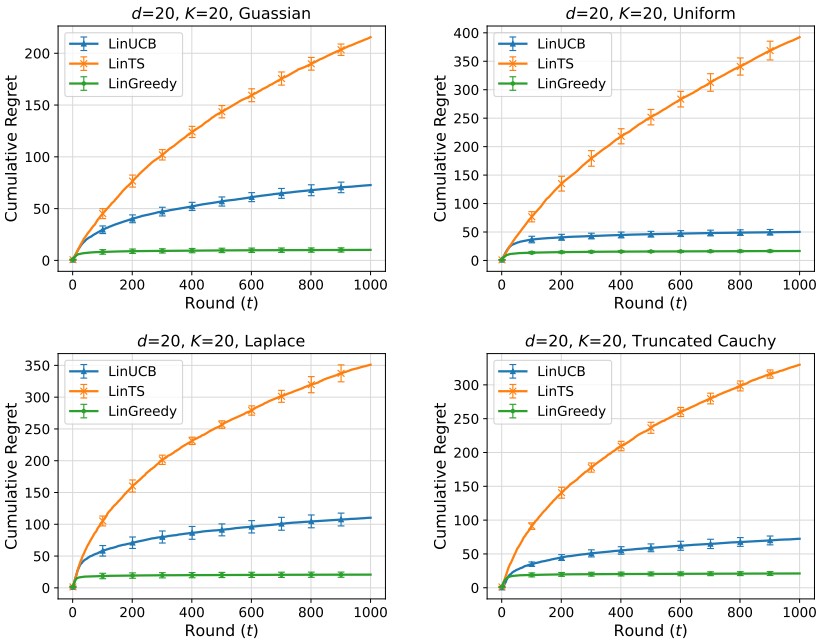

Figure 17: Results For $d = 20, K = 20$

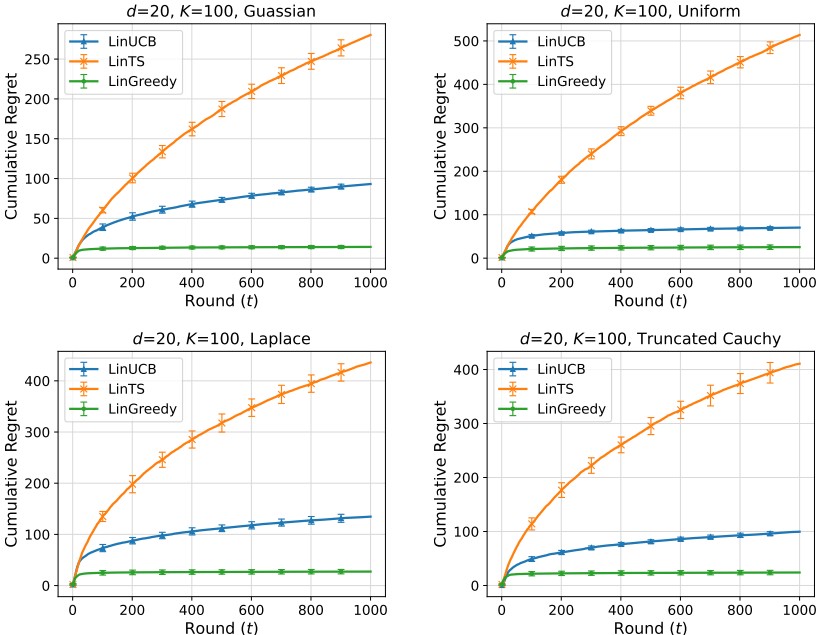

Figure 18: Results For $d = 20, K = 100$

---

[3]We used jupyter notebook to run the experiments.

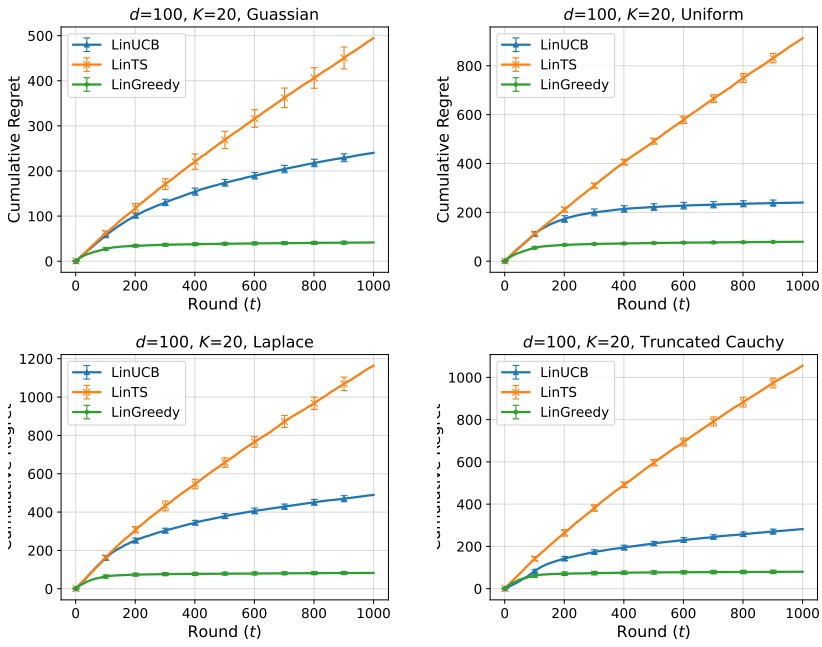

Figure 19: Results For $d = 100, K = 20$

# N  Technical Lemmas

**Lemma 22 (Gronwall Inequality)** *For $g(y) \in \mathbb{R}$ satisfies $\frac{g'}{g}(y) \geq -M$ in $[y, y + h]$, then*

$$\frac{g(y + h)}{g(y)} \geq \exp(-Mh).$$

**Proof**  See classic PDE books like Evans [16].  ∎

## N.1  Concentration Inequalities

**Lemma 23 (Matrix Chernoff : Adapted Sequence from [36])** *Consider a finite adapted sequence $\{X_k\}$ with filtration $\{\mathcal{F}_t\}_{t \geq 0}$ of positive-semi definite matrices with dimension $d$, and suppose that*

$$\lambda_{\max}(X_k) \leq R \quad \text{almost surely.}$$

*Define the finite series*

$$Y := \sum_k X_k \quad and \quad W := \sum_k \mathbb{E}_{k-1} X_k$$

*For all $\mu \geq 0$*

$$\mathbb{P}\{\lambda_{\min}(Y) \leq (1 - \delta)\mu \ \ and \ \ \lambda_{\min}(W) \geq \mu\} \leq d \cdot \left[\frac{e^{-\delta}}{(1 - \delta)^{1-\delta}}\right]^{\mu/R} \quad for \ \delta \in [0, 1)$$

**Corollary 9 (Eigenvalue Growth of Adaptive Gram Matrix)** *If $\quad \|X_i\|_2 \quad \leq \quad x_{\max} \quad$ and $\lambda_{\min}(\mathbb{E}[X_i X_i^\top \mid \mathcal{H}_{t-1}]) \geq \lambda_0$, then with probability $1 - d\exp(-c_1 \frac{\lambda_0 t}{x_{\max}})$*

$$\lambda_{\min}(\sum_{i=1}^t X_i X_i^\top) \geq \frac{\lambda_0}{4} t$$

*holds for some absolute constant $c_1$.*

**Proof** Put $\delta = \frac{3}{4}, \mu = \lambda_0 t, R = x_{\max}$ at the Lemma 23. ∎

**Lemma 24 (Theorem 1 from [1])** *Let $\{F_t\}_{t=0}^{\infty}$ be a filtration. Let $\{\eta_t\}_{t=1}^{\infty}$ be a real-valued stochastic process such that $\eta_t$ is $F_t$-measurable and $\eta_t$ is conditionally $R$-sub-Gaussian for some $R \geq 0$ i.e.*

$$\forall \lambda \in \mathbb{R} \quad \mathbf{E}\left[e^{\lambda \eta_t} \mid F_{t-1}\right] \leq \exp\left(\frac{\lambda^2 R^2}{2}\right)$$

*Let $\{X_t\}_{t=1}^{\infty}$ be an $\mathbb{R}^d$-valued stochastic process such that $X_t$ is $F_{t-1}$-measurable. Assume that $V$ is a $d \times d$ positive definite matrix. For any $t \geq 0$, define*

$$\overline{V}_t = V + \sum_{s=1}^{t} X_s X_s^{\top} \quad S_t = \sum_{s=1}^{t} \eta_s X_s.$$

*Then, for any $\delta > 0$, with probability at least $1 - \delta$, for all $t \geq 0$,*

$$\|S_t\|_{\overline{V}_t^{-1}}^2 \leq 2R^2 \log\left(\frac{\det\left(\overline{V}_t\right)^{1/2} \det(V)^{-1/2}}{\delta}\right)$$

**Lemma 25 (Lemma 10 from [1])** *Suppose $X_1, X_2, \ldots, X_t \in \mathbb{R}^d$ and for any $1 \leq s \leq t$, $\|X_s\|_2 \leq L$. Let $\overline{V}_t = \lambda I + \sum_{s=1}^{t} X_s X_s^{\top}$ for some $\lambda > 0$. Then,*

$$\det\left(\overline{V}_t\right) \leq \left(\lambda + tL^2/d\right)^d.$$

