# OpenReview forum: "Local Anti-Concentration Class: Logarithmic Regret for Greedy Linear Contextual Bandit"
_NeurIPS.cc/2024/Conference — NeurIPS 2024 poster_

### Official Review · Reviewer_Fhbd · 2024-06-26

**Soundness:** 2
**Presentation:** 1
**Contribution:** 2
**Rating:** 6
**Confidence:** 5

**Summary:**

This paper analyzes a Greedy bandit algorithm, in the context of a linear bandit problem where (1) a regression parameter is unknown and fixed for the experiment and (2) the $K$ arms from which the decision-maker can choose are sampled at the beginning of each step $t$, from a fixed context distribution. Recent works have shown that in similar problems exploration is not necessary, and greedy algorithms are sufficient to achieve satisfying theoretical guarantees. This observation is interesting because (1) some standard methods like UCB can be costly in linear bandits, and (2) in some settings exploration might be controversial, since it might imply not acting optimally with the knowledge at hand with consequences on living subjects.

The authors first propose a generic condition on the context distributions, called the LAC condition, under which they analyze Greedy. They then describe the two main statistical challenges needed for the analysis, before presenting poly-logarithmic regret guarantees under bounded context distributions. Finally, they showcase experiments that validate their approach, advocating for the use of Greedy to tackle the linear bandit problem that they consider.

**Strengths:**

First, for the reasons mentioned in previous paragraph, I believe that Greedy algorithms work in some learning settings and understanding their properties is an interesting research question. It is true that, with the development of powerful tools like optimism or Thompson Sampling, we tend to jump on these methods to tackle every bandit problem without necessarily questioning whether their exploration mechanism is necessary or not. Hence, this line of work is in my opinion interesting to recall that simple methods should be tried first before moving to those more elaborate tools.

Then, the authors put a lot of efforts in describing and providing intuitions on the two statistical challenges posed by their problem, namely the positivity of the eigenvalues of the design matrix (i.e. the implicit exploration of greedy), and the fact that randomly sampling contexts guarantee relatively large sub-optimality gaps at each time step. These two challenges are the key ingredients to derive the poly-logarithmic regret guarantees for the greedy algorithm proposed by the authors. Furthermore, the LAC condition is well-described and rather intuitive.

**Weaknesses:**

Edit: after reading the rebuttal and discussing with other reviewers I decided to increase my score
______________________________

In my opinion, the first weakness is the setting itself. Other works in the literature seem to tackle two settings: the one presented in the paper, and another one which actually seems to be the main setting of the other works, in which the arms are fixed but the regression parameter is sampled at each round. This second setting looks more natural to me: the arms are fixed but their performance vary according to external environmental factors, eventually making each of them optimal in some context. I can see the applications of this setting, but I cannot see the applications of the setting presented in the paper.

Then I believe that, although the main paper is not very technical, the presentation makes it difficult to grasp the main technical insights of the analysis. Many arguments are discussed but not properly sketched, and it is hard to precisely correlate the two « challenges » with the actual regret analysis. I decided to check the proof of the regret bound in appendix, and then things got even worse: the appendix is so badly organized that it requires at least 4 screens to check any single results, due to no clear proof scheme and various technical results spread everywhere. Overall, after considerable efforts I did not succeed in understanding precisely how each result is used in the regret analysis. I would suggest the authors to re-organize the paper by properly writing the regret analysis, and along the analysis pointing out how the main technical results are used, that would themselves be properly proved in dedicated sections. In appendix their should not be proof sketches but rigorous proofs, and discussions about intuitions should be clearly separated from the proofs. Furthermore, the unbounded context case seems to be tackled in an extremely complicated way. I do not get the point of this, it does not even seem important in the main paper, maybe tackling the bounded case only would be sufficient and remove some confusion. Second, I do not get why the authors do not eventually use the proof for bounded context with $x_{max}=O(polylog(T))$, which is eventually true with high probability with light-tailed contexts.

Overall, it seems to me that the paper requires significant re-writing to (1) make the proofs rigorous and easy to check (or even just that an expert non-author reader might be able to check them), (2) clarify the contributions by focusing on precise interesting cases,  (3) more precisely linking the « challenges » with the regret analysis. Furthermore, the setting appears as artificial in the current version of the paper.

**Questions:**

* The LAC condition implies a mode of the context distribution in zero, and some flatness around $0$. I am wondering how instrumental is this property to guarantee the implicit exploration of Greedy and the not-too-small sub-optimality gap. For instance, with a Gaussian in mind, do the guarantees generalize if we assume the existence of a mode and shift the condition to be on the norm of $x-a$ for some vector $a$? The adaptation does not seem direct to me because sampling around $0$ facilitate that diverse directions are sampled, making the gap emerge.

* Could you be provide some justification of the settings, i.e. some applications where it would make sense to assume that $K$ arms are drawn i.i.d from a context distribution at each time step, with independence between steps?

* What is special about unbounded context that make it relevant to keep the distinction in the paper?

* The dependency in $t$ in challenge 2 looks fishy to me: I don’t see why the probability bound should depend on $T$ (or 1): the probability only depends on the context distribution and K, so the bound should only depends on $\epsilon$. Furthermore, by independence the same bound should hold for all $t$. Am I missing something? In addition, assuming the result is true, how is the $1/\sqrt{T}$ shaved off in the analysis? Because I would assume that the regret analysis would consist in multiplying the error by $T$ under the case the event does not hold.

* Additional related question: what $\epsilon$ is used in the analysis?

---

> ### Author Rebuttal · Authors · 2024-08-07
>
> Thank the reviewer for taking the time to review our paper and for the comments. However, we believe there is a fundamental disagreement between the reviewer's comments and the focus of our study. We hope to remedy this through open-minded discussion. We strongly believe and remain very confident that our work presents significant values and very important results that advance the existing knowledge about greedy algorithms in contextual bandits.
>
> ---
> **[W1]**
> With all due respect, we are very concerned that the reviewer does not even agree with the stochastic context setting and suggests a new setting (the reviewer thinks it "looks more natural") where features are fixed but the parameter is sampled. Then, this leads to the reviewer's conclusion that our contribution should be somehow discounted, which we believe is an unfair evaluation.
>
> This is problematic because **all of the existing greedy contextual bandit literature so far has been proposed under stochastic contexts, focusing on how such stochasticity in context allows for greedy algorithms to achieve sub-linear regret** [8, 16, 25, 28]. In fact, the previous literature focused only on very few distributions (e.g., Gaussian and uniform), as we state in the paper, which our work significantly expand and even achieve much smaller regret.
>
> Such a comment not only **rejects the entire literature on greedy contextual bandits** but also **raises concerns about whether our work can be adequately evaluated compared to the existing results** if the problem setting itself is disregarded.
> Our work significantly expand the set of context distributions admissible for greedy bandit algorithms, which has been an open question among many researchers.
>
> There is a rich history of contextual bandits with stochastic contexts even beyond the greedy bandit literature [5, 8, 15, 16, 25, 28]. We hope for a fair re-evaluation based on the established and widely accepted problem setting compared to the relevant literature within the field.
> Please do not take this personally; we ask that the reviewer put themselves in the authors' shoes and consider whether there is any meaningful feedback to be gained from a comment that discredits the problem itself and disregards the entire history of research on greedy contextual bandits (and somehow hope that the authors' hard work can evaluated fairly). This is the main problem setting that all relevant literature has been working on. We respectfully but strongly dispute this point.
>
> ---
> **[W2]**
> Firstly, we strongly disagree with the reviewer's statement that "the main paper is not very technical." We are unsure how our results and analysis could be perceived as "not very technical" to begin with. While we appreciate the feedback on improving the presentation in the appendix and will incorporate appropriate edits, we feel that the assertion that our paper is "not very technical" is unfounded. Throughout our paper, we present rigorous analysis and stronger results than what has previously been known in the literature.
>
> Despite the disagreement, we are more than happy to provide a proof sketch—some details are already presented in Appendix D, but here, we will present more thorough arguments. We first demonstrate how our two challenges can lead to logarithmic regret.
>
> *Challenge 1: $\sqrt{t}$-rate $\ell_2$ concentration*
>
> First Challenge can lead the $\ell_2$ statistical resolution of the estimator.
> Under the Challenge 1, we can get
> \begin{align*}
> |X_{a(t)}^\top (\hat{\theta}_{t-1} -\theta^\star)| \leq cx_{\max}\frac{\sqrt{d}}{\sqrt{ \lambda_\star t}}, \quad
> |X_{a^\star(t)}^\top (\hat{\theta}_{t-1} -\theta^\star)| \leq cx_{\max}\frac{\sqrt{d}}{\sqrt{ \lambda_\star t}}
> \end{align*}
> holds with high probability.
> Here, $c = \tilde{O}(1)$ constant.
>
> However, this resolution in insufficient for the logarithmic regret, it can only get $O(\sqrt{T})$ regret bound.
>
> *Challenge 2: Logarithmic regret*
>
> But with the help of the Challenge 2 (margin condition), we can get logarithmic expected regret upper bound.
> When the greedy policy select $a(t)$, it means that
> \begin{align*}
> X_{a(t)}(t)^\top \hat{\theta}_{t-1}  \geq X_{a^\star(t)}^\top \hat{\theta}_{t-1}
> \end{align*}
> and by the definition of the optimal arm,
> \begin{align*}
> X_{a(t)}(t)^\top \theta^\star \leq X_{a^\star(t)}^\top \theta^\star
> \end{align*}
> holds.
> Under the Challenge 1, we get
> \begin{align*}
> \operatorname{reg}'(t):= X_{a^\star(t)}(t)^\top \theta^\star-  X_{a(t)}(t)^\top \theta^\star \leq 2cx_{\max}\frac{\sqrt{d}}{\sqrt{ \lambda_\star t}}.
> \end{align*}
>
> Next, we define the event $E$ as the event of $\mathbf{X}(t)$ with regret occurring as $\operatorname{reg}'(t) > 0$.
> Under the event $E$, the suboptimality gap of $\mathbf{X}(t)$ satisfies
> \begin{align*}
> \Delta(\mathbf{X}(t)) \leq   \operatorname{reg}'(t) \leq 2cx_{\max}\frac{\sqrt{d}}{\sqrt{ \lambda_\star t}}
> \end{align*}
> and by the Challenge 2, we get
> \begin{align*}
> \mathbb{P}[\operatorname{reg}'(t) >0] \leq 2c x_{\max}C_{\Delta}  \frac{\sqrt{d}}{\sqrt{ \lambda_\star t}} +  \frac{1}{\sqrt{T}} \leq 3cx_{\max}C_{\Delta} \frac{\sqrt{d}}{\sqrt{ \lambda_\star t}}
> \end{align*}
> By combining two things, we can bound the expected regret as
> \begin{align*}
> \operatorname{reg}(t) &= \mathbf{E}[\operatorname{reg}'(t)] \leq 3cx_{\max}C_{\Delta}  \frac{\sqrt{d}}{\sqrt{ \lambda_\star t}} \times 2cx_{\max}\frac{\sqrt{d}}{\sqrt{ \lambda_\star t}}\\
> &= 6c^2x_{\max}^2 C_{\Delta} \frac{d}{\lambda_\star t}.
> \end{align*}
>
> Using this argument, our only remainning goal is bounding two constants in Challenge 1 and 2.
>
> **[Bounded and unbounded contexts]**
> Please see the [Q3] parts.

---

> ### Author Response · Authors · 2024-08-07
> **Answers to Reviewer's Questions**
>
> **[Q1]**
> **No, your first sentence (premise) in the question is incorrect:** The LAC condition **does not "imply a mode of the context distribution in zero.""**
>
> It seems there might be a misunderstanding of the LAC condition's properties. If you need any clarification, please feel free to let us know.
>
> Regardless, if you are still interested in a shifted mean, it turns out that this is not an issue at all. Our regret bound still holds. For example, for the Gaussian with a shifted mean, please see Appendix C.1.
>
>
> Here is a imple explanation. Let the mean zero contexts $\mathbf{X}(t)$'s density as $f(x)$ with LAC fuction ${L}(\cdot)$.
> Then, the shifted contexts with mean $\mu$ has density $g(x) = f(x + \mathbf{\mu})$ and see that
>
> \begin{align*}
>  \lVert \nabla \log g(x) \rVert_{\infty} &= \lVert \nabla \log f(x + \mu) \rVert_{\infty} \\
>  &\leq L( \lVert x + \mu\rVert_{\infty}) \\
>  &\leq L( \lVert x \rVert_{\infty} +\lVert \mu\rVert_{\infty})
> \end{align*}
>
> Hence the density $g(x)$ has LAC with function $L'(x) = L (x + \lVert \mu \rVert_{\infty})$
> and since $\lVert \mu\rVert_{\infty} = O(1)$, it has the same rate.
>
> **[Q2]**
> We believe **there is a misunderstanding in the comment here**. We do NOT assume i.i.d. of arms at all. We only assume the independence of the entire arm set over time (note there is a clear difference between i.i.d.-ness vs. independence only; independent of each arm vs. independence of the arm set!), and hence we allow that arms can even be dependent on each other in a given arm set. A rich body of linear contextual bandit studies assumes stochastic contexts and independence of the contexts (often even stronger i.i.d. contexts) over time [5, 8, 15, 16, 25, 28].
>
> In particular, all of the relevant greedy contextual bandit literature assumed even i.i.d.-ness (often i.i.d. for each arm), which is stronger than our problem setting that only requires independence of the arm set (not each arm or i.i.d.). We emphasize again that $X_i(t), X_j(t)$ can be correlated for $i, j \in [K]$. Hence, we strongly believe that justification is provided in order for our work to be fairly evaluated (note again our problem setting is even weaker than the previous problem setting in greedy contextual bandits).
>
> **[Q3]**
> We are happy to address this question. The reason we present both bounded and unbounded contexts (particularly why we include analysis for unbounded context) is because previous works in greedy contextual bandits largely utilized unbounded noise (e.g., Gaussian) in feature setups. Hence, for fair comparisons with the existing results, we feel that it is fair to include unbounded cases [25, 28, 29].
>
> There is also a technical reason to separately consider the two cases. Boundedness assumptions affect the regret bound differently. Hence, we would like to use appropriate soft boundedness of unbounded contexts. If we were to assume conventionally accepted $\ell_2$ boundedness for light-tail unbounded distributions, there could be a dimension dependency for the high-probability bound. Generally, it has a $\tilde{O}(\sqrt{d})$ bound which is often ignored but can be significant. However, since we assume $\psi_1$ or $\psi_2$ norm boundedness for unbounded contexts, it is dimension-free and we can get a tighter bound even in a more transparent manner.
>
> **[Q4-5]**
> Please see the answer to W2 part; we think it will help understand your questions in the proof sketch. We would be happy to answer any further questions.

---

> > ### Comment · Reviewer_Fhbd · 2024-08-11
> >
> > Thank you very much for your in-depth response. After reading it and discussing with other reviewers I acknowledge that
> > * My concern about the setting is an opinion and should not be a motive for rejection.
> > * My technical questions are answered, thank you.
> >
> > Furthermore, it seems that some reviewers have been able to check in detail the theoretical results, leading them to strongly support the paper. In that case, I am happy to increase my score and recommend acceptance. However, I still believe that some rewriting might be beneficial to improve the clarity of the final version of the paper.

---

> > > ### Author Response · Authors · 2024-08-11
> > > **Thank you**
> > >
> > > Dear Reviewer Fhbd,
> > > Thank you for your open-mindedness in recognizing our work and for the increased score. We will make sure to improve our writing, particularly in the appendix, for the final version.

---

### Official Review · Reviewer_PGWV · 2024-07-09

**Soundness:** 2
**Presentation:** 2
**Contribution:** 2
**Rating:** 5
**Confidence:** 3

**Summary:**

This paper aims to expand the range of distributions that can be used efficiently in exploration-free greedy linear contextualized bandits. For this purpose, a new condition called Local-Anti Concentration is introduced. It is claimed that different distributions from the exponential family satisfy this property and they do not require the margin property to achieve $O(poly logT)$ regret.

**Strengths:**

For the Gaussian, the authors show that LAC condition holds. Hence, they show an improvement in the regret without assuming any margin condition.

**Weaknesses:**

1. 89, 264: It’s unclear what the entries of this Gram matrix represent.

2. 24-25: It’s not explained why healthcare and clinical domains might find exploration infeasible or unethical.

3. 57: The specifications of margin condition are not explained.

4. Even though there are experiments in the paper, in the guideline the answer to the question 5 is marked as N/A. It's unclear whether the code belongs to the authors or another researcher and hence, the answer. If the code belongs to the authors, the answer should have been YES or NO.

5. Lines 63 to 67 are repeated word for word in 106 to 109, except for the addition of the word "bandit". Please delete one of these sections.

6. 120: It is not clear on what D and B are supposed to be. The authors can add the word "sets" to be more clear.

7. 526-540: Appendix C.1 is not written extensively; the proof is provided only for the Gaussian distribution.  This is the main contribution of the paper as stated in 91-92 and used in different places such as 311. Without the proof, it’s unclear how well the rest of the statements for the distributions hold.

8. 490: There is a typo as these algorithms are stated as linTS and linUCB instead of LinTS and LinUCB.

9. 114: There is another typo: "unecessary" should be "unnecessary".

10. 352: Experiment results are not discussed extensively, making it hard to understand how the experiments support the claims.

**Questions:**

- 352: LinTS and LinUCB algorithms are show in the Figures in the Experiments Section. However, it is unclear what these algorithms are. Were they designed by the authors or are they the work of other researchers? In 1444, they are stated as existing bandit algorithms. If so, please provide the appropriate references.

- 246: What do the authors mean by if two challenges are “satisfied”? Do you mean “overcome”?

**Limitations:**

- Technical contribution: The most significant proof, which is the proof that the given distributions satisfy the LAC condition, was left out.

---

> ### Author Rebuttal · Authors · 2024-08-07
>
> Thank you for your feedback and for the opportunity to discuss our work with you. Most of your feedback appears to be clarifications and suggestions for stylistic edits or minor typos, which we appreciate. However, none of your comments seem critical enough to warrant a rating of "Reject: For instance, a paper with technical flaws, weak evaluation, inadequate reproducibility, and/or incompletely addressed ethical considerations."
>
> Hence, we sincerely ask for a re-evaluation of our work. We strongly believe that the significance and impact of our results are very high, significantly expanding what had previously been known and doing so at a very timely moment. And, please let us know If you have any remaining questions.
>
> ---
> **[W1]** The definition of the Gram matrix is $\Sigma(t):=\sum_{s=1}^t X_{a(s)}(s)X_{a(s)}(s)^\top$ and is stated in Algorithm 1. Readers of the linear bandit literature are familiar with this term, but we will provide the formal definition of the Gram matrix earlier in the text.
>
> **[W2]** In healthcare, if a healthcare provider treats a patient with what appears to be sub-optimal (non-greedy) just for the purpose of exploration to learn the effect of a new treatment (not for the benefit of the particular patient being treated), it would be clearly unethical or often not feasible in actual healthcare practices (beyond clinical trials). This aspect has been discussed in various previous literature, including [8].
>
> **[W3]** The exact definition of the margin condition is stated in Challenge 2 (line 266).
>
> **[W7]** (LAC examples)
> First, we prove results in line 195. For exponential, Laplace, and Student's t, we prove results for 1-dimensional distribution. However, using Proposition 1, we can extend this to multi-dimensional contexts.
>
> * Exponential: $\nabla \log f(x) = -\lambda $ and hence $| \nabla \log f(x) | \leq \lambda$.
> * Uniform: $\nabla \log f(\mathbf{x}) = \mathbf{0}$ and hence $||\nabla \log f(\mathbf{x}) ||_\infty \leq 1$. (multi-dimensional)
> * Laplace: $f(x) = \frac{1}{2 b} \exp \left(-\frac{|x-\mu|}{b}\right)$ and then $|\nabla \log f ({x})| = |\frac{1}{b}|$.
> * Student's t: $f(x)= \Gamma\big(\frac{\nu+1}{2}\big) / \sqrt{\nu \pi} \Gamma\big(\frac{\nu}{2}\big) \cdot \big(1+\frac{x^{2}}{\nu}\big)^{-(\nu+1)/2}$ then $|\nabla \log f(x)|=|\nabla \frac{\nu +1}{2}\log(1 + \frac{x^2}{\nu})| = \frac{(\nu+1)x}{\nu + x^2}\leq C(\nu)$ for some constant $C(\nu)>0$.
>
> **[W10]** Our experiment demonstrates that a greedy algorithm clearly outperforms other widely used algorithms (LinUCB and LinTS) that balance exploration and exploitation for common distributions. It appears that for those distributions, any type of exploration is unnecessary -- greedy algorithms suffice, which is the main assertion of this paper. Due to space constraints, the discussion is provided in Appendix N.
>
> **[W4-6, 8-9]** Thank you for pointing out typos and stylistic suggestions. We will incorporate the appropriate edits suggested by the reviewer.
>
> ---
> **[Q1]** LinUCB and LinTS are the most widely used linear contextual bandit algorithms. The LinUCB algorithm follows the paper by Abbasi-Yadkori et al. (2011), and the LinTS algorithm follows the paper by Agrawal and Goyal (2013) to conduct the experiments.
>
> **[Q2]** We mean that if the two challenges regarding the context distribution are satisfied, i.e., if two key constants exist, logarithmic regret can be achieved.
> However, to get the exact upper bound, rather than proving existence of the constant bounding the two constant is essential, and we discuss this throughout the whole paper.
>
> **[Details for two challenges and regret]**
> The following part breifly describe how to challenges can make logartihmic regret.
>
> *Challenge 1: $\sqrt{t}$-rate $\ell_2$ concentration*
>
> First Challenge can lead the $\ell_2$ statistical resolution of the estimator.
> Under the Challenge 1, we can get
> $$
> |X_{a(t)}^\top (\hat{\theta}_{t-1} -\theta^\star)| \leq c x_{\max} \frac{\sqrt{d}}{\sqrt{ \lambda_{\star} t}}, \quad
> |X_{a^\star(t)}^\top (\hat{\theta}_{t-1}-\theta^\star)| \leq c x_{\max} \frac{\sqrt{d}}{\sqrt{ \lambda_{\star} t}}
> $$
> holds with high probability.
> Here, $c = \tilde{O}(1)$ constant.
>
> However, this resolution in insufficient for the logarithmic regret, it can only get $O(\sqrt{T})$ regret bound.
>
> *Challenge 2: Logarithmic regret*
>
> But with the help of the Challenge 2 (margin condition), we can get logarithmic expected regret upper bound.
>
> When the greedy policy select $a(t)$, it means that
> \begin{align*}
> X_{a(t)}(t)^\top \hat{\theta}_{t-1}  \geq X_{a^\star(t)}^\top \hat{\theta}_{t-1}
> \end{align*}
> and by the definition of the optimal arm,
> \begin{align*}
> X_{a(t)}(t)^\top \theta^\star \leq X_{a^\star(t)}^\top \theta^\star
> \end{align*}
> holds.
>
> Under the Challenge 1, we get
> \begin{align*}
> \operatorname{reg}'(t):= X_{a^\star(t)}(t)^\top \theta^\star-  X_{a(t)}(t)^\top \theta^\star \leq 2cx_{\max}\frac{\sqrt{d}}{\sqrt{ \lambda_\star t}}.
> \end{align*}
>
> Next, we define the event $E$ as the event of $\mathbf{X}(t)$ with regret occurring as $\operatorname{reg}'(t) > 0$.
> Under the event $E$, the suboptimality gap of $\mathbf{X}(t)$ satisfies
>
> \begin{align*}
> \Delta(\mathbf{X}(t)) \leq   \operatorname{reg}'(t) \leq 2cx_{\max}\frac{\sqrt{d}}{\sqrt{ \lambda_\star t}}
> \end{align*}
> and by the Challenge 2, we get
> \begin{align*}
> \mathbb{P}[\operatorname{reg}'(t) >0] \leq 2c x_{\max}C_{\Delta}  \frac{\sqrt{d}}{\sqrt{ \lambda_\star t}} +  \frac{1}{\sqrt{T}} \leq 3cx_{\max}C_{\Delta} \frac{\sqrt{d}}{\sqrt{ \lambda_\star t}}
> \end{align*}
> By combining two things, we can bound the expected regret as
> \begin{align*}
> \operatorname{reg}(t) &= \mathbf{E}[\operatorname{reg}'(t)] \leq 3cx_{\max}C_{\Delta}  \frac{\sqrt{d}}{\sqrt{ \lambda_\star t}} \times 2cx_{\max}\frac{\sqrt{d}}{\sqrt{ \lambda_\star t}}\\
> &= 6c^2x_{\max}^2 C_{\Delta} \frac{d}{\lambda_\star t}.
> \end{align*}
>
> Using this argument, our only remainning goal is bounding two constants in Challenge 1 and 2.

---

> > ### Comment · Reviewer_PGWV · 2024-08-09
> >
> > I would like to thank the authors for all their clarifications. I have a few clarifying questions before I reassess my score.
> >
> > 1. Equation (3) suggests that the probability that the minimum sub-optimality gap is small is also small. Specifically, the probability that the gap falls below $\varepsilon$ is proportional to $\varepsilon$. The proportionality constant, $C_{\Delta}(t)$ has been further bounded in Theorem 3, and a growth rate of $O(\log K)$ has been established. I suspect that this is a weak bound. Here's a counterexample:
> >
> > Assume $d=1$, and randomly sample $K$ points uniformly from $[0,1]$. The suboptimality gap would be proportional to the gap between the first and second order statistics $X_{(1)} - X_{(2)}$, where we have defined $X_{(1)} > X_{(2)} > \cdots X_{(K)}$. Now, as $K$ increases, I think the minimum gap should shrink (larger the number of contexts, lower the sub-optimality gap). Hence, for a large number of contexts, say $O(1/\varepsilon)$, the probability in (3) may no longer be small, thus, violating the LAC property. Is my understanding correct? To verify my hypothesis, I suggest the authors a simple simulation: Sample $K$ points between $[0,1]$ uniformly at random. In x axis, plot $K$, and in the y axis, plot $X_{(1)} - X_{(2)}$. The authors can use $K = 5$, $100$, $500$, $1000$.
> >
> > 2. Additionally, I have a clarification question. Is the expectation in the diversity constant equation (Equation 2) with respect to contexts $X(t)$, or both $X(t)$ and $a(t)$? Also, I believe the second $X_{a(t)}(t)$ should have a transpose sign.

---

> > > ### Author Response · Authors · 2024-08-10
> > >
> > > We sincerely appreciate your feedback and your willingness to reassess your score. We are more than happy to provide the responses to your questions.
> > >
> > > ---
> > > **[1]** It is a great question.
> > > First, following the setup you mentioned, let $Z := X_{(1)} - X_{(2)}$.
> > > If the density of $Z$ is upper bounded by some constant $M$, then for any $\epsilon > 0$,
> > >
> > > $$
> > > \mathbb{P}[0 \leq Z \leq \epsilon] \leq M\epsilon
> > > $$
> > > holds. We emphasize again that $M$ is independent of the choice of $\epsilon$; it depends only on the context distribution and $K$.  In the proof of regret analysis, we choose an arbitrary $\epsilon$. Hence, we want this inequality to hold for all $\epsilon$.
> > > Then, for a fixed $K$, our next question becomes: "What is the upper bound of $M$?"
> > >
> > > Our margin constant $C_\Delta$ is closely related to the maximum density by the above argument, and we bound the maximum density of the suboptimality gap of contexts throughout the paper.
> > > (Our definition of Challenge 2 includes an additional term $\frac{1}{\sqrt{T}}$ due to some technical reasons, but it is weaker than the above inequality).
> > >
> > > For a 1-dimensional uniform distribution, it is widely known that $M \asymp K$ (you can refer to any textbook on extreme value theory, such as [12]). However, for a $d$-dimensional uniform distribution within a ball, $M$ **decreases** rapidly as $d$ increases, which is a beneficial effect of high dimensionality. For details, please see Appendix I.8, where this matter is explicitly discussed.
> > >
> > > For Gaussian or other light-tailed distributions, $M$ exhibits a logarithmic dependency on $K$. Some asymptotic results for $M$ with respect to $K$ are proven in [12], and we provide a non-asymptotic bound that maintains this logarithmic dependency.
> > > This corresponds to the unbounded contexts case in our paper, and all of our results match the known asymptotic results.
> > >
> > > Regarding the experiment you mentioned, the difference between order statistics $X_{(1)} - X_{(2)}$ (two extremes) is well-known, with its asymptotic distribution understood. As stated earlier, for the 1-dimensional uniform distribution, this difference is proportional to $K$, while for Gaussian or light-tailed distributions, it is proportional to $\log K$. For the $d$-dimensional uniform distribution, it necessarily depends on both $K$ and $d$, and decreases with $d$.
> > > Given this solid theoretical foundation, we expect experimental results to align with these predictions. However, if you would like additional experimental results, we can provide them. We have also adjusted our approach to derive a non-asymptotic bound that applies to the bandit problem, and our results remain valid under this scenario.
> > >
> > > ---
> > >
> > > **[2]**
> > > The expection is taken only with respect to $\mathbf{X}(t)$.
> > > At time $t$, for any fixed history $H_{t-1}$, we perform the greedy policy with the estimator $\hat{\theta}_{t-1}$.
> > >
> > > Hence, when the entire set of contexts $\mathbf{X}(t) = (X_1(t), \dots, X_K(t))$ is revealed, $a(t)$ is determined immediately given the history $H_{t-1}$.
> > > Then the exact statement is:
> > >
> > > "At time $t$, for any history $H_{t-1}$,
> > >
> > > $$
> > > \mathbb{E} {\tiny{\mathbf{X}(t)}} [X_{a(t)}(t) X_{a(t)}(t)^\top] \succeq \lambda_\star I_d
> > > $$
> > >
> > > holds with some $\lambda_\star > 0$."
> > >
> > > In the proof, we proved a stronger statement:
> > >
> > > "For any greedy policy with any $\theta \in \mathbb{R}^d$,
> > >
> > > $$\mathbb{E}{\tiny{\mathbf{X}(t)}} \left[X_{a(t)}(t) X_{a(t)}(t)^\top\right] \succeq \lambda_\star I_d$$
> > >
> > > holds with some $\lambda_\star > 0$."
> > >
> > > And yes, there should be a transpose sign. Thank you for pointing this out. Hope our answers provided clarification. If you have any questions, please let us know.

---

> > > > ### Comment · Reviewer_PGWV · 2024-08-11
> > > >
> > > > Thank you for addressing my questions. While I haven't had the opportunity to thoroughly go over the entire referenced paper to verify the statement, it appears to be accurate. However, due to the need for improvements in the clarity and organization of your paper, I have adjusted my score to 5.

---

> > > > > ### Author Response · Authors · 2024-08-11
> > > > > **Thank you**
> > > > >
> > > > > Dear Reviewer PGWV,
> > > > > Thank you for your open-mindedness in recognizing our work and for the increased score. We will make sure to improve our writing, particularly in the appendix, for the final version.

---

### Official Review · Reviewer_qUQr · 2024-07-12

**Soundness:** 4
**Presentation:** 4
**Contribution:** 4
**Rating:** 8
**Confidence:** 3

**Summary:**

The paper proposes a novel condition for context distribution, called *Local Anti-Concentration (LAC)*. Under LAC, the authors prove the regret of greedy algorithms for stochastic contextual linear bandits is $\mathcal{O}(\mathrm{poly} \log T)$, without additional margin assumption. The efficacy of the greedy approach for various distributions is validated numerically as well.

**Strengths:**

- Clearly, very well-written
- Clear-cut contributions that significantly improve greedy algorithms for contextual linear bandits, including a provably larger class of context distributions that can achieve $\mathcal{O}(\mathrm{poly} \log T)$ regret without additional margin condition and other technical contributions.
- Numerical verification

**Weaknesses:**

- The authors should provide (possibly not-to-far-fetched) distributions in which the LAC fails.
- The initial parameter $\theta_0$ is mentioned briefly but not much discussed. What is the dependency of the regret on $\theta_0$? How was $\theta_0$ chosen for the experiments? In practice, should one choose $\theta_0$ randomly, or is it okay to fix it? I feel this plays an important role, as the initial "exploration" (til the diversity becomes positive) heavily depends on $\theta_0$.


**(Minor) Typography suggestions**
- Some of the sentences in Section 1.2 overlap with those in the paragraph above Section 1.1. I think it would be more appropriate if Section 1.2 is absorbed into the beginning of Section 1, and Section 1.1 becomes a paragraph (\paragraph{..})
- The pseudocode in Algorithm 1 seems wrong...? the while loop should be an If-Else.
- In pg. 6, "Consider the desnity" -> "Consider the density"
- If accepted, the authors should include the full regret bound from Appendix B.3 in the main text.

**Questions:**

- Is LAC necessary for the polylog(T) regret of the greedy algorithm, or at least close to it? In other words, if the distribution is not LAC, does greedy always fail? I would be curious to see the numerical performance of greedy algorithms for distributions that are not LAC.
- Does this work for generalized linear bandits as well? How about generally structured bandits (although this seems quite unlikely) and kernelized linear bandits (this, I have no idea)?
- There was an interesting paper [1] in which, for linear bandits with a rich (continuous) action set, sublinear regret *implies* a lower bound on the minimum eigenvalue of the design matrix. Of course, in this case, the margin doesn't make sense, and there is a $\Omega(d\sqrt{T})$ regret lower bound, but as this paper and [1] are conceptually similar (to my eyes), would the greedy algorithm achieve $O(d\sqrt{T})$ regret? Even if not, it would also be nice to include some discussions regarding this in the paper.
- (minor)



[1] https://proceedings.mlr.press/v206/banerjee23b.html

**Limitations:**

Yes

---

> ### Author Rebuttal · Authors · 2024-08-07
>
> Thank you very much for recognizing the value of our results. We appreciate your feedback and are happy to provide our responses to your comments.
>
>
> ---
> **[W1]**
> We are happy to address your comment and assure you that this should not be considered a weakness. Context distributions with discrete support, particularly fixed contexts, do not satisfy the LAC condition. However, for such distributions, the greedy algorithm fails (i.e., the greedy algorithms can incur linear regret) in the worst case. To the best of our knowledge, the LAC condition is the most inclusive condition for greedy algorithms to succeed. This is a crucial finding!
>
> For further details, please read Appendix M, which contains a detailed discussion on this matter. We would be more than happy to elaborate on this. More details are also discussed in the answer to **[Q1]**.
>
> ---
> **[W2]**
> Our algorithm and regret bound are valid regardless of the choice of $\theta_0$ as long as it is bounded (and we have the freedom to choose a parameter with an adequate bound). Note that we can show that the minimum eigenvalue of the gram matrix selected by the greedy policy increases linearly with time for any $\theta_0$. Hence, we can derive the same regret bound. We will include this discussion in the revision.
>
> ---
> **[Q1]**
> LAC is a sufficient condition, not a necessary condition. However, we do not know of any distributions (yet) that do not satisfy LAC but allow poly-logarithmic regret for greedy algorithms. There are good examples of distributions that are not LAC where greedy algorithms fail (or cannot achieve poly-logarithmic regret).
>
> Firstly, it is well known that the greedy algorithm can fail in the worst case when contexts are fixed. Another example where LAC does not hold is a context distribution supported in a low-rank space. In this case as well, it is known that logarithmic regret is impossible [25], and we are not aware of the success of the greedy algorithm to the best of our knowledge. It is very important to note that our work significantly expands what has been known to be admissible for greedy algorithms. LAC is the most general condition currently known to allow greedy algorithms to achieve poly-logarithmic regret.
>
> ---
> **[Q2]**
> Yes, our analysis extends to the GLM bandit as well for regular link functions as long as the link function has a bounded first derivative, which is commonly assumed in GLM bandit literature [14, 23]. As long as the concentration of the estimator is controlled by the gram matrix, most of the analysis is similar.
>
> In the case of the kernelized bandit, it can be expressed in the form of a linear contextual bandit through the RKHS formulation. However, it usually assumes eigenvalue decay, so the minimum eigenvalue can become arbitrarily small. Hence, it might be difficult to apply the same analysis but would be an interesting future direction. So far, we have not even known what was possible for linear contextual bandits. Extension to other parametric bandits or kernelized bandits would be interesting.
>
> ---
> **[Q3]**
> We appreciate your pointer to this related work. While the paper shares some common points of addressing the minimum eigenvalue of the gram matrix, our analysis significantly differs in that one of the key ingredients of our analysis is ensuring that the minimum eigenvalue of the gram matrix increases linearly (whereas in theirs it isn't). Furthermore, ensuring the margin condition from scratch is a key step that is not addressed in the reference you provided. However, analyzing the (though not logarithmic but some sublinear) performance of the greedy algorithm within the setup of the referred paper would be an interesting direction.

---

> ### Comment · Reviewer_qUQr · 2024-08-11
>
> Thanks for the detailed response, which have addressed all my questions. I will retain my score.
>
> But, please do make sure to take the other reviewers' concerns and suggestions on the paper's organization, including
> - table of contents (a suggestion of mine that I forgot to mention)
> - significantly reorganizing Appendix so that the readers can easily locate the main proof.
> - typos
> - etc

---

> > ### Author Response · Authors · 2024-08-11
> > **Thank you**
> >
> > Reviewer qUQr,
> >
> > Thank you very much for your continued support and for recognizing the value of our work. We will definitely incorporate your feedback, along with the other reviewers' suggestions, to improve our presentation in the appendix (including adding a table of contents at the beginning of the appendix). If you have any questions in the meantime, please feel free to reach out to us!

---

### Official Review · Reviewer_Lk9D · 2024-07-13

**Soundness:** 3
**Presentation:** 3
**Contribution:** 3
**Rating:** 7
**Confidence:** 2

**Summary:**

The paper addresses the problem of linear contextual bandits with randomly generated contexts from a distribution
$f$. The goal is to determine under which conditions on $f$ a greedy algorithm (outlined in Algorithm 1) achieves reasonable regret.

By introducing the notion of Local Anti-Concentration (LAC) in Definition 1, the authors demonstrate that if $f$ satisfies the LAC condition, then the greedy algorithm achieves poly-logarithmic regret. The LAC condition encompasses a wide range of distributions, such as Gaussian, uniform, Laplace, and Student’s t-distribution, making the results of the paper general and applicable across several frameworks.

**Strengths:**

The paper is well-written and addresses a very interesting problem. It makes a significant contribution by covering a large family of distributions for the contexts and demonstrating that, for this wide range, a greedy algorithm can achieve poly-logarithmic regret. The main results, Theorems 3 and 4, are noteworthy and can be of independent interest.

**Weaknesses:**

The only weakness I see is the lack of discussion about the optimality of the achieved regret concerning the parameters of interest, i.e.,
$d$ (dimension of the context),
$K$ (number of arms), and $\alpha$ (the parameter associated to the LAC condition). While the authors addressed this by stating, "Our focus here is not solely on attaining the sharpest regret bounds, although achieving poly-logarithmic regret is highly favorable," it would have been beneficial to discuss this matter in more detail.

**Questions:**

1. I would like to know the authors' thoughts on the necessity of the LAC condition. While the paper's focus is on showing that the LAC condition is sufficient, I am curious if there are significant distributions that do not satisfy the LAC condition but for which the greedy approach still achieves poly-logarithmic regret.

2. In line 269, the authors mention, "Eq.(3) is a relaxed version of the margin condition" and "The aforementioned existing literature explicitly assumes the condition to hold." Does this mean that the margin condition imposed in previous works can be dropped and relaxed to provide a bound on Eq.(3)?

3. I would appreciate it if the authors addressed my question in the weaknesses section.

**Limitations:**

The authors adequately address the limitations.

---

> ### Author Rebuttal · Authors · 2024-08-07
>
> Thank you very much for recognizing the value of our results. We appreciate your feedback and are happy to provide our responses to your comments.
>
> ---
> **[W1]**
> In order to validate optimality of contextual bandit algorithms under stochasticity, we need to derive proper lower bounds. However, existing studies ([7, 8, 5, 15]) that have examined poly-logarithmic regret under margin conditions (similar to our paper) have not discussed lower bounds or optimality either.
> One caveat is that since the greedy algorithm is not adaptive, the dimensionality and $K$ dependence can vary depending on context distributions. In such cases, deriving distribution-dependent lower bounds would be meaningful to determine optimality. However, we conjecture that such an analysis would be quite challenging.
> Nonetheless, it would be an interesting future direction to provide a lower bound under the margin condition, not just for greedy bandit algorithms but for linear contextual bandits in general.
>
> ---
> **[Q1]**
> In short, to our best knowledge, the distributions that satisfy LAC are currently the only ones proven to be admissible for greedy algorithms! Typical cases where LAC is not satisfied include:
>
> 1. Fixed or discrete context distributions.
> 2. Low rank or nearly low rank contexts.
>
> In case 1, it is known that the greedy algorithm fails for fixed contexts. In case 2, it is known that logarithmic regret is impossible [25].
> Additionally, for distributions with double exponential density (e.g., Gumbel distribution), LAC is not satisfied. However, with slight modifications, it might be possible to show a sublinear regret bound, but we are unsure how tight the bound can be (we do not know whether poly-logarithmic regret would be possible). It is very important to note that our work significantly expands what has been known to be admissible for greedy algorithms.
>
> ---
> **[Q2]**
> Previous literature **assumes** the existence of a constant satisfying the margin condition and regards it as a **fixed** constant. However, we **derive** the upper bound of the margin constant from the LAC density.
>
> We emphasize again that we **do not assume** the existence of the margin constant. For example, many previous studies [7, 8, 5, 15] assume that there exists $C_\Delta$ satisfying:
> $$ \mathbb{P}[X_{a^\star}(t)^\top \theta^\star - \max_{i \neq a^\star} X_{i}(t)^\top \theta^\star \leq \varepsilon] \leq C_\Delta \varepsilon. $$
> With this margin assumption, one can achieve logarithmic regret. However, our paper calculates and estimates the margin constant $C_\Delta$ from scratch, only assuming the densities of contexts are LAC.
>
> Also, note that if the above holds, then our equation of the margin condition (equation (3)) holds directly. Hence, our version of the margin condition is weaker. More importantly, we **do not even impose such an assumption** to start with! Instead, we prove that distributions in LAC automatically satisfy the margin condition which is a significant contribution.

---

> > ### Comment · Reviewer_Lk9D · 2024-08-08
> > **Rebuttal acknowledgment**
> >
> > I would like to thank the authors for the rebuttal. For now, I will maintain my current score. I plan to discuss the paper with the other reviewers and look forward to the author's discussions with them as well. I will update my score accordingly.

---

> > > ### Author Response · Authors · 2024-08-09
> > > **Thank you**
> > >
> > > Reviewer Lk9D, Thank you very much for recognizing the value of our work and for your support. If you have any questions in the meantime, please feel free to reach out to us!

---

### Author Response · Authors · 2024-08-09

We would like to express our sincere gratitude for taking the time to review our paper and for the opportunity to engage in this discussion. We encourage the reviewers to reflect on the following point: Tackling and improving a long-standing, critical open theoretical problem—where progress has previously been limited (with the community having previously recognized only Gaussian and Uniform distributions as effective for greedy algorithms)—holds significant value for both the bandit research community and the larger NeurIPS community.

- Identifying which context distributions enable greedy bandit algorithms to achieve a sublinear regret is a significant challenge, and the research community had not known much beyond Gaussian and Uniform.
- Constructing the common key characteristic of these distributions admissible for greedy algorithms marks an important milestone.
- Proving that the newly identified and characterized distributions actually allow for poly-logarithmic regret is a breakthrough that no previous work has achieved.
- Furthermore, we achieve this without assuming a margin condition—proving that these distributions naturally induce margins, which is a novel finding even for Gaussian distributions.

We strongly believe that contributions that this paper presents merit acknowledgment. We are confident that our research offers substantial value to the NeurIPS community. We remain open to making minor revisions to further clarify our contributions and to address any feedback or questions from the reviewers. If there are any additional comments or concerns, please do not hesitate to contact us. We truly appreciate the time and effort you have invested in reviewing our work.

---

### Decision · Program_Chairs · 2024-09-25

**Decision:**

Accept (poster)

**Comment:**

The fact that the greedy algorithm works well in practice when the contexts have regularity has been an interesting topic of research. Obtaining polylog(T) regret bounds for greedy algorithms in the past have relied on strong assumptions such as the margin condition. The authors did an excellent job in relaxing these assumptions. The newly introduced assumption has been specialized to various examples. I believe these altogether are great contributions towards fully understanding the role of context distribution in linear bandits, which has not been satisfactory in the contemporary literature.

Meanwhile, I recommend that the authors improve clarity and organization of the paper (in particular appendix) so that readers can appreciate the results and gain insight without checking all the details and the proofs.